# Ancient Rapanui genomes reveal resilience and pre-European contact with the Americas

J. Víctor Moreno-Mayar[1,2,3,14 ✉], Bárbara Sousa da Mota[3,4,14], Tom Higham[5,6], Signe Klemm[1,2], Moana Gorman Edmunds[7], Jesper Stenderup[1,2], Miren Iraeta-Orbegozo[1,8], Véronique Laborde[9], Evelyne Heyer[10], Francisco Torres Hochstetter[11], Martin Friess[10], Morten E. Allentoft[2,12], Hannes Schroeder[1], Olivier Delaneau[13 ✉] & Anna-Sapfo Malaspinas[3,4 ✉]

Rapa Nui (also known as Easter Island) is one of the most isolated inhabited places in the world. It has captured the imagination of many owing to its archaeological record, which includes iconic megalithic statues called moai[1]. Two prominent contentions have arisen from the extensive study of Rapa Nui. First, the history of the Rapanui has been presented as a warning tale of resource overexploitation that would have culminated in a major population collapse—the 'ecocide' theory[2–4]. Second, the possibility of trans-Pacific voyages to the Americas pre-dating European contact is still debated[5–7]. Here, to address these questions, we reconstructed the genomic history of the Rapanui on the basis of 15 ancient Rapanui individuals that we radiocarbon dated (1670–1950 CE) and whole-genome sequenced (0.4–25.6×). We find that these individuals are Polynesian in origin and most closely related to present-day Rapanui, a finding that will contribute to repatriation efforts. Through effective population size reconstructions and extensive population genetics simulations, we reject a scenario involving a severe population bottleneck during the 1600s, as proposed by the ecocide theory. Furthermore, the ancient and present-day Rapanui carry similar proportions of Native American admixture (about 10%). Using a Bayesian approach integrating genetic and radiocarbon dates, we estimate that this admixture event occurred about 1250–1430 CE.

Rapa Nui, also known as Te Pito o Te Henua ('the navel of the world'), is one of the most isolated inhabited places in the world[1]. Located in the Pacific, on the easternmost tip of the Polynesian Triangle, it lies 3,700 km west of South America and more than 1,900 km east of the closest inhabited island. Despite the remoteness of Rapa Nui, archaeological and genetic evidence shows that Polynesian peoples from the west had already reached the island about 1250 CE[8–10]. The following five centuries saw the Rapanui, the inhabitants of Rapa Nui, develop a culture characterized by iconic giant stone statues (moai) and monumental stone platforms (ahu)[1,11]. Owing to the isolation of Rapa Nui, Europeans reached the island only in 1722 CE. Over the years, European visitors had a devastating impact on the Rapanui as they killed local inhabitants and introduced deadly pathogens that the islanders had not been exposed to before[1,12]. Furthermore, in the 1860s, Peruvian slave raiders kidnapped a third of the population, and only a few were repatriated after international condemnation of the slaving practices[13]. Subsequently, a smallpox outbreak decimated the Rapanui population and it fell to an estimated 110 individuals[13]. Although the island and its people have been extensively studied using archaeology, anthropology

and genetics, there are two key features of the Rapanui demographic history, pre-dating European contact, that remain contentious: a potential human-mediated major population collapse in the 1600s (an 'ecological suicide'); and a possible trans-Pacific contact between the Rapanui and Native Americans.

Rapanui history has been presented as a warning tale for humanity's overexploitation of resources. According to the theory of ecological suicide ('ecocide'), before European contact, the Rapanui would have deforested the island and decimated the local fauna to maintain a flourishing culture and a growing population of about 15,000 individuals[2,3,14]. Consequently, resource scarcity would have led to the so-called Huri Moai cultural phase—a period of famine and war that would have escalated to the point of cannibalism and ultimately culminated in a population and cultural collapse in the 1600s, abruptly ending statue carving[2,3,15]. European visitors to the island in the 1700s estimated that the Rapanui population size varied between 1,500 and 3,000 individuals, which corresponds to between 10% and 20% of the population surviving the population collapse proposed by the ecocide theory[16]. Although it is well established that the environment in Rapa

[1]Globe Institute, Faculty of Health and Medical Science, University of Copenhagen, Copenhagen, Denmark. [2]Lundbeck Foundation GeoGenetics Centre, Globe Institute, University of Copenhagen, Copenhagen, Denmark. [3]Department of Computational Biology, University of Lausanne, Lausanne, Switzerland. [4]Swiss Institute of Bioinformatics, Lausanne, Switzerland. [5]Department of Evolutionary Anthropology, University of Vienna, Vienna, Austria. [6]Human Evolution and Archaeological Science (HEAS) Network, University of Vienna, Vienna, Austria. [7]Independent Rapanui archaeologist, Hanga Roa, Chile. [8]School of Archaeology, University College Dublin, Dublin, Ireland. [9]Direction Générale Déléguée aux Collections, Muséum national d'Histoire naturelle, Paris, France. [10]Eco-anthropologie (EA), Muséum national d'Histoire naturelle, CNRS, Université Paris Cité, Musée de l'Homme, Paris, France. [11]Mankuk Consulting & Services, Santiago, Chile. [12]Trace and Environmental DNA (TrEnD) Laboratory, School of Molecular and Life Sciences, Curtin University, Perth, Western Australia, Australia. [13]Regeneron Genetics Center, Tarrytown, NY, USA. [14]These authors contributed equally: J. Víctor Moreno-Mayar, Bárbara Sousa da Mota. ✉e-mail: morenomayar@gmail.com; olivier.delaneau@regeneron.com; annasapfo.malaspinas@unil.ch

Nui was affected by anthropogenic activity (for example, deforestation), it remains unclear whether or how these changes would have led to a population collapse[1]. Several lines of bioanthropological, archaeological and historical evidence have been used to challenge the ecocide scenario[4,16–18]. Yet the collapse hypothesis has remained very popular[2].

The depletion of wood for canoe building and renovation eventually led to the isolation of the island owing to the abandonment of long-distance seafaring—a hallmark of Polynesian cultures. However, several pieces of evidence suggest that Rapa Nui did not constitute the easternmost point of long sea voyages and that Polynesian peoples eventually reached the Americas before Columbus[19–23] (but see refs. 24,25). Genetic studies on present-day individuals have supported such contact. Present-day Rapanui were found to harbour Native American and European admixture in their genomes[5]. Notably, in that work, Native American admixture (dated 1280–1495 CE) was estimated to pre-date European admixture (dated 1850–1895 CE). More recently, Native American admixture was detected not only in present-day individuals from Rapa Nui, but also from Rapa Iti, Tahiti, Palliser, Nuku Hiva (North Marquesas), Fatu Hiva (South Marquesas) and Mangareva[7]. In that study, the Native American gene flow in the different islanders was dated between 1150 (South Marquesas) and 1380 CE (Rapa Nui), in line with the date estimated in ref. 5. However, the only two ancient DNA studies of ancient Rapanui so far did not find evidence for Native American admixture[6,26]. The first study focused on mitochondrial DNA from 12 individuals, whereas the second analysed low-depth (0.0004–0.0041×) whole-genome data from 5 individuals dating before and after European contact. In the latter, downstream population genetic analyses confirmed that the five ancient individuals were Polynesian. However, even though the analysed human remains were post-dating the inferred Native American admixture time, no Native American ancestry was reported in these ancient genomes, casting doubt on the findings based on data from present-day populations.

To infer the genomic history of ancient Rapanui, we have generated whole-genome sequencing data from 15 ancient individuals who were, according to museum records, found in Rapa Nui (Methods section 'Ethics and inclusion'). We analyse these genomes and other publicly available data to: infer the genomic ancestry of these individuals; determine whether the Rapanui experienced a population collapse in the 1600s; and investigate whether Polynesians and Native Americans admixed before European contact.

## Community engagement

Throughout the course of the study, we met with representatives of the Rapanui community on the island, the Comisión de Desarrollo Rapa Nui and the Comisión Asesora de Monumentos Nacionales, where we presented our research goals and ongoing results. Both commissions voted in favour of us continuing with the research. The results of our study were communicated to the community several times, including before first submission (Methods section 'Ethics and inclusion'). We presented the research project in public talks, a short video and radio interviews on the island giving us the opportunity to enquire about the questions that are most relevant to the Rapanui community. These discussions have informed the research topics we investigated in this work.

## Ancient genomes and radiocarbon dates

Following strict museum guidelines, we sampled petrous bone and teeth material from 15 individuals—labelled as Rapanui—from the Muséum national d'Histoire naturelle, France, Pinart (1877) and Métraux (1935) collections using a minimally invasive method[27] (Supplementary Information section 1). This included the retrieval of loose teeth whenever possible (4 individuals) and 60–120 mg of petrous bone powder in other cases (11 individuals). We obtained direct radiocarbon dates from 11 individuals at the Oxford Radiocarbon Accelerator Unit and Bristol

Radiocarbon Accelerator Mass Spectrometer. Results are reported in radiocarbon years before present, corrected for isotopic fractionation using the $^{13}C/^{12}C$ ratio. The results were calibrated to take into account marine reservoir effects derived from the consumption of marine proteins, and broadly range between 1670 and 1950 CE[28–30] (Supplementary Information section 2). Although most of this range postdates European contact in 1722, these skeletal remains must pre-date the date in which they were collected according to the museum archives (that is, 1877 (11 individuals) and 1934–1935 CE (4 individuals)). Thus, it is unlikely that these individuals were born after the 1860s Peruvian slave raids and subsequent epidemics that decimated the island population to an estimated 110 individuals[1,13]. We sequenced the whole genomes from the 15 ancient individuals to an average depth of coverage between 0.4 and 25.6× (Supplementary Information sections 2 and 3 and Supplementary Table 1). For all libraries, we estimated <5% contamination (Supplementary Information section 3 and Supplementary Table 1). To leverage the whole-genome dataset, we produced imputed diploid genotypes with GLIMPSE[31]. We validated the imputed data with benchmarking and downsampling experiments, and by repeating allele sharing and local ancestry inference analyses with subsets of the reference panels (Supplementary Information sections 4 and 5). In what follows, we rely on these imputed genotypes for population genomics analyses and confirm our results using standard pseudohaploid calls when adequate. In addition to the ancient Rapanui genome data, we used present-day Rapanui single nucleotide polymorphism (SNP) array data and ancient genome data from two individuals of previously demonstrated Polynesian genetic ancestry, but uncertain origin location, from the Museu Nacional, Brazil[32]. We refer to the latter as Ancient Polynesians (Supplementary Information section 4.2).

## Ancient genomes from Rapanui ancestors

We explored the broad genetic affinities between the 15 ancient individuals and a panel of 30 worldwide populations (including Fiji, 7 Polynesian islands and present-day Rapanui) genotyped across 755,094 SNP sites[5,33,34] (Supplementary Information section 4.1). We used this panel to maximize the overlap between the reference present-day Polynesian individuals and the present-day Rapanui, who, according to medical and historical records, are unlikely to carry admixture from other Polynesian islands[35]. Multidimensional scaling[36] showed that all individuals were placed among Polynesian genomic diversity across dimensions (Supplementary Fig. 11 and Supplementary Information section 6). We estimated allele frequencies on the basis of the 15 individuals and computed $f_3$-statistics[37,38] of the form $f_3$(Yoruba; $X$, Ancient Individuals), where $X$ represents different present-day populations. We find that the ancient individuals shared the most drift with present-day Rapanui, followed by other Polynesian islanders (Fig. 1a, Extended Data Fig. 1, Supplementary Table 4 and Supplementary Information section 7). Concordantly, the ancient individuals remained most closely related to present-day Rapanui even when we considered a broader set of Oceanian reference individuals[7,39] (Supplementary Fig. 12, Supplementary Table 5 and Supplementary Information section 7). To substantiate these observations using linkage disequilibrium information, we used IBDseq[40] and ancIBD[41] to infer the genomic regions that are identical by descent (IBD) between the newly sequenced individuals and other Remote Oceanians (Fijians and Polynesians; Supplementary Information section 8). We observe that all individuals sequenced in this study share the largest IBD genomic fraction—distributed across the longest IBD segments—with each other, followed by present-day Rapanui, the two Ancient Polynesians from an unknown sampling location and present-day Polynesians in an east–west order and Fijians (Fig. 1b, Supplementary Fig. 20 and Supplementary Table 15). We observe a corresponding pattern for present-day Rapanui with <5% European admixture, who share the largest IBD genomic fraction with the ancient individuals and other present-day Rapanui. The two Ancient Polynesians, whose

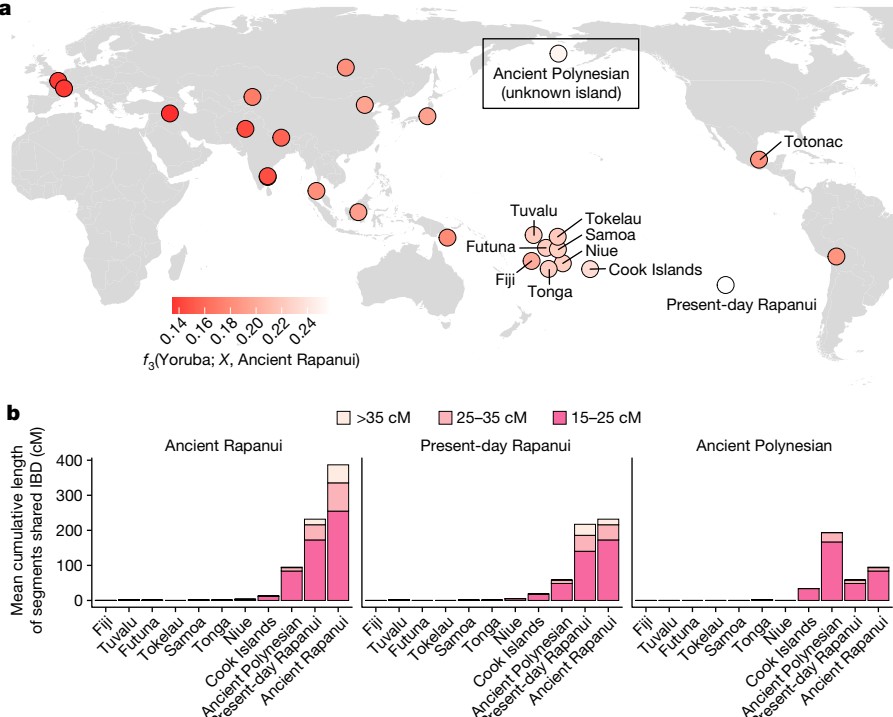

**Fig. 1 | Shared drift and identity by descent between Ancient Rapanui and present-day populations. a**, $f_3$-statistics of the form $f_3$(Yoruba; $X$, Ancient Rapanui), for each population $X$ in a worldwide genotype panel including 755,094 SNP sites. For each ancient individual, we sampled one random allele at each site and pooled these 'pseudohaploid calls' to estimate allele frequencies. In addition, we include data from two previously published Polynesian individuals originally from an unknown island. The lighter a point is, the greater shared drift between a population ($X$) and the ancient individuals sequenced in this study. Raw results are presented in Extended Data Fig. 1 and Supplementary Table 4. Confirmatory $f_3$-statistics for a broader set of Oceanian individuals[7,39] are presented in Supplementary Fig. 12 and Supplementary Table 5. **b**, Average IBD sharing between pairs of individuals from different Polynesian groups as estimated using IBDseq[40]. For all possible pairs of individuals between two groups (for example, the 45 possible pairs between the 15 Ancient Rapanui and the 3 present-day Rapanui) we show the average cumulative length of segments shared IBD, stratified by segment length (colour scheme). We show results for the 15 Ancient Rapanui (left), three representative present-day Rapanui with low European admixture (middle) and two Ancient Polynesians (right) with unknown sampling location. For this analysis, we imputed the ancient individual sequence data to obtain diploid genotypes. Results for each individual (not pooled means) are presented in Supplementary Fig. 20 (including called IBD segments <15 cM). IBDseq estimates stratified by length are presented in Supplementary Table 15.

genotypes were imputed following the same strategy as for individuals sequenced in this study, share the largest IBD genomic fraction mutually. Therefore, we consider these results not to be an artefact related to diploid genotype imputation. Note that IBDseq operates on unphased genotypes and that Remote Oceanians are characterized by historically low effective population sizes, which is expected to give rise to extensive IBD sharing. Furthermore, calling shorter IBD segments accurately is challenging and can vary depending on the calling method (Supplementary Information section 8). Thus, we focus on longer IBD segments (>15 cM) that are not method dependent (Supplementary Information section 8 and Supplementary Figs. 21 and 22). Overall, our results demonstrate that the 15 ancient individuals sequenced in this study are Rapanui ancestors, most closely related to present-day Rapanui, also suggesting that the museum records are correct in this case. The confirmation of the origin of these individuals through genomic analyses will inform repatriation efforts led by the Rapanui repatriation programme (Ka Haka Hoki Mai Te Mana Tupuna), of which M.G.E. is a former member. In what follows, we refer to the 15 Rapanui ancestors we sequenced in this study as Ancient Rapanui to distinguish them from present-day individuals from the island.

## Biological kinship and relatedness

The 15 Ancient Rapanui genomes provide an opportunity to investigate the social dynamics on the island before the 1860s (that is, before the Peruvian slave raids and forced repatriations that had a major impact on the island)[1,13]. Using READ[42] and NGSrelate2[43], we did not find any first- or second-degree relatives among the Ancient Rapanui (Supplementary Figs. 23 and 24 and Supplementary Information section 9) and detected a single pair of third- to fourth-degree relatives on the basis of IBD sharing (highest estimated IBD sharing >15 cM between any two individuals is <1,000 cM; Supplementary Information section 8 and Supplementary Table 15). Moreover, estimated inbreeding coefficients for all pairs of Ancient Rapanui were <0.01 (when using genotype likelihoods; <0.02 when estimating from imputed genotypes), showing that their immediate ancestors were not closely related to each other.

To obtain a general overview of the genomic diversity on the island and potential consanguinity, we identified runs of homozygosity (ROH) in the 15 Ancient Rapanui genomes and a set of reference ancient and present-day worldwide genomes[32,44,45] using hapROH[46] with pseudo-haploid data and PLINK[47] with imputed genotypes excluding transition polymorphisms (Supplementary Information section 10). Compared to other worldwide populations, Ancient Rapanui carry a large proportion of their genomes in ROH (for example, average length for Rapanui = 198 cM versus average length for Eurasia = 25 cM for hapROH; Fig. 2a and Supplementary Table 18; see Supplementary Information section 10 and Supplementary Table 19 for PLINK results). However, most of these ROH are relatively short; on average, 80% of the total ROH are <12 cM and none of the 15 individuals carries a long

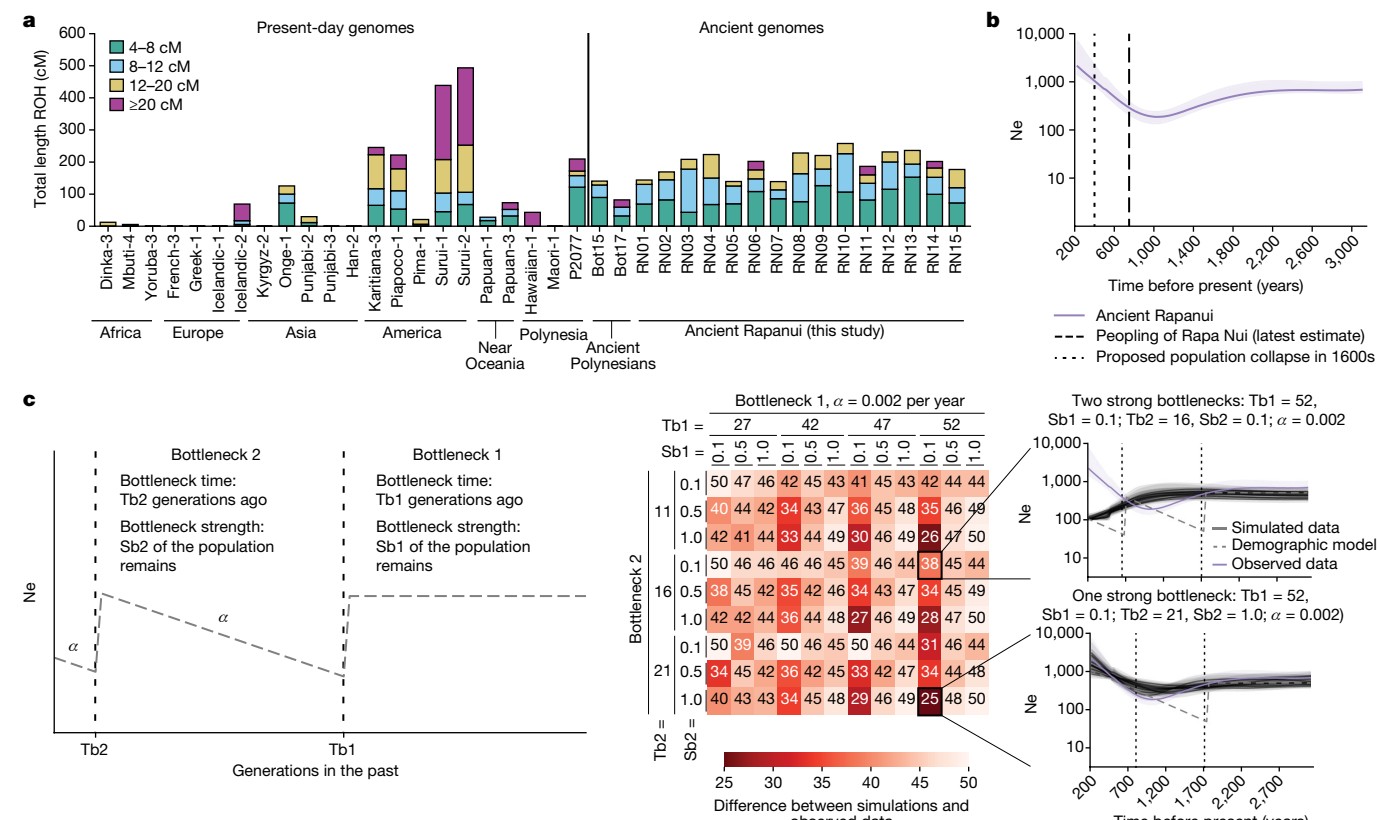

**Fig. 2 | ROH and Rapanui population size estimates through time. a**, Total ROH stratified by length in worldwide present-day and ancient Polynesian genomes as inferred using hapROH (Supplementary Information section 10). **b**, HapNe-LD effective population size estimates for 15 imputed Ancient Rapanui genomes. Assuming the ancient individuals were born in about 1800 CE (Fig. 4) and 29 years per generation, we indicate the latest estimate for the peopling of Rapa Nui (1250 CE), and the 1600s collapse proposed by the ecocide theory. **c**, msprime coalescent-based simulations and HapNe-LD effective population size inference for 15 ancient genomes under a model with two bottlenecks followed by growth. The oldest bottleneck (bottleneck 1) represents the peopling of Rapa Nui. The more recent bottleneck (bottleneck 2) represents the ecocide theory collapse. Bottlenecks are defined by a time (Tb1, Tb2) and a strength (Sb1, Sb2) parameter. Strengths indicate the proportion of the population left after each bottleneck (for example, a bottleneck with strength

0.1 is very strong (10% of the population is left)). We assume the population grows exponentially with rate $\alpha$ after each bottleneck. We compared estimates for the observed and simulated data across a grid of bottleneck and growth parameters (Tb1, Sb1, Tb2, Sb2 and $\alpha$). The heat map (middle) shows a measure of the difference between the effective population size estimates for observed (**b**) and simulated data for a set of representative simulation parameters, across 10 replicates (full range; Supplementary Information section 11). We consider different times for each bottleneck (Tb1, Tb2) and three strengths: 0.1 (strong), 0.5 (intermediate) and 1.0 (non-existent). $\alpha = 0.002$ is the growth rate that minimized the difference between the inferences for observed and simulated data. For reference, we plot the estimates for two simulations (one strong bottleneck and two strong bottlenecks) (right). Black lines correspond to ten independent simulation replicates for each parameter set. Shaded areas show 95% bootstrap confidence intervals.

proportion of their genome in long ROH. By contrast, in individuals with high consanguinity (for example, the Paiter Suruí from southern Amazon[48] and the Punjabi from Pakistan[49]), a large proportion of ROH are long: the two Paiter Suruí individuals carry more than 76% ROH ≥ 12 cM (>335 cM) and Punjabi-2 bears 18 cM out of 30 cM in ROH ≥ 12 cM. We observed that the Ancient Rapanui ROH distribution is similar to that of a present-day Rapanui genome (P2077)[45]. Nevertheless, the latter carries longer ROH than the ancient individuals (38 cM of the genome in ≥20 cM runs). Furthermore, in contrast to that of the Ancient Rapanui, the ROH distribution among six other present-day Rapanui[5] was highly variable (Supplementary Information section 10.4). Although the exact distribution of ROH depends on the inference method (PLINK infers shorter ROH), the estimated inbreeding coefficients (Supplementary Information section 9.1.2) and the relative differences between the ROH length distributions in the Ancient Rapanui and populations with a history of consanguinity support the hypothesis that the Rapanui have a low historical effective population size. Yet unions between kins were seemingly infrequent pre-Peruvian slave raids and could have become more frequent in recent times[50].

## No evidence for 1600s Rapanui collapse

We use the 15 Ancient Rapanui genomes to test the ecocide hypothesis using biological data as they postdate the proposed collapse and yet they are unlikely to be affected by the demographic events that followed the 1860s Peruvian slave raids[1,13]. We reconstructed the Rapanui effective population size over the last 100 generations with HapNe-LD[51], which relies on linkage disequilibrium information and does not require high-confidence IBD calls. In population genetics, directly translating effective population sizes into census sizes is challenging owing to, for example, census inclusion criteria or variance in reproductive rates[52,53]. Therefore, we follow common practice and interpret our subsequent simulation results (see below) qualitatively and in relative terms (that is, by focusing on the estimated effective population size trajectories and the timing of their shifts).

Our inferred HapNe-LD curve is pictured in Fig. 2b and Supplementary Fig. 28. It shows a decreasing population size that reaches its minimum about 28 generations (about 800 years) before the Ancient Rapanui time of birth, followed by a steady increase. These results do not suggest a population collapse in the 1600s, as argued for in the ecocide

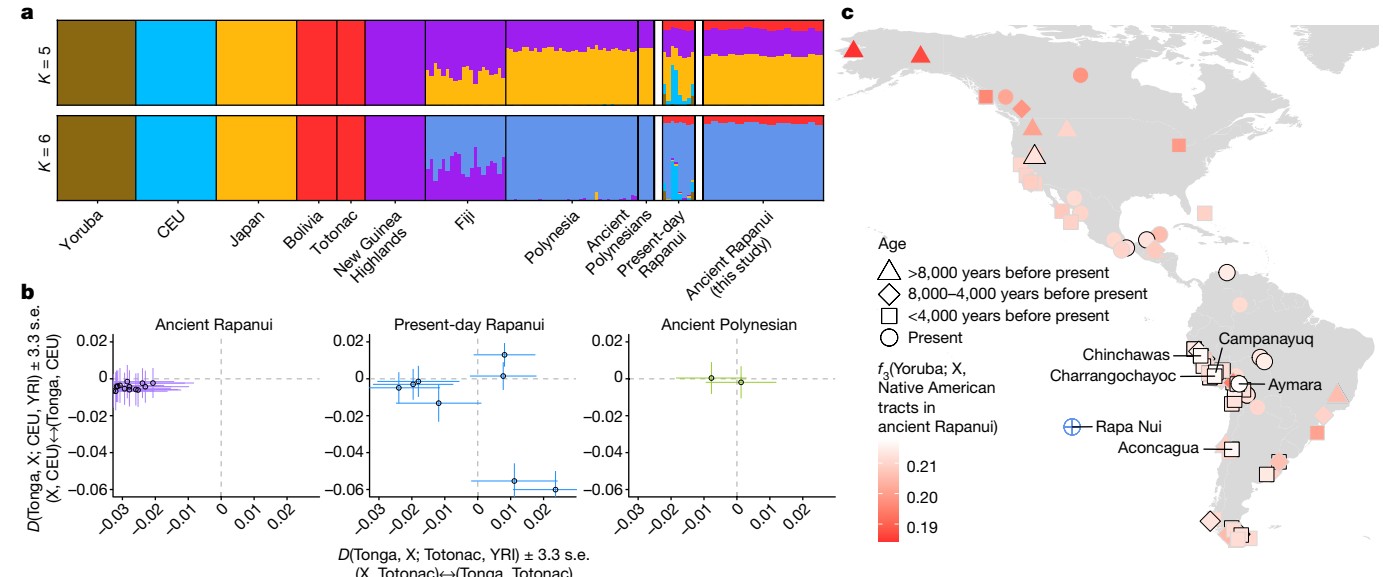

**Fig. 3 | aProportion and source of Native American admixture in Rapanui. a**, ADMIXTURE-estimated proportions assuming $K = 5$ and $K = 6$ ancestry components. For $K = 5$, we considered five reference populations (African (Yoruba), European (CEU), Asian (Japan), Native American (Bolivia and Totonac) and Near Oceanian (New Guinea Highlands)). To minimize the effect of strong drift in Remote Oceanians (Fijians and Polynesians), we ran ADMIXTURE separately for each Remote Oceanian ($n = 78$), including the Ancient Rapanui ($n = 15$), present-day Rapanui ($n = 8$) and the Ancient Polynesians from an unknown island ($n = 2$). For $K = 6$, we included all Fijians and present-day Polynesians (excluding present-day Rapanui) in all runs. Wider bars represent ancient individuals (pseudohaploid calls). **b**, $D$-statistics testing for Native American and European admixture in ancient and present-day Rapanui and the two Ancient Polynesians from an unknown island. Totonac represent Native American ancestry and Utah residents (CEU) represent European ancestry. Points represent $D$-statistics. Error bars represent about 3.3 standard errors

($P \approx 0.001$ in a $Z$ test; 755,094 SNPs in 5-Mb blocks). Under each axis, we indicate the pair of populations with excess allele sharing, depending on the sign of $D$. To maximize the test sensitivity, we considered imputed diploid genotypes for the ancient individuals (Supplementary Information section 5). **c**, $f_3$-statistics showing the genetic affinities between the Native American tracts in Ancient Rapanui and ancient and present-day Native American populations. For each Ancient Rapanui, we masked Polynesian ancestry tracts and pooled Native American ancestry tracts to estimate $f_3$-statistics. Lighter colours represent greater shared drift between a Native American population (X) and the Ancient Rapanui. We label the five Native American populations that lead to the largest $f_3$-statistics. Point shapes represent the age of Native American populations in years before present. Points with a darker outline correspond to point estimates overlapping with the 95% confidence interval of the highest $f_3$-statistic (illustrated in Extended Data Fig. 2).

theory. However, as a given HapNe-LD curve could ultimately correspond to more than one demographic scenario and the Ancient Rapanui carry Native American admixture (see below), we also used extensive population genetics simulations[54] to test whether this result could be consistent with a 1600s ecocide and collapse scenario. We simulated genome data assuming a population history compatible with the most recent findings for the Rapanui population (Supplementary Fig. 29 and Supplementary Table 22) and characterized by five main parameters. Those parameters involve two population bottlenecks determined by their starting date (Tb1 and Tb2), their strength (Sb1 and Sb2) and the population growth rate between bottlenecks ($\alpha$; Fig. 2c and Supplementary Table 23). In our model, bottleneck 1 (Tb1 and Sb1) represents a bottleneck presumably associated with the initial peopling of the island, whereas bottleneck 2 (Tb2 and Sb2) represents the bottleneck associated with a potential population collapse as proposed in refs. 2,3. For each simulated dataset, we inferred the effective population size over the last 100 generations and computed a metric measuring the difference between the HapNe-LD curves for the real and simulated data (Supplementary Figs. 30 and 31 and Supplementary Information section 11). We highlight that the inferred HapNe-LD curves are not intended for estimating absolute effective population sizes. Rather, we use them to discriminate population histories using simulations.

We find that the HapNe-LD curve distance metric is minimized when we consider models with bottleneck 2 = 0% (no size reduction) and with a growth rate $\alpha < 0.3\%$ per year. The observed data are not consistent with models in which bottleneck 2 after initial peopling of the island was strong or intermediate (when only 10–50% of the population remains) or when the growth rate after the first bottleneck ($\alpha$) was high

(>0.3% per year; Fig. 2c). Furthermore, we obtain a $P$ value < $10^{-5}$ using a permutation test in which we split the simulations with Sb2 ≤ 50% (strong or intermediate bottleneck 2) from the ones with Sb2 > 50% (weak bottleneck 2). We also used the total ROH in an individual (SROH) distribution as an alternative summary statistic to the HapNe-LD curves (Supplementary Information section 11.3). We found that the Ancient Rapanui SROH distribution did not match scenarios involving a very strong bottleneck 2 (Sb2 ≤ 0.2; Supplementary Figs. 34 and 35 and Supplementary Table 24). These results do not support a major population collapse on Rapa Nui after its initial peopling and before the 1800s. Rather, they suggest that the island was home to a small population whose effective size steadily increased after initial peopling of the island until the 1860s. See also Supplementary Information section 11.2.2.2 (Supplementary Figs. 32 and 33) for a discussion on the challenges for modelling present-day Rapanui population size trajectories due to their heterogeneous ancestry profiles.

## Rapanui–Native American admixture

We investigated potential contacts between the ancestors of Rapanui and other populations. As an initial exploratory approach, we used ADMIXTURE[55] and pseudohaploid data from the SNP array dataset including African, European, East Asian, and Near and Remote Oceanians (Supplementary Information sections 4.1 and 12). To minimize the effect of strong genetic drift in Remote Oceanians on the admixture proportion estimation, we ran ADMIXTURE ($K = 5$ ancestry components) on separate datasets including all reference individuals and one Remote Oceanian at a time (Fig. 3a and Supplementary Fig. 36). For $K = 6$, we

followed a similar strategy, but we included Fijians and Polynesians as additional reference individuals. These analyses assigned a high proportion of the Ancient Rapanui ancestry (about 90% in average) to a Polynesian-like ancestry component. Furthermore, all ancient and present-day Rapanui carried on average about 10% (6.0–11.4%) and about 8% (6.4–10.3%) of a Native American-like ancestry component, respectively. In contrast, European-like ancestry was only detected in present-day Rapanui, not in Ancient Rapanui. To confirm these results, we computed $D$-statistics[37,38] of the form $D$(Polynesian, Ancient Rapanui; Native Americans or European, Yoruba) (Fig. 3b, Supplementary Figs. 13 and 14 and Supplementary Information section 7). In agreement with ADMIXTURE results, our data showed that Ancient Rapanui share significantly more alleles with Native Americans than other Polynesians do ($|Z| > 3.3$), whereas they remained symmetrically related to Europeans ($|Z| < 3.3$). By contrast, we could not reject the hypothesis that the Ancient Polynesians and present-day Polynesians are symmetric to Native Americans and Europeans ($|Z| < 3.3$). Using an $f_4$-ratio[37,38] of the form $f_4$(Tonga, Yoruba; Ancient Rapanui, Native American)/$f_4$(Tonga, Yoruba; Ancient Polynesians, Native American), we estimated 6.5–12.4% Native American admixture in the Ancient Rapanui (Supplementary Information section 7).

To reconcile these results with the previous study in which no evidence of Native American ancestry was found in ancient Rapanui individuals, we reanalysed the publicly available data from ref. 6. In agreement with ref. 6, we did not obtain statistically significant $D$-statistics suggesting Native American admixture. However, through a downsampling experiment, we show that low depth can explain the absence of statistically significant $D$-statistics (Supplementary Fig. 15 and Supplementary Information section 7). Furthermore, using our ADMIXTURE approach, we detected a Native American-like component in three out of five previously published ancient Rapanui represented by low-depth sequencing data (Supplementary Fig. 37, Supplementary Table 25 and Supplementary Information sections 4.6 and 12).

To explore the origin of the Native American ancestry in the ancient Rapanui, we carried out local ancestry inference using RFMix[56] to identify Polynesian, Native American and European ancestry tracts in their genomes (Supplementary Information section 13). By restricting our analyses to the Native American tracts (mean aggregate genomic fraction = 7.9–12.3%), we computed $f_3$-statistics of the form $f_3$(Yoruba; Native American, Native American tracts in Ancient Rapanui) to assess the most likely source for Native American admixture in the Ancient Rapanui (Supplementary Information section 4.4). This $f_3$-statistic was maximized for ancient and present-day populations from the Central Andean Highlands (Fig. 3c and Supplementary Information section 7), a result that is robust to the exclusion of the Peruvian individuals in the reference panel used for imputation (Supplementary Fig. 16 and Supplementary Information sections 5 and 7). Moreover, we obtained compatible results when we computed $D$-statistics on a dataset for which we did not restrict to local Native American tracts (Supplementary Fig. 16 and Supplementary Information section 7). We note that point estimates for some non-Andean populations overlap with the confidence intervals of the highest $f_3$ (and $D$) values, probably owing to low resolution of the reference dataset (Supplementary Information section 4.4) and short internal branches separating Native American groups[57,58]. However, present-day and ancient Andean populations (top 12 $f_3$-statistics out of 107 computed) consistently yield the highest $f_3$-statistics (Extended Data Fig. 2 and Supplementary Figs. 17–19).

## Pre-European trans-Pacific contact

We used admixture linkage disequilibrium- and local ancestry-based methods to characterize the admixture event between the ancestors of the Ancient Rapanui and Native Americans. We used ALDER[59] and DATES[60] to model the 15 Ancient Rapanui as a mixture of two source populations: Polynesians and a second source representing African, European, East Asian, Papuan or Native American populations (Supplementary Information section 14). We observed that admixture linkage disequilibrium curves had a lower decay rate when we modelled Ancient Rapanui as a Polynesian–Native American mixture (Fig. 4a and Supplementary Figs. 41 and 42). By contrast, the decay of all admixture linkage disequilibrium curves was qualitatively similar when we modelled the two Ancient Polynesian individuals—who lack Native American ancestry—regardless of the sources. Using DATES, we inferred that the Rapanui–Native American admixture occurred 17–32 generations before the average time of birth of the ancient individuals. To narrow down the Native American admixture date, we relied on inferred local ancestry tracts in Ancient Rapanui. We obtained tract length distributions per ancestry and used tracts[61] to estimate an admixture date for the joint 15 Ancient Rapanui individuals. We inferred <1% of European tracts in Ancient Rapanui suggesting these are essentially noise (Fig. 4b,c, Supplementary Fig. 43 and Supplementary Information section 14). Therefore, we estimated the parameters for a model in which the Ancient Rapanui derive from a two-way admixture event between a Polynesian and a Native American source. In this case, we estimate that the admixture occurred 15–17 generations before the average date of birth of the Ancient Rapanui (Supplementary Figs. 44 and 45 and Supplementary Information section 14).

To account for different sampling times for the 15 ancient individuals, we inferred a joint date of admixture for the Rapanui population incorporating genetic admixture dates and radiocarbon ages. For each individual, we first estimated an admixture date using tracts and obtained a 95% confidence interval through 500 bootstrap replicates (Supplementary Fig. 46). Then we used Bayesian modelling to determine admixture time estimates. We used the dates from the Pinart and Métraux collections and modelled them as two separate dated phases with termini ante quem corresponding to the time human remains were collected according to the museum archives and published documents[62–64], accounting for their marine C intake and consequent reservoir offsets. We assumed generation times of 25–30 years and modelled the most likely dates for the Rapanui–Native American admixture event. Each model yielded results that were in statistical agreement (Supplementary Figs. 47–51). Assuming 29 years per generation our favoured model gave a range of 1336–1402 CE (at 68.3% probability) and 1246–1425 CE (95.4%; Fig. 4d and Supplementary Information section 15). These estimates overlap significantly with the most recent estimates for the peopling of Rapa Nui (1150–1280 CE[15]) and strongly suggest that the admixture event did not pre-date the peopling (Supplementary Fig. 52). Moreover, they show that the Polynesian ancestors of Rapanui were in contact with Native Americans significantly before the first appearance of Europeans on the island (364 ± 41 years before 1722).

## Discussion

In this work, we present high-quality ancient Polynesian genomes, which were found to be of Rapanui origin. Although these individuals—radiocarbon dated to the 1800s—postdate the first arrival of Europeans in Rapa Nui, they do not bear European admixture. More importantly, they pre-date the Peruvian slave raids[13] and the devastating smallpox outbreak of the 1860s[1]. As such, this dataset likely constitutes a close representation of Rapanui genomic diversity after the initial peopling of the island and before European contact. Owing to the quality of the data, we were able to address two long-standing contentions on the Rapanui past: a highly publicized theory of a self-inflicted population collapse in Rapa Nui, and the extent of the Polynesian voyages across the Pacific and their early contacts with Native Americans.

Using biological (genomic) data, we found no evidence that the Rapanui underwent a population collapse in the 1600s, originally proposed to be a consequence of deforestation, resource overexploitation and warfare[2–4]. Although trees once covered Rapa Nui, it has

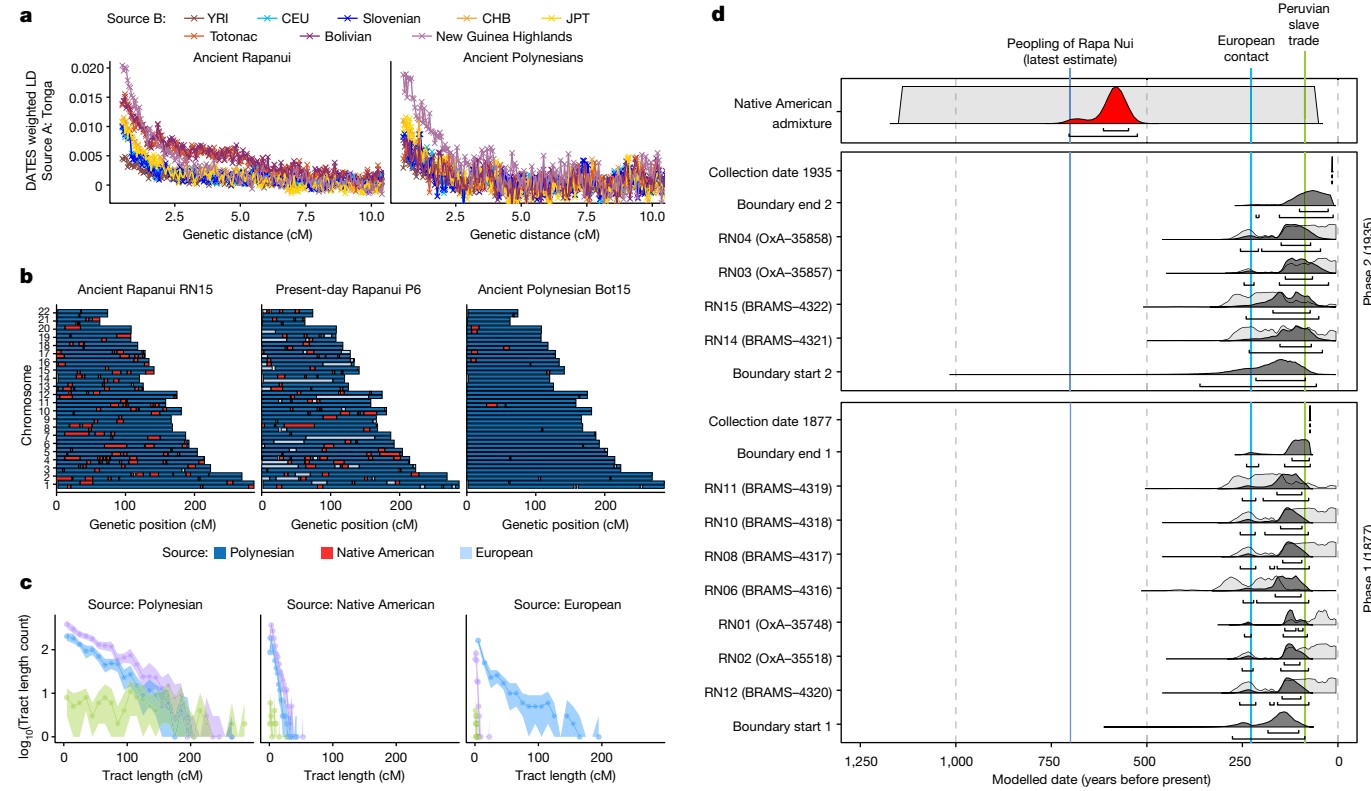

**Fig. 4 | Dating the admixture between the Polynesian ancestors of Rapanui and Native Americans using genetic and radiometric data jointly. a**, We used DATES to estimate admixture linkage disequilibrium (LD) curves for Ancient Rapanui and the Ancient Polynesians from an unknown island. We considered pseudohaploid calls for ancient individuals and eight two-source combinations to model their ancestry. All combinations include individuals from Tonga representing Polynesian ancestry and a representative from Africa (Yoruba (YRI)), Europe (Utah residents (CEU) and Slovenians), East Asia (Han Chinese from Beijing (CHB) and Japanese from Tokyo (JPT)), the Americas (Totonac and Bolivians) or Near Oceania (New Guinea Highlanders). Both test populations show similar qualitative patterns across sources, except for Native Americans, which yield increased admixture linkage disequilibrium in Ancient Rapanui. **b**, Spatial distribution of Polynesian, Native American and European ancestry tracts in representative ancient and present-day individuals. We used RFmix to infer local ancestry tracts in ancient and present-day Rapanui and the Ancient Polynesians from an unknown island. For each representative

individual, we show 22 autosome pairs and paint the inferred ancestry tracts depending on the source. **c**, Ancestry tract length distributions for ancient and present-day Rapanui and the Ancient Polynesians from an unknown island. Colours correspond to the source populations from RFmix (Polynesian, Native American and European). Points and lines correspond to the observed data and the shaded regions correspond to 95% confidence intervals based on 500 non-parametric bootstrap replicates. We use the inferred tract length distributions to estimate the genetic admixture date between Rapanui and Native Americans using tracts. **d**, We used Bayesian modelling to group two phases of reservoir-corrected radiocarbon-dated human remains of known collection dates and used the number of generations since Native American admixture (tracts estimates for each individual (Supplementary Information section 14)) to calculate an overall Native American admixture date for the Ancient Rapanui (red distribution at the top). BRAMS, Bristol Radiocarbon Accelerator Mass Spectrometer.

been proposed that their decline is likely to be a compound consequence of direct human action and the proliferation of rats brought by Polynesian settlers, as observed in other Polynesian islands[65–68]. According to some authors, the Rapanui census population size would have reached as many as 15,000 individuals[2,69]. However, European records from the eighteenth and nineteenth centuries and Métraux's estimates suggest that the Rapanui population was as small as about 3,000 individuals[16,70]. Although such a small population could have been the result of the proposed population collapse, this estimate is also compatible with a population that would have steadily grown after the initial peopling when assuming pre-industrial growth rates[16,70]. These accounts are consistent with our inference that the effective population size monotonously increased after the initial peopling of the island but remained low during the past 1,000 years.

We highlight that our results (and the genomic data they are based on) shed light on the Rapanui demographic history exclusively. Thus, they cannot be used to directly evaluate the ecological impact of human activity on the island. Whereas anthropogenic impact is widespread in Polynesia[8], we specifically reject the hypothesis that such changes in

Rapa Nui resulted in population collapse in the 1600s, before European contact. Instead, our results support that the Rapanui population was resilient despite a changing environment.

Using ADMIXTURE, local ancestry inference and *f*-statistics, we detected about 10% Native American admixture (and no European admixture) in all of the 15 Ancient Rapanui—a genomic diversity pattern not consistent with a post-European contact admixture event. We confidently dated this admixture event to about 1250–1430 CE (that is, well before Columbus arrived in the Americas and the 1722 European arrival in Rapa Nui) using a new method that relies on joint genetic and radiometric data. Although it is well established that the Polynesian Rapanui ancestors arrived on the island during a period of rapid and purposeful exploration voyages across thousands of kilometres of sea[9], the possibility of subsequent round trips to the Americas remains controversial. In particular, the only other ancient whole-genome Rapanui study carried out so far did not find evidence for Rapanui–Native American contact. Through downsampling experiments of our data, we find that the average depth of coverage of that dataset (0.0004–0.0041×) does not yield enough power to carry out the specific statistical tests

using $D$-statistics we applied here (Supplementary Fig. 15 and Supplementary Information section 7). That we infer the Native American component in Ancient Rapanui to be most closely related to Pacific Coast South Americans and not North Americans or populations east of the Andes further substantiates trans-Pacific contacts between Polynesians and Native Americans. We note that the genomic record in Central and South America (particularly in Ecuador and Colombia) and Central America is still sparse. Thus, a better proxy for the Native American populations that interacted with the Polynesian ancestors could be identified as more pre-European colonization genomic data from these regions and other Polynesian islands become available.

Although our findings strongly support pre-European trans-Pacific contacts, it remains challenging to establish the number and directionality of the trips that mediated them using genomic data. Our admixture date estimates are compatible with single-pulse estimates reported for present-day Rapanui[5,7]. They overlap with the most recent estimates for the peopling of the island[15], suggesting that a brief time may separate the admixture event and the peopling of Rapa Nui. Yet, that they postdate the estimates obtained for other present-day Polynesians by about 100–200 years[7]—and are concordant across ancient and present-day Rapanui individuals[5,7]—suggests trans-Pacific contact could have occurred more than once across Polynesian populations. Notably, archaeological evidence and oral history[71,72] attest that Polynesian peoples held the technology and know-how to embark on round trips to the Americas before Europeans reached South America. Thus, we anticipate that additional pre-European ancient genomic data from other Polynesian islands will allow a more nuanced reconstruction of this process.

In addition to providing notable insights into Rapanui population history, the genomes presented here confirm that the 15 ancient individuals that we sampled are Rapanui in origin. When presenting this work to the Rapanui community and their representatives, the need to repatriate these ancestral remains to the island was discussed as a central goal for the community. Our findings should contribute to these future efforts.

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

## Methods

### Community engagement and sampling

Following strict museum guidelines, we sampled 11 petrous bones and 4 teeth originating from 15 individuals stored at the Muséum national d'Histoire naturelle, Musée de l'Homme in Paris after collection in 1872–1877 or 1935. Throughout the course of the study, we met with representatives of the Rapanui community on the island, the Comisión de Desarrollo Rapa Nui (CODEIPA) and the Comisión Asesora de Monumentos Nacionales (CAMN), where we presented our research goals and ongoing results. Both commissions voted in favour of us continuing with the research (see 'Ethics and inclusion' section for additional details).

As for any collection from the nineteenth century, little is known about the circumstances under which the individual bone remains were collected. Some information can, however, be found in the Musée de l'Homme online database (http://colhelper.mnhn.fr/). Out of the 15 individual samples, 11 are reported to have been collected by Alphonse Pinart in 1877 and the remaining by Alfred Métraux in 1935. According to the museum archives, the bone remains come from either of two locations on the island of Rapa Nui: La Pérouse Bay and Vaihou.

### Laboratory procedures

Ancient DNA work was conducted in dedicated clean laboratory facilities at the Globe Institute, University of Copenhagen. We extracted DNA from teeth and petrous bone powder following ref. 73 and refs. 74,75, respectively. We generated double-stranded[73,76] and single-stranded[77] libraries from standard and USER-treated[78] DNA extracts and sequenced the whole genomes from the 15 Ancient Rapanui individuals using Illumina HiSeq and NovaSeq instruments. A detailed description of the laboratory work is included in Supplementary Information section 1.

### Radiocarbon dating

Accelerator mass spectrometry dating was undertaken at the Oxford Radiocarbon Accelerator Unit (University of Oxford, UK) and Bristol Radiocarbon Accelerator Mass Spectrometer (University of Bristol, UK), with measurements undertaken using MiCaDaS accelerators. We used established methods to pretreat bone and extract collagen for accelerator mass spectrometry dating. Stable isotope analyses of carbon and nitrogen were undertaken using elemental analyser–isotopic ratio mass spectrometer methods (Oxford Radiocarbon Accelerator Unit). We used Bayesian modelling with OxCal 4.4[29] software to combine the radiocarbon likelihoods, correcting for marine dietary uptake and local ocean reservoir offsets ($\Delta R$) using the Mix_Curve method (details are provided in Supplementary Information section 2, Supplementary Fig. 1 and Supplementary Table 2).

### Sequencing data processing and mapping

We used AdapterRemoval v1.5.3[79] to trim Illumina adaptor sequences, leading N bases and trailing quality-2 runs and collapse paired-end reads overlapping over ≥11 bases. Collapsed reads were mapped using bwa aln v0.6.2-r126[80] without seeding following ref. 81. We retained mapped reads with a mapping quality greater than 30, removed PCR duplicates using picard MarkDuplicates (http://picard.sourceforge.net), carried out local realignment using GATK[82] and computed the MD tag and extended BAQ for each read using the samtools calmd command. A detailed description of the data processing procedures is presented in Supplementary Information section 3 and mapping statistics are included in Supplementary Table 1.

### Ancient DNA sequencing data authentication

We determined the chromosomal sex of the Ancient Rapanui individuals by examining the depth of coverage ratio on the X chromosome and the autosome (Supplementary Fig. 2). We assessed the authenticity of the sequencing data by inspecting the length distributions and misincorporation patterns of the mapped reads using bamdamage[36] (Supplementary Fig. 3). In addition, we computed type-specific error rates using ANGSD v0.930[83] (Supplementary Figs. 4 and 5). We estimated the proportion of present-day human mitochondrial DNA (mtDNA) contamination using contamMix[84] on the basis of a majority rule consensus for each individual and an alignment of 311 worldwide mtDNA sequences[85]. We estimated the proportion of present-day human nuclear DNA contamination in XY individuals using contaminationX[86] on reads with a mapping quality ≥30, bases with quality ≥20, sites with depth between 3 and 20 and the HapMap CEU allele frequencies[87]. A detailed description of the authentication procedures is included in Supplementary Information section 3.

### mtDNA and Y-chromosome haplotype calling

We called majority rule mtDNA consensus sequences for each individual and used HaploGrep2[88] to assign mtDNA haplogroups. To call Y-chromosome haplogroups, we used pathPhynder[89] and the ISOGG Y-chromosome reference data (last curated 13 May 2021) to place each XY individual into a worldwide Y-chromosome phylogeny.

### Reference data

For multidimensional scaling, $f$-statistics, ADMIXTURE, local ancestry inference and admixture dating, we considered SNP array data from refs. 5,33,34. These include 574 individuals from 30 worldwide populations genotyped using the Affymetrix Human SNP Array 6.0 across 755,094 autosomal SNP sites. Additionally, we considered a broader set of Oceanian reference individuals in refs. 7,39. For analyses requiring control ancient Polynesian whole genomes, we considered whole-genome sequencing data from two Ancient Polynesians from an unknown island[32]. For analyses relying on a whole-genome worldwide panel, we considered the Simons Genome Diversity Project[44] (v4 callset) as distributed in https://sharehost.hms.harvard.edu/genetics/reich_lab/sgdp/variant_set/. To explore the genetic affinities between the Ancient Rapanui, we used a panel including ancient and present-day Native Americans from refs. 58,90–98. To carry out genotype imputation, we used the same reference panel as in refs. 99,100. Finally, we reanalysed low-coverage ancient Rapanui genomes from ref. 6. A detailed description of the reference data is presented in Supplementary Information section 4.

### Genotype imputation

We used GLIMPSE v1.1.1[31] to impute whole-genome sequencing data using the 1000 Genomes v5 phase 3[100,101] as a reference panel following ref. 99. A detailed description of our imputation validation is presented in Supplementary Information section 5 and validation results are presented in Supplementary Figs. 6–10 and Supplementary Table 3.

### Population structure analyses

We used multidimensional scaling to explore the broad continental genetic affinities of the Ancient Rapanui individuals, following ref. 36 (Supplementary Information section 6). As a complementary approach, we explored the genetic ancestry components of the Ancient Rapanui using ADMIXTURE[55]. To minimize the effect of strong drift in Polynesian individuals, we ran ADMIXTURE separately for each Rapanui individual. For every run, we considered 20 independent replicates and present the results for the replicate with the highest likelihood. We present a detailed description of our ADMIXTURE analyses in Supplementary Information section 12 and estimated admixture proportions in Supplementary Table 25.

### $f$-statistics

We used $f$-statistics[37] to explore the drift sharing patterns between the Ancient Rapanui and a set of worldwide reference populations ($f_3$-statistics), and to test specific population history hypotheses ($D$-statistics); for example, whether the Ancient Rapanui bear

Native American ancestry. We computed *f*-statistics using FrAnTK[38] and estimated standard errors for each statistic using a weighted block-jackknife procedure over 5-Mb blocks following ref. 37. All maps were plotted using the ggplot2 v3.3.2[102] R package. A detailed description of these analyses is presented in Supplementary Information section 7 and results for each test are shown in Supplementary Tables 5–14.

### IBD segment sharing

As a complementary approach to finding the population with publicly available genome-wide data that is most closely related to the Ancient Rapanui, we inferred the genomic IBD segments that are shared between pairs of individuals. We used IBDseq[40] and ancIBD[41] on the imputed genotypes and focused on segments ≥15 cM, which were inferred consistently across methods. A detailed description of this analysis is presented in Supplementary Information section 8 and cumulative estimated per-individual IBD sharing is shown in Supplementary Tables 15 and 16.

### Relatedness between Ancient Rapanui individuals

We used READ[42] and ngsRelate2[43] to estimate kinship between the 15 Ancient Polynesians and the two Ancient Polynesians[32]. We applied READ to a pseudohaploid version of the genomes extracted at 755,094 SNP sites (Supplementary Information section 4.1). We used ngsRelate2 to estimate kinship and inbreeding coefficients using both genotype likelihoods and imputed genotypes. We calculated genotype likelihoods at the 1000 Genomes bi-allelic sites and restricted this analysis to transversions to minimize the effect of post-mortem damage. The imputed genotypes had a minor allele frequency (MAF) > 1% and genotype probability (GP) above 0.99. Details and results for these analyses can be found in Supplementary Information section 9.

### ROH

We used hapROH[46] and PLINK v1.9.20200712[47] to detect ROH. We used hapROH to find ROH in haploid mode (e_model = "haploid" and random_allele = True). To do so, we used pileupCaller (https://github.com/stschiff/sequenceTools) following ref. 103 to generate pseudohaploid calls at the 1240K sites with output in eigenstrat format, as required by hapROH. Following ref. 104, we ran PLINK with the following command to detect ROH in imputed ancient genomes and high-coverage present-day genomes: plink --bfile input --homozyg --homozyg-kb 500 --homozyg-gap 100 --homozyg-density 50 --homozyg-snp 50 --homozyg-window-het 1 --homozyg-window-snp 50 --homozyg-window-threshold 0.05 --out output. We restricted this analysis to transversion SNPs and filtered imputed genotypes (MAF > 1% and GP ≥ 0.99). We describe these analyses in detail in Supplementary Information section 10 and present validation results in Supplementary Figs. 25–27 and Supplementary Tables 17–21.

### Effective population size reconstructions

We estimated recent effective population sizes (that is, for the past 100 generations) with HapNe-LD[105] in diploid mode (HapMap recombination rates[87]). We considered imputed genotypes (MAF > 1% and GP ≥ 0.99) for the 15 Ancient Rapanui. In addition, we applied HapNe-LD on SNP data from eight present-day Rapanui individuals[5] genotyped at 755,094 SNP sites (Supplementary Information section 4.1). We used msprime[54] to simulate whole-genome data under various bottleneck scenarios, which we use to contextualize and interpret the HapNe-LD estimates. See Supplementary Information section 11 for a detailed description of these analyses.

### Local ancestry inference

We conducted local ancestry inference for ancient and present-day Rapanui as well as for the Ancient Polynesian individuals using RFmix[56] together with the HapMap genetic map[87]. For all individuals in this analysis, we phased their diploid genotypes (imputed or from SNP array calls) using shapeit2[106] together with the phased 1000 Genomes project Phase 3 reference dataset[100] and the HapMap genetic map[87]. A detailed description of this analysis and validation experiments is presented in Supplementary Information section 13 and Supplementary Figs. 38–40.

### Admixture dating using genetic data

We followed three approaches to date the Rapanui–Native American admixture event using genetic data. We used ALDER[59] and DATES[60] to build admixture linkage disequilibrium curves for the Ancient Rapanui by modelling their ancestry as a mixture of a Polynesian source (represented by Tongans) and a set of worldwide representative populations (including Native Americans). To complement this approach, we ran ALDER by using the Ancient Rapanui itself, as one of the source populations. Finally, we built Polynesian, Native American and European tract length distributions with 50 length bins for ancient and present-day Rapanui and used tracts[61] to estimate admixture dates for two- and three-source models. For every tracts run, we initialized all parameters from ten random starting points and kept the maximum-likelihood estimates in each case. To build 95% confidence intervals, we obtained maximum-likelihood estimates for 500 non-parametric bootstrap replicates. Additionally, we obtained per-individual tracts estimates and confidence intervals. These estimates and confidence intervals were used in the Bayesian modelling—together with radiocarbon data—to estimate the admixture date (see below). A detailed description of these analyses is presented in Supplementary Information section 14.

### Admixture dating using genetic and radiocarbon data jointly

We used a new Bayesian modelling approach to estimate admixture times using genetic admixture dates and radiocarbon dates for each individual jointly. Using OxCal[29], we modelled two separate dated phases with termini ante quem corresponding to the time human remains were collected, accounting for their marine C intake and consequent reservoir offsets. For each dated individual, we used the mean number of generations elapsed since introgression based on the per-individual tracts estimates as a constraint to calculate an 'introgression' date in the Bayesian model. Model specifications are presented in Supplementary Information section 15.

### Ethics and inclusion

We held meetings with the representatives of the Rapanui community on the island, CODEIPA and CAMN. During these meetings we presented our research goals and preliminary results and enquired and discussed about research questions of particular interest to the Rapanui community (see below). Both commissions voted in favour of us continuing with the research. Summaries of these meetings and the discussions that followed can be found in the meeting minutes: 1.10.22, 4.11.22 (CODEIPA); N17, N18 (CAMN). Additionally, we presented our results to the Rapanui community in a public lecture at the Rapa Nui Museum, where we discussed our findings and interpretations with the public. We conducted these activities with assistance from the staff of the Rapa Nui Museum.

In the current manuscript we researched some of the topics suggested by the community including: kinship relationships between the 15 Ancient Rapanui individuals and how these relate to ancestral social structure; the genetic relationship between Ancient Rapanui, present-day Rapanui and other Polynesian islanders; the potential to explore the population size at the time of the peopling of the island; the difference between the time of the initial peopling of the island and the contact with Native Americans; the relationship between a potential 'cultural' collapse in the 1600s and the Rapanui population size; the source population for Native American ancestry in the Rapanui; the date of the Rapanui–Native American contact; the relationship between our results and previous studies (for example, ref. 26).

A draft of this manuscript was shared with representatives from the Rapa Nui museum before submission. Following peer review, the published manuscript will be stored at the Rapa Nui Museum Library and results will be summarized in an outreach poster that will be displayed at the Rapa Nui Museum. Finally, we will submit a report with a summary of the published results to both commissions.

When presenting this work to the Rapanui community and their representatives, the need to repatriate these ancestral remains to the island became a central goal for the community. We believe that our findings establishing the Rapanui genomic ancestry of the 15 ancestors will inform these repatriation efforts.

Throughout the text, tables and figures, given our population genetics results showing that the 15 ancient individuals sequenced in this study are Rapanui in origin, we refer to them as Ancient Rapanui and identify the individuals as RN01–RN15.

### Reporting summary

Further information on research design is available in the Nature Portfolio Reporting Summary linked to this article.

### Data availability

Following consultation with CAMN, the Ancient Rapanui sequencing data will not be deposited in a public repository and will be made available upon request to J.V.M.-M. and A.-S.M. Access requests will be managed jointly with CAMN representatives. Ancient Rapanui sequencing data are available for population history research and not for public posting, medical research or commercial purposes. Publicly available data were obtained from the following sources—access granted by authors[5,7,32–34,39,45,90,107]; publicly available data in the European Nucleotide Archive: PRJEB22217 (ref. 6), PRJEB20398 (ref. 91), PRJEB29074 (ref. 58), PRJEB24629 (ref. 92), PRJEB28961 (ref. 93), PRJEB37446 (ref. 94), PRJEB39010 (ref. 95), and PRJEB31736 and ERP114329 (refs. 99,100); publicly available data in the Short Read Archive: SRA047577 (ref. 108), SRX381032 (ref. 96), SRS937952 (ref. 97) and PRJEB25445 (ref. 98); publicly available data in custom repositories: http://cdna.eva.mpg.de/neandertal/altai/ModernHumans/bam/ (ref. 109), https://sharehost.hms.harvard.edu/genetics/reich_lab/sgdp/variant_set/ (ref. 44). Figures were plotted using R v4.2.2 (packages ggplot2 v3.3.2, ggridges v0.5.3, ggrepel v0.8.0 and RColorBrewer v1.1-3) and python v3 (packages matplotlib v3.5.3 and pandas v1.0.1).

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

**Acknowledgements** We thank the Rapanui representative commissions CODEIPA and CAMN for their support to this research, J. Ramírez, P. Valenzuela and M. Atam López for guidance and assistance with community consultation, J. Ramos-Madrigal, T. Pinotti, A. Garrido Marques, L. Excoffier, D. J. Meltzer, C. Bronk Ramsey and E. Willerslev for discussions, V. de Bakker for assistance with texts in Dutch, and the GeoGenetics Sequencing Core facility, Globe Institute, University of Copenhagen for assistance with data generation. Research was financed by a European Research Council (grant agreement no. 679330) and a Swiss National Science Foundation (PP00P3_176977) grant to A.-S.M. J.V.M.-M. is supported by the European Research Council (grant agreement no. 101078151) and VILLUM FONDEN (VIL53099). B.s.d.M. was additionally supported by a Swiss National Science Foundation (PP00P3_176977) grant to O.D. H.S. is supported by the European Research Council (grant agreement no. 101045643).

**Author contributions** The project was conceived by J.V.M.-M., B.s.d.M. and A.-S.M with input from M.G.E., CAMN Rapa Nui, and other members from the Rapanui community and F.T.H. J.V.M.-M., O.D. and A.-S.M. supervised the population genomic analyses. M.E.A. and H.S. supervised the molecular laboratory work with input from J.V.M.-M. and A.-S.M. V.L., E.H., M.F., M.E.A., H.S. and A.-S.M. conducted sampling. S.K., J.S. and M.I.-O. processed ancient DNA. J.V.M.-M., B.s.d.M. and T.H. assembled datasets. J.V.M.-M. and B.s.d.M. analysed the genetic data. T.H. conceptualized and conducted the radiocarbon analyses and the introgression dating. T.H., M.G.E., V.L., E.H., F.T.H. and M.F., provided anthropological and archaeological contextualization. J.V.M.-M., B.s.d.M., S.K., A.S.-M., F.T.H. and M.G.E. presented the work and consulted with representatives of the Rapanui community. J.V.M.-M., B.s.d.M. and A.-S.M wrote the manuscript with input from T.H., M.G.E., V.L., E.H., M.F., M.E.A., H.S. and all other authors.

**Competing interests** O.D. is a current employee of the Regeneron Genetics Center, which is a subsidiary of Regeneron Pharmaceuticals. The remaining authors declare no competing interests.

**Additional information**
**Correspondence and requests for materials** should be addressed to J. Víctor Moreno-Mayar, Olivier Delaneau or Anna-Sapfo Malaspinas.

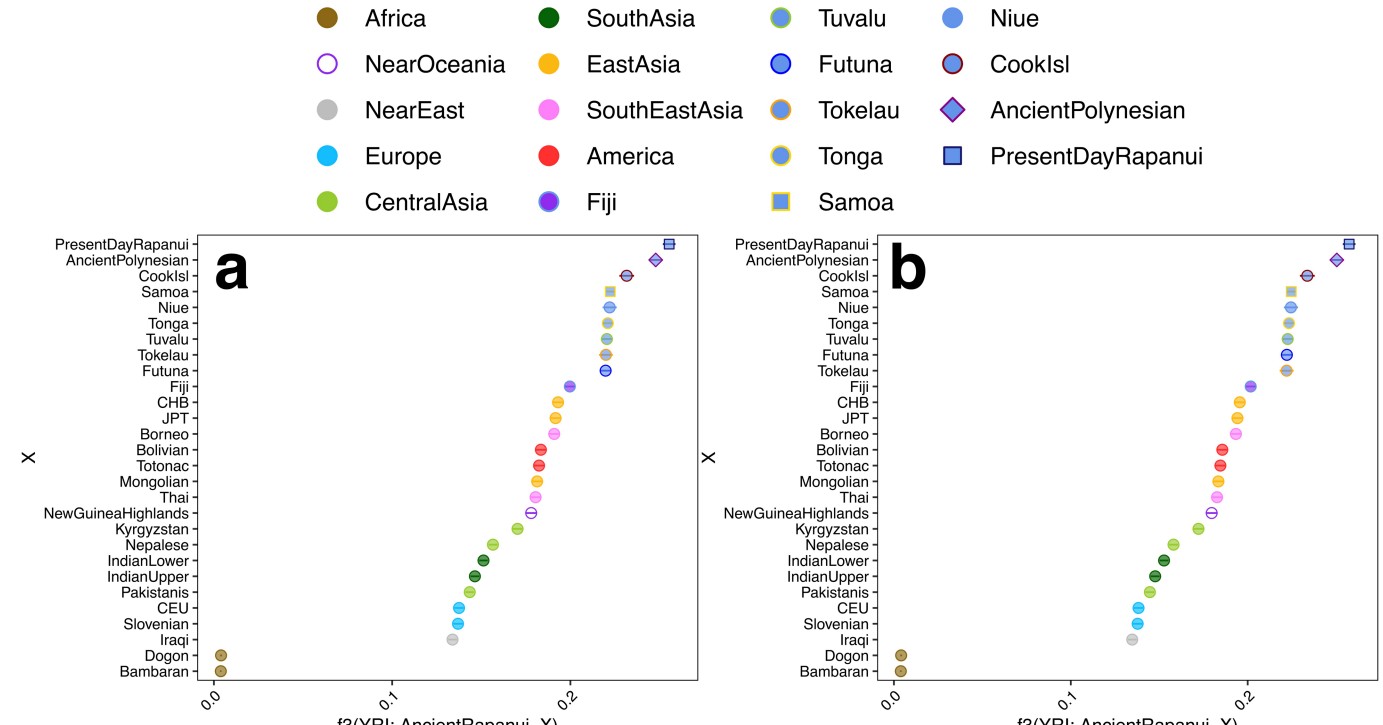

**Extended Data Fig. 1 | $f_3$-statistics measuring shared drift between 'Ancient Rapanui' individuals and other present-day and ancient populations.** We estimated the 'Ancient Rapanui' allele frequencies for the sites included in the SNP array dataset (Supplementary Information section 4) and computed $f_3$-statistics of the form $f_3(Yoruba; X, Ancient Rapanui)$. For each population $X$, the point represents the point estimate for $f_3$, and error bars correspond to the 95% confidence intervals. Non-Polynesian populations are coloured according to their broad continental ancestry and Polynesian populations are coloured according to their island of origin. **a.** $f_3$-statistics results for pseudo-haploid calls. **b.** $f_3$-statistics results for imputed diploid genotypes. Panel a. corresponds to the results shown in Fig. 1a. Raw results for these $f_3$-statistics are presented in Supplementary Table 4.

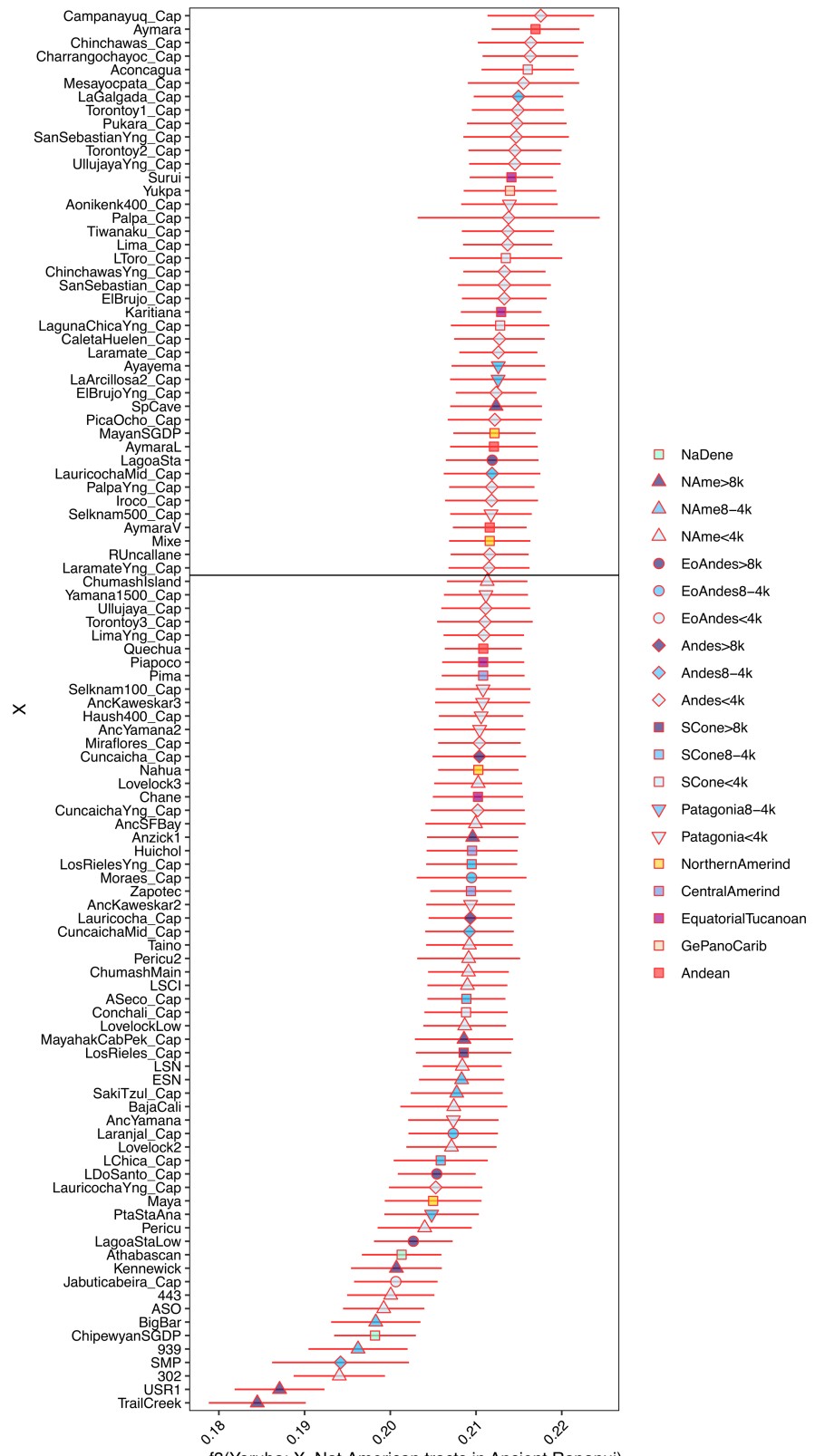

**Extended Data Fig. 2 | $f_3$-statistics showing the genetic affinities between the Native American tracts in 'Ancient Rapanui' and ancient and present-day Native American populations.** This figure shows 95% confidence intervals (727,455 SNPs in 5 Mb blocks) for the $f_3$-statistics reported in Fig. 3c. The horizontal line indicates point estimates overlapping with the 95% confidence interval of the highest $f_3$-statistic. The top 12 $f_3$ values correspond to ancient and present-day Andean populations (Andes8-4k, Andes<4k, present-day Andean and Aconcagua[58]). For present-day Native American populations (those that do not include an age range) symbols and colours correspond to their language family following[57], e.g., NaDene, and for ancient Native American populations, they correspond to their sampling location and age. Abbreviations for ancient populations: 'NAme': North America, 'EoAndes': East of Andes, 'SCone': Southern Cone, >8k: ancient data pre-dating 8,000 years ago, 8–4k: ancient data ranging between 8,000 and 4,000 years ago, <4k: ancient data post-dating 4,000 years ago. Raw results are reported in Supplementary Table 10.

# Reporting Summary

## Statistics

For all statistical analyses, confirm that the following items are present in the figure legend, table legend, main text, or Methods section.

| n/a | Confirmed | |
|---|---|---|
| ☐ | ☒ | The exact sample size (*n*) for each experimental group/condition, given as a discrete number and unit of measurement |
| ☐ | ☒ | A statement on whether measurements were taken from distinct samples or whether the same sample was measured repeatedly |
| ☐ | ☒ | The statistical test(s) used AND whether they are one- or two-sided *Only common tests should be described solely by name; describe more complex techniques in the Methods section.* |
| ☐ | ☒ | A description of all covariates tested |
| ☐ | ☒ | A description of any assumptions or corrections, such as tests of normality and adjustment for multiple comparisons |
| ☐ | ☒ | A full description of the statistical parameters including central tendency (e.g. means) or other basic estimates (e.g. regression coefficient) AND variation (e.g. standard deviation) or associated estimates of uncertainty (e.g. confidence intervals) |
| ☐ | ☒ | For null hypothesis testing, the test statistic (e.g. *F*, *t*, *r*) with confidence intervals, effect sizes, degrees of freedom and *P* value noted *Give P values as exact values whenever suitable.* |
| ☐ | ☒ | For Bayesian analysis, information on the choice of priors and Markov chain Monte Carlo settings |
| ☐ | ☒ | For hierarchical and complex designs, identification of the appropriate level for tests and full reporting of outcomes |
| ☐ | ☒ | Estimates of effect sizes (e.g. Cohen's *d*, Pearson's *r*), indicating how they were calculated |

*Our web collection on statistics for biologists contains articles on many of the points above.*

## Software and code

Policy information about availability of computer code

| Data collection | DNA libraries were sequenced in Illumina HiSeq 4000 and NovaSeq instruments. Radiocarbon measurements were undertaken at the ORAU (Oxford Radiocarbon Accelerator Unit) and BRAMS (Bristol Radiocarbon AMS) facilities. |
|---|---|
| Data analysis | Here we list the software that was used in this work. References for each software are included in the main text and supplementary information.<br><br>- CASAVA v1.8.2. (Illumina) was used to produce base calls.<br>- Illumina adapter sequences were trimmed using AdapterRemoval v1.5.3.<br>- Filtered reads were mapped using bwa aln v0.6.2-r126 and post-processed using picard-tools v2.7.1-SNAPSHOT, GATK v3.8-1-0-gf15c1c3ef and samtools v1.12.<br>- Deamination patterns were explored using bamdamage (distributed together with bammds).<br>- Type-specific error rates were estimated using ANGSD v0.930. mtDNA contamination estimates were obtained using contamMix v1.0-5 and X-chromosome contamination estimates were obtained using contaminationX commit 60e2b58.<br>- mtDNA haplogroups were called using HaploGrep2.<br>- Y-chromosome haplogroups were called using pathPhynder v1.a.<br>- Diploid genotypes were directly called using bcftools v1.12.<br>- Imputation was conducted using GLIMPSE v1.1.1.<br>- The imputation reference panel was lifted over with Picard liftoverVCF v1.18.11.<br>- SNP data was processed using plink v1.9.20200712.<br>- MDS transformations were carried out using R v4.2.2, following bammds.<br>- f-statistics were computed using FrAnTK commit 6d61ab8. |

- All maps were plotted using the ggplot2 v3.3.2 R package.
- IBD segements were called using IBDseq v04Sep15.e78 and ancIBDv0.5.
- Relatedness between individuals was estimated using READ commit f541d55 and ngsRelate v2 commitc327f744d76a17e1d17ecac88fd92b3ac82a0b07.
- Runs of homozygosity were called using plink v1.9.20200712 and hapROH v0.51a0.
- Effective population sizes were estimated using HapNe-LD v1.20220802.
- Coalescent simulations were conducted using msprime v1.2.0.
- Model-based clustering was run using ADMIXTURE v1.3.0.
- Local ancestry inference was performed using RFmix on genotypes phased with shapeit2 v2.904.3.10.0-693.11.6.el7.x86_64.
- Admixture dating was conducted using ALDER v1.03, DATES v753 and tracts v1.
- Radiocarbon date calibration and admixture dating bayesian modelling were performed using OxCal 4.4.
- Figures were plotted using R v4.2.2 (packages ggplot2 v3.3.2, ggridges v0.5.3, ggrepel v0.8.0 and RColorBrewer v1.1-3) and python v3 (packages matplotlib v3.5.3 and pandas v1.0.1).

For manuscripts utilizing custom algorithms or software that are central to the research but not yet described in published literature, software must be made available to editors and reviewers. We strongly encourage code deposition in a community repository (e.g. GitHub). See the Nature Portfolio guidelines for submitting code & software for further information.

# Data

Policy information about availability of data

All manuscripts must include a data availability statement. This statement should provide the following information, where applicable:
- Accession codes, unique identifiers, or web links for publicly available datasets
- A description of any restrictions on data availability
- For clinical datasets or third party data, please ensure that the statement adheres to our policy

Following consultation with the Comisión Asesora de Monumentos Nacionales (CAMN), the 'Ancient Rapanui' sequencing data will be made available upon request to the corresponding authors. Access requests will be managed jointly with CAMN representatives. 'Ancient Rapanui' sequencing data is not available for public posting, medical research or commercial purposes.

Publicly available data was obtained from the following sources:

- Access granted by authors: (Xing et al. 2010; Wollstein et al. 2010; Raghavan et al. 2014; Malaspinas et al. 2014; Moreno-Mayar et al. 2014; Raghavan et al. 2015; Malaspinas et al. 2016; Ioannidis et al. 2020; Ioannidis et al. 2021)
- Publicly available data in the European Nucleotide Archive: PRJEB22217(Fehren-Schmitz et al. 2017), PRJEB20398(Moreno-Mayar, Potter, et al. 2018), PRJEB29074(Moreno-Mayar, Vinner, et al. 2018), PRJEB24629(de la Fuente et al. 2018), PRJEB28961(Posth et al. 2018), PRJEB37446(Nakatsuka, Lazaridis, et al. 2020), PRJEB39010(Nakatsuka, Luisi, et al. 2020), PRJEB31736/ERP114329(Auton et al. 2015; Sousa Da Mota et al. 2023),
- Publicly available data in the Short Read Archive: SRA047577(Meyer et al. 2012), SRX381032(Rasmussen et al. 2014), SRS937952(Rasmussen et al. 2015), PRJEB25445(Scheib et al. 2018)
- Publicly available data in custom repositories: (http://cdna.eva.mpg.de/neandertal/altai/ModernHumans/bam/)(Prüfer et al. 2013), (https://sharehost.hms.harvard.edu/genetics/reich_lab/sgdp/variant_set/)(Mallick et al. 2016).

# Research involving human participants, their data, or biological material

Policy information about studies with human participants or human data. See also policy information about sex, gender (identity/presentation), and sexual orientation and race, ethnicity and racism.

| | |
|---|---|
| Reporting on sex and gender | We determined the chromosomal sex of the ancient individuals by comparing the depth of coverage in the autosomes and the X-chromosome. We highlight that our findings on chromosomal sex do not provide any information on the gender identity of the individuals we sequenced in this study. We do not carry any analyses where we stratify the data by chromosomal sex. |
| Reporting on race, ethnicity, or other socially relevant groupings | We refer to the 15 ancient individuals that were sequenced in this study as 'Ancient Rapanui'. This is based on the museum records indicating these individuals were sampled in Rapa Nui and our results showing the ancient individuals are most closely related to present-day Rapanui. For the publicly available reference data, we use labels that describe their broad continental genomic ancestry as established in their original publications. |
| Population characteristics | NA |
| Recruitment | Following strict museum guidelines, we sampled petrous bone and teeth material from 15 individuals—labelled as Rapanui—from the Musée National d'Histoire Naturelle, France, Pinart (1877) and Métraux (1935) collections using a minimally-invasive method. This included the retrieval of loose teeth whenever the remains were sufficiently well preserved and enough teeth were present (four individuals) and up to 120mg (60-120mg) of petrous bone powder in other cases (11 individuals). |
| Ethics oversight | Throughout the course of the study, we met with representatives of the Rapanui community on the island, the Comisión de Desarrollo Rapa Nui (CODEIPA) and the Comisión Asesora de Monumentos Nacionales (CAMN), where we presented our research goals and ongoing results. Both commissions voted in favour of us continuing with the research and the results of our study have been communicated to the community prior to the first submission of our manuscript. Furthermore, we presented the research project in public talks, a short video and radio interviews on the island giving us the opportunity to inquire about the questions that are most relevant to the Rapanui community. These discussions have informed the research |

topics we investigated and ultimately the results presented in this work. See a detailed description in the Ethics and Inclusion Section.

Note that full information on the approval of the study protocol must also be provided in the manuscript.

# Field-specific reporting

Please select the one below that is the best fit for your research. If you are not sure, read the appropriate sections before making your selection.

☐ Life sciences ☐ Behavioural & social sciences ☒ Ecological, evolutionary & environmental sciences

For a reference copy of the document with all sections, see nature.com/documents/nr-reporting-summary-flat.pdf

# Ecological, evolutionary & environmental sciences study design

All studies must disclose on these points even when the disclosure is negative.

| | |
|---|---|
| Study description | We radiocarbon dated and sequenced the whole-genomes from 15 ancient individuals from the Polynesian island of Rapa Nui. We use the genomic data to study the historical social structure of the Rapanui, their genetic affinities to other worldwide populations and their demographic history including their historical effective population size and admixture history. |
| Research sample | We sampled 15 individuals—labelled as Rapanui—from the Musée National d'Histoire Naturelle, France, Pinart (1877) and Métraux (1935) collections using a minimally-invasive method. |
| Sampling strategy | Sample sizes were not pre-determined. For the 15 individuals, we retrieved loose teeth whenever the remains were sufficiently well preserved and enough teeth were present (four individuals) and up to 120mg (60-120mg) of petrous bone powder in other cases (11 individuals). |
| Data collection | A detailed description of the radiocarbon and genome data collection for the 15 'Ancient Rapanui' individuals is included in Supplementary Sections 1, 2 and 3. Data collection for reference datasets, their provenance and the analyses they were included in is detailed in Supplementary Section 5. |
| Timing and spatial scale | We sampled ancestral remains that were included in the Musée National d'Histoire Naturelle, France, Pinart and Métraux collections in 1877 and 1935, respectively. |
| Data exclusions | We did not exclude any of the 15 'Ancient Rapanui' individuals from any analysis. |
| Reproducibility | For each individual, we generated high-quality whole-genome sequencing data to improve statistical power. Where applicable, we conducted population genetics analyses using different call types (pseudohaploid, called diploid and imputed diploid genotypes) and filtering loci where post-mortem damage has a stronger effect (transition polymorphisms). Furthermore, we replicated different analyses using different reference datasets. We find qualitatively concordant results across the different datasets. |
| Randomization | We did not carry out any randomisation procedure. Our study is focused on understanding the genomic diversity of the Rapanui through time. Thus, we carry out analyses per-individual or by pooling all individuals into a single population. |
| Blinding | Our study did not require any blinding procedure. Our study do not measure different group responses to, e.g., different treatments. We analyse genomic data from each individual separately or by pooling all individuals into a single 'population'. |

Did the study involve field work? ☐ Yes ☒ No

# Reporting for specific materials, systems and methods

We require information from authors about some types of materials, experimental systems and methods used in many studies. Here, indicate whether each material, system or method listed is relevant to your study. If you are not sure if a list item applies to your research, read the appropriate section before selecting a response.

## Materials & experimental systems

| n/a | Involved in the study |
|-----|------------------------|
| ☒ | ☐ Antibodies |
| ☒ | ☐ Eukaryotic cell lines |
| ☐ | ☒ Palaeontology and archaeology |
| ☒ | ☐ Animals and other organisms |
| ☒ | ☐ Clinical data |
| ☒ | ☐ Dual use research of concern |
| ☒ | ☐ Plants |

## Methods

| n/a | Involved in the study |
|-----|------------------------|
| ☒ | ☐ ChIP-seq |
| ☒ | ☐ Flow cytometry |
| ☒ | ☐ MRI-based neuroimaging |

## Palaeontology and Archaeology

**Specimen provenance**

The 15 'Ancient Rapanui' individuals were sampled following strict museum guidelines at the Musée National d'Histoire Naturelle, France. We met with representatives of the Rapanui community on the island, the Comisión de Desarrollo Rapa Nui (CODEIPA) and the Comisión Asesora de Monumentos Nacionales (CAMN), where we presented our research goals and ongoing results. Both commissions voted in favour of us continuing with the research and the results of our study have been communicated to the community prior to the first submission of our manuscript.

**Specimen deposition**

The 15 'Ancient Rapanui' individuals we sequence in this study are being kept at the Musée National d'Histoire Naturelle, Musée de l'Homme in Paris. The confirmation of the origin of these individuals through genomic analyses will inform repatriation efforts led by the Rapanui repatriation program (Ka Haka Hoki Mai Te Mana Tupuna).

**Dating methods**

We obtained direct radiocarbon dates from 11 individuals at the ORAU (Oxford Radiocarbon Accelerator Unit) and BRAMS (Bristol Radiocarbon AMS). All lab codes and dates are reported in Table S2. Calibration methods are detailed in Supplementary Sections 3 and 16.

☒ Tick this box to confirm that the raw and calibrated dates are available in the paper or in Supplementary Information.

**Ethics oversight**

Throughout the course of the study, we met with representatives of the Rapanui community on the island, the Comisión de Desarrollo Rapa Nui (CODEIPA) and the Comisión Asesora de Monumentos Nacionales (CAMN), where we presented our research goals and ongoing results. Both commissions voted in favour of us continuing with the research and the results of our study have been communicated to the community prior to the first submission of our manuscript. Furthermore, we presented the research project in public talks, a short video and radio interviews on the island giving us the opportunity to inquire about the questions that are most relevant to the Rapanui community. These discussions have informed the research topics we investigated and ultimately the results presented in this work. See a detailed description in Supplementary Section 1.

Note that full information on the approval of the study protocol must also be provided in the manuscript.

