## [Peer Review File · Nature]

Manuscript Title: Ancient Rapanui genomes reveal resilience and pre-European contact with the Americas

Reviewer Comments & Author Rebuttals

Reviewer Reports on the Initial Version:

Referee #1 (Remarks to the Author):

This is a well-written paper which presents data and opinions on two issues that are of major importance in the prehistory of Rapanui, or Easter Island. It is entirely based on genetic data. Since I am not a geneticist I am not able to comment on the technical aspects of this paper, but I detect that the arguments are well presented and the paper achieves a high level of competence in the analysis of ancient DNA and the statistical techniques involved in such analyses.

I did have a couple of issues of uncertainty.

One is the possibility that the radiocarbon dates, as plotted in figure S1 in the supplementary information, do not seem to exclude completely the possibility that the 15 skulls analysed postdate the arrival of Roggeveen in 1722, and this possibility leads on to another possibility, that the Native American part of the genome could also be post-contact, given that many other ships visited the island later in the 18th century. . However, the modelling presented by the authors is fairly suggestive that the Native American contact is genuinely pre-Roggeveen. But I would note that both the argument for Native American admixture and also the argument that there was no population collapse in the 17th century are based on statistical simulations rather than direct observations of skeletal or archaeological materials from the times in question. This does not mean that the authors conclusions are wrong, but it seems to me that ancient DNA from some petrous bones dating unambiguously to before 1500 CE will need to be analysed before there is absolute certainty about the American contact issue.

A couple of small points:

Line 637: ref is wrong. Should be Kirch, Patrick V.

Line 73: double-hulled canoes? I would suggest "...possibly using double-hulled canoes", given that there is no proof that their boats actually were double-hulled. Large outrigger canoes with platforms for passengers and cargo might also have been used, as in Micronesia, although this is not an essential issue for discussion in this paper. No one will ever know for certain what their boats looked like!

The Polynesian skulls from the Botocudo source near Rio sound to me like a case of mis-labeling. I cannot prove it, but the idea of Polynesians migrating all the way across South America to the vicinity of Rio de Janeiro seems rather unbelievable. Perhaps a member of the museum staff in Rio many decades ago muddled up some skulls before they were properly labelled. This does not affect their significance as skulls of Polynesian ancestry, but I would state on lines 193-194 ".. two skulls of previously-demonstrated Polynesian genetic ancestry, but uncertain origin location, stored in the Museu Nacional in Rio de Janeiro, Brazil. We refer to these as 'Ancient Polynesians',"

Referee #2 (Remarks to the Author):

This article presents the first high-quality ancient genomes from Rapa Nui, the most remote inhabited island in the Polynesian triangle. The data and paper are highly relevant to address important questions of human history in the Pacific. The authors did overall a great job in generating the data, its analyses, and the article is generally also very well written.

I want to also applaud the authors on their description of the community engagement in the main text, which puts this important part of this type of research right into the focus where it belongs.

My main criticism concerns the lack of tabular data attached to the paper, which also prevented me from reviewing this work to the level that I would like to. I think it should be self-evident that all quantitative figures in the main article should have their underlying data accessible as Supplementary Datasets. While the Supplement lists many needed details of the analyses, the actual underlying data, most importantly the IBD sharing tables, but also results of F-statistics etc. are not available in table-form, as far as I could see.

While I have no a priori reason to doubt the authors' expertise in analysing the data, I would like to be fully convinced of some of the patterns the authors report, which is only possible with a direct look at the data.

In particular:

- Data underlying Fig 1a: I would like to see the F3-statistics in table form including error bars
- Data underlying Fig 1b: I would like to see at the minimum for each individual a table of length-stratified cumulative IBD (e.g. stratified using bins of certain length windows) with each other reference individual/group. Even better would be a table listing all shared IBD segments, perhaps above a certain length cutoff. That might then be too large a table for Excel, but could be released as CSV file or so.
- Data underlying Fig. 2a: I would like to see the data used to plot this. Summarising per length-bin as in the Figure legend would be sufficient.
- Data underlying Fig. 3c: I would like to see a table of the F3-statistics including error bars for each plotted reference group on the map.

Those are the ones I would like to check myself, and I would like to wait with my final recommendation regarding publication of the article until I have had a chance to review them.

Further general comments:

- In general I think the ethnic term "Rapanui" as opposed to the name of the island "Rapa Nui" should be properly introduced as denoting a people and their ancestors. Right now, the text uses the ethnic term without introduction.

- I don't think the current title of the paper accurately summarises its content. "A genomic history of the Rapanui" sounds way more general than what the paper actually does, which is a) reporting new

ancient genomic data from 15 samples, and b) use it to address mostly two important questions. I can come up with many research other questions about the "genomic history of the Rapanui" that are not addressed by this paper. So I would encourage the authors to come up with a title that summarises their two main findings better.

- All genomic lengths, as used in IBD sharing, ROH, and tract length distributions, should - in my view - be given in genetic map lengths rather than physical lengths, so centi-Morgans (cM) instead of basepairs (bp). The recombination rate isn't constant across chromosomes, so cM is the more appropriate unit. I think also the binning should be based on cM rather than bp. If there are good reasons for not following this common practice, fine, but otherwise I suggest to change the reporting (and binning?) of the respective analyses.

Comments on Figures:

- Figure 1a: The color scale seems funny to me. I would reverse it, so having deeper red for higher affinity.
- Figure 1b: I don't think this figure works well. There are too many plots, and the important details are far too small to appreciate. Perhaps one could show one plot for all Ancient Rapanui, one for all Present-day Rapanui, and one of the two Polynesians? Alternatively, use a different visualisation style, like bar-charts using only IBD tracts above a certain length threshold? I think that should already show the signal well (given that one focuses on the right hand sides of the plots anyway). The scale goes from 5 to 8, which roughly corresponds to 0.1 cM to 100 cM, but really the interesting part is arguably somewhere between 2 and 50 cM, I think.
- Figure 3b: "Totonac" is used as a group label in the D-statistics along the x-axis. Perhaps it would be helpful to highlight that one in the map in Figure 3c, instead of the 5 groups that are not really used elsewhere as far as I could see.

Specific comments:

L36: "in the last 1500 years" -> really? Didn't the story start around 1250 CE, so more like "in the last 800 years"?

L37: 'rediscovered' -> I don't fully understand why this is put in quotation marks. Europeans weren't the first, so the verb "rediscovered" is appropriate, whereas "discovered" is not. I think either use "rediscovered" without quotation or "discovered" in quotation, right?

L40: "Notwithstanding the achievements of the first Polynesian settlers, ..." -> I don't understand how this half-sentence is connected to the second part of the sentence "..., the history of the Rapanui has been presented as a warning tale of resource overexploitation". What does the first part of the sentence has to do with the second part? In particular, what does the word "Notwithstanding" here mean exactly?

L141: "... according to the museum records, found in Rapa Nui". I think this will make every reader curious on some more details of the provenience of these samples. How did they end up in the museum? Around when and where were they "found", and what does "found" mean? Was it an archaeological excavation, or some rescue-excavation, or some illegal looting of a cemetery? If none of that is known, then also that would be valuable information.

L150-159: Great, see also my comments at the top. Good to have this in, please keep such a summary of the community engagement in when having to make cuts later on!

L201 reference to MDS analyses: Please link Supplementary Figures directly from the main text and not just through their section. In this case, just referring to "SI 7", without pointing to an actual Supplementary Figure is a bit of a detour. I would suggest to go through the entire text again and make sure that (most?) of the Supplementary Figures and Tables are actually linked from within the main text. It would make it much easier to read. For example, I totally trust the authors on how they do MDS, don't need the details, just want to see the plots!

L232: The section title "Social structure in Rapanui" is again too general, I find. There is really only very specific aspects of "social structure" that genetic data can inform on (in particular without any proper archaeological data that comes with it). I would try to find something more accurate, that describes what is done in this section, which is about inbreeding and pairwise biological relationships. So perhaps something like "biological connectedness and reproductive practice" or something like that?

L274 "simulation results" -> But simulations have not yet been introduced yet, at this point in the text.

Referee #3 (Remarks to the Author):

Review of Moreno-Mayar et al. A genomic history of the Rapanui

The paper under review employs genetic analysis of DNA extracted from 15 ancient Rapanui individuals to address longstanding and contentious questions concerning the history of the Rapanui people over the past millennium. Specifically, these questions pertain to:

1. A purported population collapse around 1600 CE.
2. The trajectory of population growth since the initial settlement around 1250 CE.
3. The presence of genetic admixture from Native American populations.

These inquiries have been the subject of intense scholarly debate but have remained unresolved due to limitations such as the scarcity of available human samples, low genome sequencing coverage (0.0004-0.0041X) in prior DNA studies, and the utilization of limited techniques for admixture analysis.

This paper represents a significant advancement in all these facets of inquiry. The study benefits from a substantial sample size, with 15 ancient Rapanui human bone specimens, as well as precise radiocarbon dating, which includes correction for marine diet influences. Furthermore, the research employs deep whole-genome sequencing (0.4-25.6X coverage) and employs a diverse array of statistical correlation methods. These methodological advancements enable the authors to draw the following conclusions:

1. There is no evidence to support the occurrence of a population collapse in 1600 CE.
2. The Rapanui population experienced a continuous increase to reach fewer than 2000 individuals by the time of the Peruvian slave raids in the 1860s.
3. There exists genetic admixture comprising 6.5 to 12% of Native American ancestry, which is linked to Pacific Coast South American populations. The estimated admixture event is dated to 1400-1500 CE, utilizing a novel Bayesian approach.

As such, this paper offers a compelling resolution to the long-standing debates surrounding the Rapanui population history and establishes new benchmarks for genomic research. I perceive this work to be of significant interest to the broader scientific community engaged in genomic research, and I recommend its publication in Nature in its current form.

Referee #4 (Remarks to the Author):

Moreno-Mayar and collaborators analyzed 15 ancient genomes from Rapa Nui in order to evaluate i) a hypothesis about population collapse in the 1600s; ii) the potential admixture between Native American and Polynesian populations before European contact. The latter has been highly debated and previous studies did show evidence of admixture pre-European arrival in Rapa Nui and other islands in Polynesia (Moreno-Mayar et al., 2014; Ioannidis et al., 2020). However, these previous studies did not include genetic data from ancient individuals, while the research that did (Fehren-Schmitz et al., 2017) failed to identify evidence of such admixture.

Data was generated and analyzed using standard protocols in the field. They implemented an approach to imputed diploid genotypes in the ancient genomes newly generated. While this can be questioned due to the introduction of biases in the imputed data (even more considering these are ancient genomes), the authors properly evaluated the process by downsampling tests and repeating some analyses using pseudo-haploid data. In addition to the tests already implemented, I would recommend authors run a D-statistic of the form $D(\text{NoImputed}, \text{Imputed}; \text{reference dataset}; \text{YRI})$ to evaluate an excess of allele sharing of the imputed individuals and the different populations from the reference imputation panel.

In SI materials, page 19, authors say: "Imputation of the 1X downsampled 'Ancient Rapanui' genome yielded the second least accurate results. Such results can be attributed to ancient genomes containing more errors coming from sequencing, post-mortem damage, and contamination that can have some impact on imputation and on the validation data." Can this claim be accompanied by a direct evaluation of the accuracy per position and type of substitution (e.g. transitions vs. transversion)?

Reference dataset: it isn't clear throughout the manuscript which reference panel was used and why. For example, was the data from Ioannidis et al., 2020, 2021 used? To date, this seems to be the most comprehensive sampling performed across Polynesia. If data was not included, please explain why. Otherwise, include it in the analysis.

Why was Totonac used as a representative of Native American ancestry?

Similarly to Figure S11 and S12, a D-statistic of the form $D(X, \text{AncRapanui}; \text{South America}; \text{YRI})$ can be performed to evaluate if AncRapanui has a higher allele sharing with a specific Native American population compared to other Polynesian populations. X here represents different groups in Polynesia, not only 'Ancient Polynesians'.

There is a method to estimate ROH that allows pseudo-haploid data (hapROH; Ringbauer et al., 2021) and may be more appropriate in this context than the version in PLINK. Authors should implement this analysis in both, unimputed and imputed versions of their dataset.

The authors failed to discuss some inconsistencies between previous publications, particularly the time and source of Native American admixture. The admixture identified in Ioannidis et al., 2020 is described as a single event in eastern Polynesia dated to around 1,200 CE, before the peopling of Rapa Nui. Furthermore, this is associated with populations in present-day Colombia. Can the authors elaborate on the differences between studies and their implications? Is their study suggesting an additional admixture event to the one suggested by Ioannidis et al., 2020?

Author Rebuttals to Initial Comments:

We would like to thank the referees for providing valuable feedback to our manuscript. Please find below our detailed response to the referees' comments (in black), our replies (in blue), and specific changes to manuscript text (*in italics*). Note that cited lines correspond to line numbers in the newly submitted manuscript.

Referee #1: archaeology of the Pacific

This is a well-written paper which presents data and opinions on two issues that are of major importance in the prehistory of Rapanui, or Easter Island. It is entirely based on genetic data. Since I am not a geneticist I am not able to comment on the technical aspects of this paper, but I detect that the arguments are well presented and the paper achieves a high level of competence in the analysis of ancient DNA and the statistical techniques involved in such analyses.

We thank the referee for the positive comments.

I did have a couple of issues of uncertainty.

One is the possibility that the radiocarbon dates, as plotted in figure S1 in the supplementary information, do not seem to exclude completely the possibility that the 15 skulls analysed postdate the arrival of Roggeveen in 1722, and this possibility leads on to another possibility, that the Native American part of the genome could also be post-contact, given that many other ships visited the island later in the 18th century. However, the modelling presented by the authors is fairly suggestive that the Native American contact is genuinely pre-Roggeveen. But I would note that both the argument for Native American admixture and also the argument that there was no population collapse in the 17th century are based on statistical simulations rather than direct observations of skeletal or archaeological materials from the times in question. This does not mean that the authors conclusions are wrong, but it seems to me that ancient DNA from some petrous bones dating unambiguously to before 1500 CE will need to be analysed before there is absolute certainty about the American contact issue.

We agree with the referee about the possibility that the 15 'Ancient Rapanui' whole-genomes we sequenced in this study could postdate 1722. However, as discussed in the main text and Supplementary Section S1, one of the three objectives of this work is also to contribute to future repatriation efforts. In our view, demonstrating that all 15 are most closely related to present-day Rapanui is a significant finding that should help repatriation efforts, regardless of their age.

In what follows, we comment on the remaining two key findings of our work: no evidence of population collapse in the 1600s and Transpacific pre-European contact between the Polynesian ancestors of Rapanui with Native Americans. To test the collapse hypothesis

using genomic data, we necessarily require data from individuals that postdate the proposed collapse. In case of a collapse, the genomes of such individuals would show a decrease in diversity characteristic of a population bottleneck. For testing the Transpacific contact hypothesis, we rely on statistical modelling techniques based on well-established Mendelian laws of inheritance.

No collapse in the 1600s

The ages of the 15 'Ancient Rapanui' individuals place them in the ideal time window to reach one of the main conclusions of our work—no evidence of a population collapse in the 1600s. That is, the 15 individuals postdate the 1600s but are unlikely to be affected by the well-documented population decline and forced repatriations that followed the 1860s Peruvian slave raids ^{1,2}. Assuming there was indeed a population collapse in the 1600s, if the 15 'Ancient Rapanui' genomes pre-dated the 1600s, they would not carry the tell-tale genomic diversity patterns of the collapse. Furthermore, our study adds biological/genomic data to the ever-growing body of evidence (bioanthropological, archaeological and historical) rejecting the so-called 'Huri Moai cultural phase' ³⁻⁵.

We introduced this rationale in lines 278:

"We use the 15 'Ancient Rapanui' genomes to test this hypothesis using biological data, since they postdate the proposed collapse and, yet they are unlikely to be affected by the demographic events that followed the 1860s Peruvian slave raids ^{1,7}. "

and 428:

"Although these individuals—radiocarbon-dated to the 1800s—postdate the first arrival of Europeans in Rapa Nui, they do not bear European admixture. More importantly, they pre-date the Peruvian slave raids ⁷ and the devastating smallpox outbreak of the 1860s, which reduced the Rapanui population down to an estimated 110 individuals ¹. Although this post-European contact population decline could have obscured the relationship between ancient and present-day Rapanui, we do not observe genetic discontinuity between these two populations. As such, this dataset constitutes a close representation of Rapanui genomic diversity after the initial peopling of the island and before European contact. "

Transpacific pre-European contact with Native Americans

We highlight that the observed genomic diversity patterns in the 15 'Ancient Rapanui' individuals, regardless of the statistical inference of the admixture date are strongly inconsistent with post-European admixture. Here we enumerate the relevant lines of evidence that are readily measurable from the genomic data, the same way they would be

measured from a pre-1722 genome and without using a statistical method to date the admixture event:

1. Native American admixture proportions in 'Ancient Rapanui': For the 15 'Ancient Rapanui', we estimated they bear on average ~10% (6.0-11.4%) Native American admixture. For the earliest individuals in this study (RN06: 9.1%, RN11: 10% and RN15:11.3%) in particular, these estimates are too low for a European-mediated Native American admixture event. The hypothetical scenario where Native American genomic ancestry is lost from the Rapanui population at the fastest possible rate, is one where a single Native American individual enters the population and produces offspring exactly in 1722. After this, their offspring must only mix with individuals not carrying any Native American genomic ancestry. Under this extreme scenario, American admixture proportions will be 50% in the first generation (1722), 25% in the second (~1750), and 12.5% in the third (~1780). Since the three earliest individuals in this study are unlikely to have died much later than 1800 (Pinart collection dates to 1877, and see calibrated age ranges in Figure S1), we consider their Native American admixture proportions cannot be explained by post-1722 admixture, even under the most extreme scenario. Instead, the largely uniform Native American admixture proportions are indicative of an earlier admixture event between the Polynesian Rapanui ancestors and Native Americans^{6,7}. Importantly, historical archives suggest that although European contact started in 1722, exchanges became more generalised decades after that date⁸, thus decreasing the compatibility of the observed genomic data with a post-European contact admixture event with Native Americans.

2. No European admixture in 'Ancient Rapanui': Since it is unlikely that post-contact admixture only involved individuals carrying Native American genomic ancestry only, we expect to detect European admixture in at least one of the 15 'Ancient Rapanui' assuming they descend from admixture events post-dating 1722. However, using methods that rely on different features of the genomic data, such as *f*-statistics (Supplementary Section S8), ADMIXTURE (Supplementary Section S13) and local ancestry inference (Supplementary Section S14), we showed that none of the 'Ancient Rapanui' individuals carry European admixture. By contrast, European admixture has been confidently detected in post-contact Rapanui, *e.g.*,^{7,9}.

3. Native American ancestry tract length distribution in 'Ancient Rapanui': We found that the Native American ancestry tracts present in the 'Ancient Rapanui' are shorter than those resulting from a recent admixture event. For instance, see Figure 4b, where European tracts in a present-day Rapanui (from a more recent admixture event) are considerably longer than Native American tracts. Using a similar rationale as that detailed in point 1., the observed Native American tract length distribution is not compatible with a post-1722 Native American admixture event. In this case, we note that our local ancestry inference should be sufficiently powered given Polynesian, Native American and European genomic ancestries diverged from each other over 15,000 years ago¹⁰⁻¹².

Thus, line 474 in the main text now reads:

"Using ADMIXTURE, local ancestry inference and f-statistics, we detected ~10% Native American admixture (and no European admixture) in all the 15 'Ancient Rapanui'; a genomic diversity pattern not consistent with a post-European contact admixture event. We confidently dated this admixture event to ~1400-1500 CE (i.e., well before Columbus and European contact in 1722) using a novel method that relies on joint genetic and radiometric data. "

A couple of small points:

Line 637: ref is wrong. Should be Kirch, Patrick V.

We thank the reviewer for noting this mistake. We have now fixed the reference in the main text and Supplementary Information file.

Line 73: double-hulled canoes? I would suggest "...possibly using double-hulled canoes", given that there is no proof that their boats actually were double-hulled. Large outrigger canoes with platforms for passengers and cargo might also have been used, as in Micronesia, although this is not an essential issue for discussion in this paper. No one will ever know for certain what their boats looked like!

We agree with the referee. We have now edited the text as suggested.

Line 73 now reads:

"Despite its remoteness, archaeological and genetic evidence shows that Polynesian peoples from the west had already reached the island possibly using double-hulled canoes by ~1250 CE²⁻⁴. "

The Polynesian skulls from the Botocudo source near Rio sound to me like a case of mis-labelling. I cannot prove it, but the idea of Polynesians migrating all the way across South America to the vicinity of Rio de Janeiro seems rather unbelievable. Perhaps a member of the museum staff in Rio many decades ago muddled up some skulls before they were properly labelled. This does not affect their significance as skulls of Polynesian ancestry, but I would state on lines 193-194 "... two skulls of previously-demonstrated Polynesian genetic ancestry, but uncertain origin location, stored in the Museu Nacional in Rio de Janeiro, Brazil. We refer to these as 'Ancient Polynesians',"

This is a good suggestion and we have now edited the text accordingly.

Line 195 now reads:"[...] *and ancient genome data from two individuals of previously-demonstrated Polynesian genetic ancestry, but uncertain origin location, stored in the Museu Nacional in Rio de Janeiro, Brazil. We refer to the latter as 'Ancient Polynesians'.*"

Referee #2: ancient genomics

This article presents the first high-quality ancient genomes from Rapa Nui, the most remote inhabited island in the Polynesian triangle. The data and paper are highly relevant to address important questions of human history in the Pacific. The authors did overall a great job in generating the data, its analyses, and the article is generally also very well written.

I want to also applaud the authors on their description of the community engagement in the main text, which puts this important part of this type of research right into the focus where it belongs.

We thank the referee for the positive remarks.

My main criticism concerns the lack of tabular data attached to the paper, which also prevented me from reviewing this work to the level that I would like to. I think it should be self-evident that all quantitative figures in the main article should have their underlying data accessible as Supplementary Datasets. While the Supplement lists many needed details of the analyses, the actual underlying data, most importantly the IBD sharing tables, but also results of F-statistics etc. are not available in table-form, as far as I could see.

While I have no a priori reason to doubt the authors' expertise in analysing the data, I would like to be fully convinced of some of the patterns the authors report, which is only possible with a direct look at the data.

In particular:

- Data underlying Fig 1a: I would like to see the F3-statistics in table form including error bars
- Data underlying Fig 1b: I would like to see at the minimum for each individual a table of length-stratified cumulative IBD (e.g. stratified using bins of certain length windows) with each other reference individual/group. Even better would be a table listing all shared IBD segments, perhaps above a certain length cutoff. That might then be too large a table for Excel, but could be released as CSV file or so.
- Data underlying Fig. 2a: I would like to see the data used to plot this. Summarising per length-bin as in the Figure legend would be sufficient.
- Data underlying Fig. 3c: I would like to see a table of the F3-statistics including error bars for each plotted reference group on the map.

Those are the ones I would like to check myself, and I would like to wait with my final recommendation regarding publication of the article until I have had a chance to review them.

Following the referee's suggestion, we now include supplementary Tables S3-14, 16-19. These tables include raw results for all *f*-statistics, IBD sharing and ROH estimates, for all versions of the data. We reference the corresponding raw data tables in the captions of all the figures that rely on these data.

Further general comments:

- In general I think the ethnic term "Rapanui" as opposed to the name of the island "Rapa Nui" should be properly introduced as denoting a people and their ancestors. Right now, the text uses the ethnic term without introduction.

We now define Rapanui in the text.

Line 76 now reads:

"The following five centuries saw the Rapanui (demonym for the island) society flourish and develop a culture characterised by iconic giant stone statues (moai) and monumental stone platforms (ahu)."

- I don't think the current title of the paper accurately summarises its content. "A genomic history of the Rapanui" sounds way more general than what the paper actually does, which is a) reporting new ancient genomic data from 15 samples, and b) use it to address mostly two important questions. I can come up with many research other questions about the "genomic history of the Rapanui" that are not addressed by this paper. So I would encourage the authors to come up with a title that summarises their two main findings better.

Following the referee's suggestion, we now propose a new title for our manuscript:

"Ancient Rapanui genomes reveal a resilient island population and pre-European Transpacific contact with Native Americans"

- All genomic lengths, as used in IBD sharing, ROH, and tract length distributions, should - in my view - be given in genetic map lengths rather than physical lengths, so centi-Morgans (cM) instead of basepairs (bp). The recombination rate isn't constant across chromosomes, so cM is the more appropriate unit. I think also the binning should be based on cM rather than bp. If there are good reasons for not following this common practice, fine, but otherwise I suggest to change the reporting (and binning?) of the respective analyses.

We agree with the referee's suggestion and we have now homogenised the units for all figures showing genomic lengths. This change involved the *IBDseq*¹³ and ROH results. The remaining genomic length results were already reported in cM. Since a genetic position for every nucleotide is not included in the HapMap¹⁴ genetic map, we used a linear interpolation

based on the two genetic map coordinates immediately flanking any given physical position to obtain an equivalent genetic map coordinate. Figures 1b, S19, 2a, S22 are now displayed using genetic map lengths.

Comments on Figures:

- Figure 1a: The color scale seems funny to me. I would reverse it, so having deeper red for higher affinity.

We explored the alternative colour scheme as suggested by the referee (Figure R1). However, we consider the original colour scheme conveys the f_3 differences for Polynesians more accurately. In particular, present-day Rapanui and the 'Ancient Polynesian' individuals look identical even though they are not (see below).

Figure R1. Figure 1a with inverted color scheme.

Nevertheless, we agree with the reviewer that this plot could be difficult to interpret at a quick glance. Therefore, we updated the legend to explain our colour scheme and avoid any confusion. Furthermore, to support this result and provide the reader with more details, we now include former Figure S10 as Extended Data Figure 1. Figure 1 and Extended Data Figure 1 captions cross-reference each other and the raw results Supplementary Table as follows:

Figure 1a caption reads:

"**a.** f_3 -statistics of the form $f_3(\text{Yoruba}; X, \text{Ancient Rapanui})$, for each population in a worldwide genotype panel including 755,094 SNP sites. For each ancient individual, we sampled one

random allele at each site and pooled these 'pseudohaploid calls' to estimate allele frequencies. In addition, we include data from two previously published Polynesian individuals originally from an unknown island. The whiter a point is, the greater shared drift between a population (X) and the ancient individuals sequenced in this study. Raw results are presented in Extended Data Figure 1 and Table S4."

Extended Data Figure 1 and its caption are now displayed as follows:

"Extended Data Figure 1. f_3 -statistics measuring shared drift between 'Ancient Rapanui' individuals and other present-day and ancient populations. We estimated the 'Ancient Rapanui' allele frequencies for the sites included in the SNP array dataset (Section S5) and computed f_3 -statistics of the form $f_3(\text{Yoruba}; X, \text{Ancient Rapanui})$. For each population, the point represents the point estimate for f_3 , and error bars correspond to the 95% confidence intervals. Non-Polynesian populations are coloured according to their broad continental ancestry and Polynesian populations are coloured according to their island of origin. **a.** f_3 -statistics results for pseudo-haploid calls. **b.** f_3 -statistics results for imputed diploid genotypes. Panel a. corresponds to the results shown in Figure 1a. Raw results for these f_3 -statistics are presented in Table S4."

- Figure 1b: I don't think this figure works well. There are too many plots, and the important details are far too small to appreciate. Perhaps one could show one plot for all Ancient Rapanui, one for all Present-day Rapanui, and one of the two Polynesians? Alternatively, use a different visualisation style, like bar-charts using only IBD tracts above a certain length threshold? I think that should already show the signal well (given that one focuses on the

right hand sides of the plots anyway). The scale goes from 5 to 8, which roughly corresponds to 0.1 cM to 100 cM, but really the interesting part is arguably somewhere between 2 and 50 cM, I think.

We agree with the referee. Following the referee's suggestion, we have created a simplified version of Figure 1b, where we show the average cumulative length of segments shared IBD between Polynesian islanders, stratified by segment length. In brief, for a given pair of populations, each bar represents the mean fraction of the genome that all possible pairs of individuals (between populations) share IBD (as estimated by *IBDseq*¹³).

In addition, we now include Figure S19 in the Supplementary Information, which shows the *IBDseq* results for each individual, instead of the pooled means. *IBDseq* estimates, stratified by length, for each pair of individuals are included in Table S14.

Figure 1b and its caption are now displayed as follows:

"Figure 1 | Shared drift and identity by descent (IBD) between 'Ancient Rapanui' and present-day populations. a. [...] b. Average IBD sharing between pairs of individuals from

different Polynesian groups as estimated using IBDseq. For all possible pairs of individuals between two groups, e.g., the 45 possible pairs between the 15 'Ancient Rapanui' and the three present-day Rapanui, we show the average cumulative length of segments shared IBD, stratified by segment length (color scheme). We show results for the 15 'Ancient Rapanui', three representative present-day Rapanui with low European admixture and two 'Ancient Polynesians' with unknown sampling location (see orange panel labels). For this analysis, we imputed the ancient individual sequence data to obtain diploid genotypes. Results for each individual (not pooled means) are presented in Figure S19 and IBDseq estimates stratified by length are presented in Table S14. "

Figure S19 and its caption are displayed as follows:

"Figure S19. IBD sharing between each ancient individual and 78 ancient and present-day Polynesians and Fijians, as estimated using IBDseq. For each pair of individuals, we plot the cumulative length of the genome shared IBD across increasing IBD segment length bins. We show results for the 15 'Ancient Rapanui', three representative present-day Rapanui with low European admixture and two 'Ancient Polynesians' with unknown sampling location (see orange panel labels). For this analysis, we imputed the ancient individual sequence data to obtain diploid genotypes. Colours correspond to the individual

island of origin. This figure shows results for each individual included in the pooled mean data presented in Figure 1b. Individual IBDseq estimates stratified by length are presented in Table S14."

- Figure 3b: "Totonac" is used as a group label in the D-statistics along the x-axis. Perhaps it would be helpful to highlight that one in the map in Figure 3c, instead of the 5 groups that are not really used elsewhere as far as I could see.

We now indicate the sampling location of the Totonac in Figure 1a (see above). Note that we keep the labels in Figure 3c to highlight the South American populations that yield the largest f_3 -statistics.

Specific comments:

L36: "in the last 1500 years" -> really? Didn't the story start around 1250 CE, so more like "in the last 800 years"?

For conciseness (since this is a sentence in the abstract), we simplified the text and line 35 now reads:

"Despite its remoteness, archaeological and genetic evidence shows that Polynesians peopled Rapa Nui from the west hundreds of years before Europeans arrived and renamed it Easter Island in 1722 CE."

L37: 'rediscovered' -> I don't fully understand why this is put in quotation marks. Europeans weren't the first, so the verb "rediscovered" is appropriate, whereas "discovered" is not. I think either use "rediscovered" without quotation or "discovered" in quotation, right?

We agree with the referee and we have now simplified this sentence. We show the new version in the point above.

L40: "Notwithstanding the achievements of the first Polynesian settlers, ..." -> I don't understand how this half-sentence is connected to the second part of the sentence "..., the history of the Rapanui has been presented as a warning tale of resource overexploitation". What does the first part of the sentence has to do with the second part? In particular, what does the word "Notwithstanding" here mean exactly?

We use 'notwithstanding' as a synonym for 'despite' to avoid a repetition as the latter is used in the same paragraph. We agree the sentence was a bit convoluted. We have now edited it as follows

Line 41 now reads:

"Notwithstanding the achievements of the first Polynesian settlers, i.e., exploring the Pacific and establishing flourishing societies across remote Polynesian islands in a few thousand years, the history of the Rapanui has been presented as a warning tale of resource overexploitation that would have culminated in a major population collapse. "

L141: "... according to the museum records, found in Rapa Nui". I think this will make every reader curious on some more details of the provenience of these samples. How did they end up in the museum? Around when and where were they "found", and what does "found" mean? Was it an archaeological excavation, or some rescue-excavation, or some illegal looting of a cemetery? If none of that is known, then also that would be valuable information.

Unfortunately, information about the collection process is limited. Following the referee's suggestion, we have added a paragraph in Supplementary **Section S1** where we provide background information and refer to the Museum's collection online database (we refer to this section in the main text). The new paragraph, which we also cite in the main text, reads:

"As for any collection from the 19th century, little is known about the circumstances under which the individual bone remains were collected. Some information can however be found in the Musée de l'Homme online database (<http://colhelper.mnhn.fr/>). Out of the 15 individual samples, 11 are reported to have been collected by Alphonse Pinart in 1877 and the remaining by Alfred Métraux in 1935. According to the museum archives, the bone remains come from either two locations on the island of Rapa Nui: La Pérouse Bay and Vaihou. By confirming that the individuals have Rapanui origin based on genetic data, we expect to contribute to future repatriation efforts. "

L150-159: Great, see also my comments at the top. Good to have this in, please keep such a summary of the community engagement in when having to make cuts later on!

We thank the reviewer for this positive comment.

L201 reference to MDS analyses: Please link Supplementary Figures directly from the main text and not just through their section. In this case, just referring to "SI 7", without pointing to an actual Supplementary Figure is a bit of a detour. I would suggest to go through the entire text again and make sure that (most?) of the Supplementary Figures and Tables are actually linked from within the main text. It would make it much easier to read. For example, I totally trust the authors on how they do MDS, don't need the details, just want to see the plots!

Following the referee's suggestion, we have included references to specific supplementary figures throughout the main text.

L232: The section title "Social structure in Rapanui" is again too general, I find. There is really only very specific aspects of "social structure" that genetic data can inform on (in particular without any proper archaeological data that comes with it). I would try to find something more accurate, that describes what is done in this section, which is about inbreeding and pairwise biological relationships. So perhaps something like "biological connectedness and reproductive practice" or something like that?

We agree with the referee. The new title for this section is:

"Biological kinship, relatedness and runs of homozygosity in Rapanui "

L274 "simulation results" -> But simulations have not yet been introduced yet, at this point in the text.

We thank the referee for noting this. We now include a reference to the upcoming results in the text.

Line 285 now reads:

"Therefore, we follow common practice and interpret our subsequent simulation results (see below) qualitatively and in relative terms, i.e., by focusing on the estimated effective population size trajectories and the timing of their shifts. "

Referee #3: radiocarbon dating

Review of Moreno-Mayar et al. A genomic history of the Rapanui

The paper under review employs genetic analysis of DNA extracted from 15 ancient Rapanui individuals to address longstanding and contentious questions concerning the history of the Rapanui people over the past millennium. Specifically, these questions pertain to:

1. A purported population collapse around 1600 CE.
2. The trajectory of population growth since the initial settlement around 1250 CE.
3. The presence of genetic admixture from Native American populations.

These inquiries have been the subject of intense scholarly debate but have remained unresolved due to limitations such as the scarcity of available human samples, low genome sequencing coverage (0.0004-0.0041X) in prior DNA studies, and the utilization of limited techniques for admixture analysis.

This paper represents a significant advancement in all these facets of inquiry. The study benefits from a substantial sample size, with 15 ancient Rapanui human bone specimens, as well as precise radiocarbon dating, which includes correction for marine diet influences. Furthermore, the research employs deep whole-genome sequencing (0.4-25.6X coverage) and employs a diverse array of statistical correlation methods. These methodological advancements enable the authors to draw the following conclusions:

1. There is no evidence to support the occurrence of a population collapse in 1600 CE.
2. The Rapanui population experienced a continuous increase to reach fewer than 2000 individuals by the time of the Peruvian slave raids in the 1860s.
3. There exists genetic admixture comprising 6.5 to 12% of Native American ancestry, which is linked to Pacific Coast South American populations. The estimated admixture event is dated to 1400-1500 CE, utilizing a novel Bayesian approach.

As such, this paper offers a compelling resolution to the long-standing debates surrounding the Rapanui population history and establishes new benchmarks for genomic research. I perceive this work to be of significant interest to the broader scientific community engaged in genomic research, and I recommend its publication in Nature in its current form.

We would like to thank the referee for their positive comments.

Referee #4: ancient genomics

Moreno-Mayar and collaborators analyzed 15 ancient genomes from Rapa Nui in order to evaluate i) a hypothesis about population collapse in the 1600s; ii) the potential admixture between Native American and Polynesian populations before European contact. The latter has been highly debated and previous studies did show evidence of admixture pre-European arrival in Rapa Nui and other islands in Polynesia (Moreno-Mayar et al., 2014; Ioannidis et al., 2020). However, these previous studies did not include genetic data from ancient individuals, while the research that did (Fehren-Schmitz et al., 2017) failed to identify evidence of such admixture.

Data was generated and analyzed using standard protocols in the field. They implemented an approach to imputed diploid genotypes in the ancient genomes newly generated. While this can be questioned due to the introduction of biases in the imputed data (even more considering these are ancient genomes), the authors properly evaluated the process by downsampling tests and repeating some analyses using pseudo-haploid data. In addition to the tests already implemented, I would recommend authors run a D-statistic of the form $D(\text{NoImputed}, \text{Imputed}; \text{reference dataset}; \text{YRI})$ to evaluate an excess of allele sharing of the imputed individuals and the different populations from the reference imputation panel.

We followed the referee's suggestion and estimated D -statistics of the form $D(\text{NoImputed}, \text{Imputed}; \text{reference dataset}; \text{YRI})$, using the different reference populations in the imputation reference panel (1000 Genomes). Here, we called diploid genotypes for two high-depth 'Ancient Rapanui', RN13 (17x) and RN14 (26x) ('NoImputed'), and compared them to their imputed versions. While we found a trend for more genetically similar populations to be closer to the imputed dataset, no D -statistic resulted in a significant result (at a significance level of ~ 0.001). We include here the resulting plot as well as the text we added to Supplementary Section S6.2.1.4.

"To investigate the bias introduced by imputation (see also Bárbara's paper for extensive work on the topic), we determined whether the affinity of the imputed 'Ancient Rapanui' genomes to the populations included in the imputation reference panel is increased. To do so, we estimated D -statistics of the form $D(\text{high-coverage genome } X, \text{ imputed } 1x \text{ genome } X; \text{ reference population, Yoruba})$, where genome X represents either RN13 or RN14 (see method details in Section S8). To produce a more accurate ground truth using the non-imputed high-coverage genomes (called diploid genotypes), which are affected by C-to-T damage, we restricted this analysis to transversion sites. We observed that the reference populations that yielded the most negative D values, i.e., being closer to the imputed genomes, tended to be of East Asian or American origin (e.g., Han Chinese in Beijing (CHB), Southern Han Chinese (CHS), Japanese in Tokyo (JPT) and Peruvian in Lima (PEL)), while we found European populations on the other extreme. This ordering might reflect the ancestry of the reference haplotypes which the imputation algorithm copied from, as the

'Ancient Rapanui' are expected to be more closely related to East Asian individuals and they carry ~10% of Native American ancestry. However, for the two 'Ancient Rapanui' genomes and all reference populations, the *D*-statistics 99%-confidence interval included 0, which supports the idea that none of the imputed genomes had significantly higher (or lower) affinity to any of the reference populations when compared to their high-coverage counterpart (Figure S8, Table S3). Importantly, the magnitude of the *D*-statistics is extremely small, i.e., in the order of 10^{-4} , 100 times smaller than the largest *D*-statistics absolute values obtained when we test for Native American admixture, e.g., Figure S11. This difference suggests that there is only a small number of SNPs that can be used for *D*-statistics tests due to the high genotype concordance between the imputed and called diploid genotypes, even when the same populations that were used for imputation (1000 Genomes) are included as a test population. Based on these results, we conclude that imputation did not produce a significant bias towards the imputation reference panel."

Figure S8 is now displayed as follows:

Figure S8. D-statistics testing for potential imputation bias towards reference panel populations. We computed D-statistics of the form $D(\text{high-depth genome, imputed 1x genome; 1000G population, Yoruba})$ for two 'Ancient Rapanui' genomes: **a.** RN13 (ancient Rapanui genome at 17x) and **b.** RN14 (ancient Rapanui genome at 26x). We kept only transversion polymorphisms with a minor allele frequency (MAF) above 0.5%, and the imputed genotypes had been filtered for $\text{MAF} > 1\%$ and $\text{GP} > 0.99$. Reference populations in 1000 Genomes panel: ACB: African Caribbean in Barbados, ASW: African ancestry in Southwest USA, BEB: Bengali from Bangladesh, CDX: Chinese Dai in Xishuangbanna, China, CEU: Utah residents with Northern and Western European ancestry, CHB: Han Chinese in Beijing, China, CHS: Southern Han Chinese, CLM: Colombian in Medellin, Colombia, ESN: Esan in Nigeria, FIN: Finnish in Finland, GBR: British in England and Scotland, GIH: Gujarati Indian from Houston, Texas, GWD: Gambian in Western Divisions in the Gambia, IBS: Iberian populations in Spain, ITU: Indian Telugu from the UK, JPT: Japanese in Tokyo, Japan, KHV: Kinh in Ho Chi Minh City, Vietnam, LWK: Luhya in Webuye, Kenya, MSL: Mende in Sierra Leone, MXL: Mexican ancestry in Los Angeles, California, PEL: Peruvian in Lima, Peru, PJI: Punjabi from Lahore, Pakistan, PUR: Puerto Rican in Puerto Rico, STU: Sri Lankan Tamil from the UK, TSI: Toscani in Italy, YRI: Yoruba in Ibadan, Nigeria. Points represent D-statistics, and error bars represent ~ 3.3 SEs (p -value of ~ 0.001 in a Z test). Numbers next to each point indicate the Z-score for each test. Raw results are reported in Table S3.

In SI materials, page 19, authors say: "Imputation of the 1X downsampled 'Ancient Rapanui' genome yielded the second least accurate results. Such results can be attributed to ancient genomes containing more errors coming from sequencing, post-mortem damage, and contamination that can have some impact on imputation and on the validation data." Can this claim be accompanied by a direct evaluation of the accuracy per position and type of substitution (e.g. transitions vs. transversion)?

This is an interesting yet difficult question. In our experience, evaluating imputation accuracy on ancient genomes is challenging due to the ancient genomes' higher error rates. As a result, even when the depth of the ancient genome is high, we expect differences between the observed genotypes and the ground truth. Moreover and relatedly, both the 'validation' genotypes (used to compute imputation accuracy) and the imputed genotypes themselves are impacted by ancient DNA *post-mortem* damage. Therefore, it is difficult to disentangle the effect on each of these two datasets. Recently, we conducted extensive simulations on the topic ¹⁵, and we showed that the impact of C-to-T substitutions on imputation is limited.

As it is an interesting question and following the referee's suggestion, we investigated this further in this work and assessed whether there were differences in imputation performance at CT SNPs (C/T and T/C) compared to the remaining SNPs. Since we relied on single-stranded libraries, which do not show an increased G-A substitution rate (Figure S3), we focused on CT SNPs.

We observed a very small difference in imputation accuracy between the two SNP types in the two ancient genomes we used, but we observed a similar difference when we make the same comparison using a present-day genome, P2077. To further explore this question, we evaluate the effect of deamination on ROH, an aggregate proxy, where the impact of spurious heterozygous calls can be visualized more promptly and can be easier to interpret. We describe the results in a new supplementary section 'S6.2.1.3. Evaluating *post-mortem* damage impact on imputation experiments' and included the text and the corresponding figure below.

"Differences in imputation accuracy between ancient and present-day genomes from the same population may be attributed to ancient genomes containing more errors coming from sequencing, post-mortem damage and contamination that can have some impact on imputation and on the validation data. We investigate the impact of post-mortem damage on our results using two downsampled (1x) high-depth ancient Rapanui genomes, RN13 and RN14. RN14 contains both non-UDG and UDG-treated libraries, which leads to a substantial increase of transition SNPs being affected by post-mortem damage, whereas the RN13 genome is comprised of UDG-treated libraries only. This difference is also reflected in their respective error rates (RN14: 0.65%; RN13: 0.07%, Figure S4). As a control, we conduct the same experiments on the imputed 1x P2077, a present-day Rapanui genome, that has a similar imputation accuracy as RN13.

When assessing imputation performance at SNPs affected by deamination (C/T and T/C), we found no considerable decrease in accuracy compared to non-C-T sites (Figure S7a,b,d), and similar differences were found in the present-day genome. Moreover, such decrease was smaller than the difference between the imputation accuracies of RN13 and RN14. We further estimated ROH using all SNPs in the 1000G UMICH dataset (Section S5.5) (MAF>1%) and restricting to transversion SNPs. We found that, for the same SNP subsets, the total ROH lengths in the imputed and high-depth genomes were similar (Figure S7d). Furthermore, when we focused on different ROH size categories, we found similar total lengths, except for RN14 when using all sites. For RN14, ROH with sizes between 12 cM and 20 cM are absent from the validation dataset (called diploid genotypes) when considering transitions in contrast to the ROH estimated using imputed genotypes. However, this difference between the validation and imputed datasets disappears when we restrict the analysis to transversions.

To exemplify our observations, we use chromosome 10 (Figure S7e), where we observed ROH that were either absent or were shorter in the high-depth RN14 (all sites) while there was a complete overlap between the two imputed (transversions only and all sites) and the high-depth transversion datasets. We interpret that the larger amount of deaminated SNPs in RN14 may have led to an excess of heterozygous sites that break ROH in the validation dataset. While this analysis is inconclusive regarding the impact of post-mortem damage

on imputation, it shows, on the one hand, how challenging it is to obtain a true validation dataset for damaged ancient genomes even when high-depth data is available and, on the other hand, that imputation can mitigate the effect of deamination, as shown in ⁵⁸.

Figure S7. Imputation accuracy at potentially deaminated sites. We compared imputation performance at C/T and T/C (C \leftrightarrow T) polymorphisms with the remaining SNPs when imputing downsampled to 1x Rapanui genomes: P2077, present-day genome, and RN13 and RN14, two high-depth 'Ancient Rapanui' genomes sequenced in this study (a, b and c). RN13's sequences are the result of UDG-treated libraries, whereas RN14 is the result of a mixture of UDG with non-UDG-treated libraries (Table S1). We also evaluated the impact of deamination in runs of homozygosity (ROH) detection for RN13 and RN14, using either imputed (no downsampling) or the high-depth data at all sites or at transversions only (d and e). a. Squared Pearson correlation, r^2 , between imputed dosages and high-depth genotypes as a function of minor allele frequency (MAF) when restricting to C \leftrightarrow T sites (triangles) and to the remaining sites (filled circles) when imputing 1x RN13, 1x RN14 and 1x P2077. b. Difference in r^2 between non-C \leftrightarrow T as a function of MAF for the curves in a. c. Non-reference discordance (NRD) per sample and for C \leftrightarrow T (purple) and non-C \leftrightarrow T (orange) SNPs. d. Total length of inferred ROH in RN13 and RN14 discriminated by ROH length for imputed and high-depth data at all SNPs vs. transversion polymorphisms. e. ROH segments detected in chromosome 10 comparing the abovementioned combinations of SNPs and data. We highlighted two regions in RN14 where either the ROH was absent or was shorter in the high-depth genome while being consistent across the remaining three combinations.

Reference dataset: it isn't clear throughout the manuscript which reference panel was used and why.

We have extended Supplementary Section S5, where we describe each of the datasets we used in each analysis. Furthermore, throughout the main text and the Supplementary Information we now refer to the dataset that we used in each analysis and refer to the appropriate Supplementary Section. For example, line 202 now reads:

"We explored the broad genetic affinities between the 15 ancient individuals and a panel of 30 worldwide populations (including Fiji, seven Polynesian islands and present-day Rapanui) genotyped across 755,094 SNP sites ^{22,40,41} (SI 5.1)."

and line 383 now reads:

"We used ALDER ⁶² and DATES ⁶³ (together with the SNP array dataset (SI 5.1)) to model the 15 'Ancient Rapanui' as a mixture of two source populations: "

For example, was the data from Ioannidis et al., 2020, 2021 used? To date, this seems to be the most comprehensive sampling performed across Polynesia. If data was not included, please explain why. Otherwise, include it in the analysis.

We agree with the referee that the Ioannidis et al. SNP array dataset would have been an interesting addition to our analyses. Unfortunately, although we have applied to access the data following the process detailed in the original publication and have sent all relevant documents weeks ago, we have not received a reply by the time we submit our revised manuscript. However, we consider that including such dataset would not be central for supporting or rejecting our main findings, *i.e.*, confirming the Rapanui origin of the 15 ancient individuals we sequenced, rejecting a population collapse in the 1600s and dating the Transpacific contact with Native Americans.

More importantly, we consider that the SNP array dataset we used (described in Supplementary Section S5.1) contains the best available reference data for present-day Rapanui. According to medical and historical records, present-day Rapanui in this dataset are unlikely to carry admixture from other Polynesian islands ¹⁶. We describe this dataset and this argument in line 202:

"We explored the broad genetic affinities between the 15 ancient individuals and a panel of 30 worldwide populations (including Fiji, seven Polynesian islands and present-day Rapanui) genotyped across 755,094 SNP sites ^{22,40,41} (SI 5.1). We used this panel—where all individuals were genotyped using the same SNP array—to maximise the overlap between the reference present-day Polynesian individuals and the present-day Rapanui

who, according to medical and historical records are unlikely to carry admixture from other Polynesian islands ⁴². "

Additionally, throughout the study, we use whole-genome data from two 'Ancient Polynesian' individuals ¹⁷. These data allowed us to validate our results on data with a similar error profile. Moreover, for analyses that rely on the contrast between two populations, e.g., tests using *f*-statistics, we use these data as a reference Polynesian population not carrying European or Native American admixture but is also affected by *post-mortem* damage. These data are introduced in line 194 and Supplementary Section S5.2.

Why was Totonac used as a representative of Native American ancestry?

We have now extended our description in Supplementary Section S5.1 to specify the Native American populations that are included in the SNP array reference panel. Furthermore, we repeated all Totonac tests using a Bolivian Native American population and show that the results of this test for detecting Native American admixture in the Rapanui are similarly powered regardless of the Native American proxy we use. In Supplementary Section S8.3, we now include a paragraph detailing these results and the expectations for these tests and introduce the specific analyses where we explore which ancient or present-day Native American populations are most closely related to the Native American population that admixed with the ancestors of the 'Ancient Rapanui'. This new paragraph in Supplementary Section S8.3 reads as follows:

*"We repeated the test for Native American admixture in the 'Ancient Rapanui', the present-day Rapanui and the 'Ancient Polynesians' using the Bolivian individuals in the SNP array panel (Section 5.1) as an alternative proxy. For all tests, we observed consistent results for the Totonac and the Bolivians as expected (Table S5). Since these tests are aimed solely at detecting Native American admixture in the Polynesian individuals, we expect them to be sufficiently powered regardless of the Native American population we use as a proxy. Assuming the 'Ancient Rapanui' bear Native American admixture, a *D*-statistic of the form $D(\text{Tonga, AncientRapanui}; \text{Native American proxy, Yoruba})$ will be proportional to the length of the branch between the common ancestor of the real 'admixing Native American population' and the 'Native American proxy' on the one hand, and the common ancestor of the Tongans, 'Ancient Rapanui' and the 'Native American proxy' on the other. Assuming that a) the divergence between the ancestors of Native Americans and East Asians dates to ~25 thousand years (ka) ago ^{19,72} and that b) the divergence between South Native American groups occurred at most 13 ka ago ^{46,48,50}, this *D*-statistic would be proportional to the drift that leads to a small population throughout ~12 ka ⁶⁶. In the following test (using the SGDP dataset, Section S5.3) and in Section S8.5 we use a more comprehensive reference panel to explore which ancient or present-day Native American populations are most closely related to the Native American population that admixed with the ancestors of the 'Ancient Rapanui'. "*

Similarly to Figure S11 and S12, a D -statistic of the form $D(X, \text{AncRapanui}; \text{South America}; \text{YRI})$ can be performed to evaluate if AncRapanui has a higher allele sharing with a specific Native American population compared to other Polynesian populations. X here represents different groups in Polynesia, not only 'Ancient Polynesians'.

Following the referee's suggestion, we have computed D -statistics testing for excess allele sharing between the 'Ancient Rapanui' and different ancient and present-day Native American populations compared to Polynesians in the SNP array panel. In brief, we observed statistically significant results indicating that the 'Ancient Rapanui' share more alleles with Native Americans than other Polynesians do, with the exception of present-day Rapanui. In that case we could not reject that ancient and present-day Rapanui are symmetrically related to all tested ancient and present-day Native American populations. Notably, we obtained a similar geographic distribution of the D values regardless of the Polynesian population we used as a contrast. Although this observation is expected (see *ADMIXTURE* analyses in Supplementary Section S13), it supports our result indicating that the Native American component in the 'Ancient Rapanui' is most closely related to ancient and present-day Indigenous populations from eastern South America.

We caution that due to the small SNP overlap between the '1240k' SNP capture set and the Affymetrix Human SNP Array 6.0, the resolution of these new tests is substantially lower compared to that of the f_3 -statistics restricted to the 'Ancient Rapanui' Native American component or the D -statistics using the 'Ancient Polynesian' individuals as a contrast population. Furthermore, the interpretation of D -statistics can also be complicated as they capture the specific evolutionary history of the Polynesian population in each case. Therefore, we opted for keeping the f_3 -statistics results in Figure 3 in the main text and the remaining D -statistics results in the Supplementary Information. We have added a description of this analysis and results to Supplementary Section S8.3, together with two figures that are displayed as follows.

"To confirm the results obtained using the 'Ancient Polynesian' individuals as a contrast population, we repeated the D -statistics above using Fijians and the different Polynesian populations in the SNP array dataset (Section S5.1) as a contrast population, i.e., $D(\text{Polynesian}, \text{Ancient Rapanui}; \text{Native American}, \text{Yoruba})$. Results are summarised in Figure S16 and Table S11. For all contrast populations, we obtained statistically significant values of D indicating that the 'Ancient Rapanui' share more alleles with Native Americans than other Polynesians do, with the exception of present-day Rapanui. For the latter we could not reject that ancient and present-day Rapanui are symmetrically related to all tested ancient and present-day Native American populations. Furthermore, when we compared the geographic distribution of these D -statistics, we observed comparable results, regardless of the Polynesian test population (Figure S17, Table S11). These results support that the Native American component in the 'Ancient Rapanui' is most closely related to ancient and

present-day Indigenous populations from western South America (particularly from the Andes), as shown by the f_3 -statistics restricted to the 'Ancient Rapanui' Native American component or the D-statistics using the 'Ancient Polynesian' individuals as a contrast population. However, we caution that due to the small SNP overlap between the '1240k' SNP capture set and the Affymetrix Human SNP Array 6.0 (~200,000 sites), the resolution of these new tests is substantially lower compared to that of the f_3 - and D-statistics tests described above. Moreover, we note that the interpretation of these D-statistics can be complicated as they are expected to capture the specific population history of different Polynesians, e.g., D-statistics involving Fijians are larger due to excess Papuan ancestry compared to 'Ancient Rapanui'. "

Significance ○ |Z|<3.3 ● |Z|>3.3

Figure S16. D-statistics measuring symmetry between 'Ancient Rapanui' and other Polynesians with respect to ancient and present-day Native Americans. We computed $D(\text{Fiji/Polynesian}, \text{AncientRapanui}; \text{Native American}, \text{Yoruba})$ for all Polynesian populations in the SNP array dataset (Section S5.1) and all ancient and present-day populations in the Native American reference panel (Section S5.4). Points represent D statistics, and colours (red and grey) represent their statistical significance for $D \neq 0$ ($|Z| > 3.3$ represents a p -value of ~ 0.001). Grey dashed lines show $D=0$. Under the x -axis, we indicate the pair of populations with excess allele sharing, depending on the sign of D . Present-day Native American populations are sorted according to their language family following ¹⁸ and ancient Native American populations are sorted according to their sampling location and age: 'NAme': North America, 'EoAndes': East of Andes, 'SCone': Southern Cone, >8k: ancient data pre-dating 8,000 years ago, 8-4k: ancient data ranging between 8,000 and 4,000 years ago, <4k: ancient data post-dating 4,000 years ago. Raw results are reported in Table S11."

"Figure S17. Geographic distributions of the D-statistics presented in Figure S16. For each Polynesian test population, we show the results for all tests based on $\geq 5,000$ ABBA+BABA sites. Lighter colours represent greater shared drift between a Native American population and the 'Ancient Rapanui'. In each panel, we label the five Native American populations that lead to the largest D-statistics. Point shapes represent the age of Native American populations in years before present. Raw results are reported in **Table S11.**"

There is a method to estimate ROH that allows pseudo-haploid data (hapROH; Ringbauer et al., 2021) and may be more appropriate in this context than the version in PLINK. Authors should implement this analysis in both, unimputed and imputed versions of their dataset.

Following the referee's suggestion, we ran *hapROH* using pseudohaploid and diploid data for the 15 'Ancient Rapanui', the two 'Ancient Polynesians' and a number of present-day genomes. For the ancient genomes, we used imputed data to run *hapROH* in diploid mode. We further compared these estimates with the PLINK-inferred ROH. While we observed some differences in ROH length bins, the estimated total fraction of the genome contained in ROH ($ROH \geq 4cM$) is similar across the three different ROH sets for most of the individuals. We highlight the similarity between *hapROH*-inferred ROH using pseudohaploid and imputed for the ancient genomes. This result is an additional line of evidence supporting the robustness of imputation as previously shown in ¹⁵. Furthermore, we note two advantages of using the standard PLINK inference together with the imputed genotypes. First, this setup allows us to infer ROH smaller than 4 cM, providing more detailed insights into the genomic history of the individuals. Second, it makes it possible to use any sets of SNPs and not only the '1240K' sites. We show this analysis' results in **Figure S22** and discuss them as follows in Supplementary Section **S11.3**:

"We used *hapROH*⁸², a tool that can detect ROH longer than 4 cM in both pseudohaploid and diploid data. We compared the inferred ROH with the $ROH \geq 4cM$ we previously inferred using PLINK. We ran *hapROH* in both pseudohaploid and imputed modes for the 15 'Ancient Rapanui' and the two 'Ancient Polynesians' and a subset of the present-day genomes in **Figure 2a**. The pseudohaploid and diploid ROH were highly consistent (Pearson correlation of 0.978 between total ROH length) and they mostly differed in their length bins, with the pseudohaploid set containing comparatively longer ROH (**Figure S22, Table S17-19**). We found larger differences when comparing *hapROH*- and PLINK-inferred ROH, particularly in the case of the Paiteer Suruí genomes, for which the total ROH lengths differed by at least 100 cM owing to fewer long ROH ($ROH \geq 20$ cM) being detected using PLINK. In the case of the ancient genomes, none had ROH longer than 20 cM when using PLINK, while *hapROH* detected $ROH \geq 20$ cM for Bot17, RN06, RN11 and RN14. Despite these differences, the total proportion of the genome contained in ROH remained similar across the three sets of inferred ROH, where the two most similar ROH sets were the two ROH

sets detected with hapROH, followed by the two diploid sets found with hapROH and PLINK, respectively (Figure S22d).

While it is challenging to determine which of the three ROH sets is the most accurate, the observed differences do not affect our conclusions regarding inbreeding in the ancient Rapanui, as confirmed by the estimated low inbreeding coefficients (Section S10) and small effective population sizes (Section S12). Furthermore, the observed ROH distribution strongly resembles the expected for a small population according to Ringbauer et al., 2021⁸². If we consider the hapROH estimates for pseudohaploid data, the Paither Suruí, who are known to have endogamic and consanguineous practices⁸⁴, have 231-242 cM of their genome in $ROH \geq 20$ cM, while only three 'Ancient Rapanui' have a single ROH longer than 20 cM (20-23 cM). Moreover, the 'Ancient Rapanui' ROH distribution is largely homogeneous across the 15 individuals, which is likely the result of a shared population history. "

We highlight these results in the main text in line 267:

"We note that we obtain qualitatively compatible results when we conduct these analyses using the imputed genotypes and pseudohaploid calls (S11). "

"Figure S22. HapROH-inferred runs of homozygosity (ROH) estimates for worldwide and 'Ancient Rapanui' genomes. Total lengths of ROH categorised by segment size for 11 present-day genomes, the two 'Ancient Polynesians' and the 15 ancient Rapanui,

estimated with a. hapROH and pseudohaploid genomes, b. hapROH and diploid genomes (imputed in the case of the ancient genomes), c. PLINK and diploid genomes. d. Comparing total ROH length estimated with PLINK and hapROH, and pseudohaploid and diploid data. We report Pearson correlation, r , and corresponding p -value. Aggregate results are reported in Tables S16-19."

The authors failed to discuss some inconsistencies between previous publications, particularly the time and source of Native American admixture. The admixture identified in Ioannidis et al., 2020 is described as a single event in eastern Polynesia dated to around 1,200 CE, before the peopling of Rapa Nui. Furthermore, this is associated with populations in present-day Colombia. Can the authors elaborate on the differences between studies and their implications? Is their study suggesting an additional admixture event to the one suggested by Ioannidis et al., 2020?

We agree with the referee that these are interesting points to discuss further. Below, we address them separately.

Source of Native American admixture

For our analyses aimed at identifying the population that is most closely related to the Native American component in the 'Ancient Rapanui', we compiled a reference dataset that prioritises whole-genome and genome-wide data from individuals pre-dating European contact (Supplementary Section S5.4). Although extensive exchange between Native American populations is well documented throughout the Holocene^{19–21}, using ancient genomic data allows us to exclude genetic signatures stemming from colonial practices (that still occur today) that could obscure pre-European patterns of allele sharing.

Unfortunately, the genomic record in South America (particularly in Ecuador and Colombia, where Ioannidis et al.²² find a good proxy for the Native American ancestry in Polynesians) and Central America is still sparse. Therefore, we conducted an additional test using f_3 -statistics (similar to Figure 3c) where we extended the Native American reference dataset with SNP array data from present-day individuals from the Pacific Coast, Amazonia and Andes²³. After excluding individuals carrying European or African admixture, we still observe that the Andean populations share the most drift with the Native American ancestry tracts in the 'Ancient Rapanui'. We caution that, similar to the Polynesian D -statistics we described above, the resolution of this test is lower as it is restricted to the sites present in the Human Origins SNP array.

Although our results consistently indicate that Andean populations share the most drift with the Native American component in 'Ancient Rapanui', we present this contention as an active research avenue that will likely be resolved as more ancient genomic data from Polynesia and South America become available. Line 490 now reads:

"That we infer the Native American component in 'Ancient Rapanui' to be most closely related to Pacific Coast South Americans and not North Americans or populations east of the Andes further substantiates trans-Pacific contacts between Polynesians and Native Americans. We note that the genomic record in South America (particularly in Ecuador and Colombia) and Central America—where feasible present-day proxies for the Native American component in Polynesians have been identified ²³—is still sparse. Thus, a better proxy for the Native American population that interacted with the ancestors could be identified as more pre-European colonisation genomic data from these regions and other Polynesian islands become available. "

We show this analysis' results in Figure S18 and present them as follows in Section S8.5:

"Although these results strongly support that Andean populations share the most drift with the Native American component in the 'Ancient Rapanui', we conducted an additional test where we extended the Native American reference dataset (Section 5.4) with present-day SNP array data from ⁷⁴. These data include individuals from the Andes, Amazonia and the Pacific Coast of South America, where present-day proxies for the Native American component in Polynesians have been identified ⁶⁵. To identify the individuals from ⁷⁴ that carry European or African admixture, we first computed the following eight D-statistics for each individual (Table S12):

- Four D-statistics to identify individuals with European admixture

$D(\text{SpiritCave/USR1}, \text{present-day individual}; \text{FrenchSGDP/SpanishSGDP}, \text{Chimp})$

- Four D-statistics to identify individuals with African admixture

$D(\text{SpiritCave/USR1}, \text{present-day individual}; \text{YorubaSGDP/MbutiSGDP}, \text{Chimp})$

In the absence of European and African admixture, we expect $D \approx 0$ for all tests. Thus, we excluded all individuals for which at least one of the D-statistics yielded $|Z| > 2.57$ (corresponding to a p-value ~ 0.01). Using the filtered dataset, we computed outgroup f_3 -statistics of the form $f_3(\text{Yoruba}; \text{Native American}, \text{Native American tracts in ancient Rapanui})$ as detailed above. In agreement with other tests presented in this section, we observe that the Andean populations share the most drift with the Native American ancestry tracts in the 'Ancient Rapanui' (Figure S18, Table S13). We caution that genomic data from present-day individuals may carry genetic signatures stemming from colonial practices (that still occur today) that could obscure pre-European patterns of allele sharing. Thus, we anticipate that a better proxy for the Native American population that interacted with the ancestors could be identified as more pre-European colonisation genomic data from Polynesia and South America become available. "

Figure S18. f_3 -statistics exploring the genetic affinities between the 'Ancient Rapanui' and ancient and present-day Native Americans. We extended the Native American reference dataset (Section S5.4) with present-day individuals from the Andes, Amazonia and the Pacific Coast (points with black outline). Using this extended dataset, we computed outgroup f_3 -statistics of the form $f_3(\text{Yoruba}; \text{Native American}, \text{Native American tracts in ancient Rapanui})$. For each population, the point represents the point estimate for f_3 , and error bars correspond to the 95% confidence interval. Point shapes and colours represent sampling locations and age: 'NAmE': North America, 'EoAndes': East of Andes, 'SCone': Southern Cone, >8k: ancient data pre-dating 8,000 years ago, 8-4k: ancient data ranging between 8,000 and 4,000 years ago, <4k: ancient data post-dating 4,000 years ago. Raw results are reported in Table S13. "

Native American admixture date

Using tracts on the pooled 15 'Ancient Rapanui' individuals, we estimate Native American admixture occurred 15-17 generations before their average date of birth. Conservatively assuming 1850 as the average date of birth and 29 years/generation, this interval corresponds to 1357-1415 CE. Furthermore, using the new Bayesian approach integrating genetic and radiocarbon dates, we estimate admixture occurred 1388-1469 CE. These estimates are largely compatible with ²², where Rapanui-Native American admixture was dated to 1380 CE.

Ioannidis et al. find that these estimates are ~100-200 years more recent than those for other Polynesians and attribute this difference to post-contact Chilean related admixture. However, based on the radiocarbon ages of the 'Ancient Rapanui' individuals sequenced in this study, the uniform distribution of Native American admixture proportions and ancestry tracts and the absence of European admixture in the 'Ancient Rapanui', we consider this scenario less likely. Furthermore, the Native American tract length distribution in the 'Ancient Rapanui' (Figure S34) and our analysis of the population size trajectory slopes (Section S23) indicate a scenario akin to a single pulse of migration. These results suggest at least two different admixture events for the Rapanui and other Polynesian islanders. However, in the main text, we present it as a topic that requires more data, e.g., pre-European contact ancient genomes from other islands, to be fully resolved.

Line 501 now reads:

"Although our findings strongly support pre-European trans-Pacific contacts, it remains challenging to establish the number and directionality of the trips that mediated them using genomic data. Our admixture date estimates are compatible single-pulse estimates reported for present-day Rapanui ^{22,23}. Yet, that they postdate the estimates obtained for other present-day Polynesians by ~100-200 years ²³ suggests Transpacific contact occurred more than once. Notably, archaeological evidence and oral history ^{77,78} attest that Polynesian

peoples held the technology and know-how to embark on round trips to the Americas before Europeans reached South America. Thus, we anticipate that additional pre-European ancient genomic data from other Polynesian islands will allow a more nuanced reconstruction of this process. "

References

1. Hunt, T. L. & Lipo, C. P. *The statues that walked: unraveling the mystery of Easter Island*. (Free Press, 2011).
2. Maude, H. E. *Slavers in paradise: the Peruvian slave trade in Polynesia, 1862-1864*. (Stanford University Press, 1981).
3. DiNapoli, R. J., Crema, E. R., Lipo, C. P., Rieth, T. M. & Hunt, T. L. Approximate Bayesian Computation of radiocarbon and paleoenvironmental record shows population resilience on Rapa Nui (Easter Island). *Nat Commun* **12**, 3939 (2021).
4. DiNapoli, R. J., Lipo, C. P. & Hunt, T. L. Revisiting Warfare, Monument Destruction, and the 'Huri Moai' Phase in Rapa Nui (Easter Island) Culture History. *Journal of Pacific Archaeology* **12**, 1–24 (2020).
5. Mieth, A., Kühlem, A., Vogt, B. & Bork, H.-R. Environmental Change and Cultural Continuity: Extraordinary Achievements of the Rapanui Society after Deforestation. in *The Prehistory of Rapa Nui (Easter Island)* (eds. Rull, V. & Stevenson, C.) vol. 22 483–520 (Springer International Publishing, 2022).
6. Liang, M. & Nielsen, R. *Understanding Admixture Fractions*. (2014).
7. Moreno-Mayar, J. V. *et al.* Genome-wide Ancestry Patterns in Rapanui Suggest Pre-European Admixture with Native Americans. *Current Biology* **24**, 2518–2525 (2014).
8. Thomas, N. *Islanders: the Pacific in the age of empire*. (Yale University Press, 2010).
9. Fehren-Schmitz, L. *et al.* Genetic Ancestry of Rapanui before and after European Contact. *Current Biology* **27**, 3209-3215.e6 (2017).
10. Moreno-Mayar, J. V. *et al.* Terminal Pleistocene Alaskan genome reveals first founding population of Native Americans. *Nature* **553**, 203–207 (2018).

11. Sikora, M. *et al.* The population history of northeastern Siberia since the Pleistocene. *Nature* (2019) doi:10.1038/s41586-019-1279-z.
12. Malaspinas, A.-S. *et al.* A genomic history of Aboriginal Australia. *Nature* (2016) doi:10.1038/nature18299.
13. Browning, B. L. & Browning, S. R. Detecting Identity by Descent and Estimating Genotype Error Rates in Sequence Data. *The American Journal of Human Genetics* **93**, 840–851 (2013).
14. The International HapMap 3 Consortium. Integrating common and rare genetic variation in diverse human populations. *Nature* **467**, 52–58 (2010).
15. Sousa Da Mota, B. *et al.* Imputation of ancient human genomes. *Nat Commun* **14**, 3660 (2023).
16. Thorsby, E. The Polynesian gene pool: an early contribution by Amerindians to Easter Island. *Philosophical Transactions of the Royal Society B: Biological Sciences* **367**, 812–819 (2012).
17. Malaspinas, A.-S. *et al.* Two ancient human genomes reveal Polynesian ancestry among the indigenous Botocudos of Brazil. *Current Biology* **24**, R1035–R1037 (2014).
18. Reich, D. *et al.* Reconstructing Native American population history. *Nature* **488**, 370–374 (2012).
19. Moreno-Mayar, J. V. *et al.* Early human dispersals within the Americas. *Science* **362**, eaav2621 (2018).
20. Posth, C. *et al.* Reconstructing the Deep Population History of Central and South America. *Cell* (2018) doi:10.1016/j.cell.2018.10.027.
21. Nakatsuka, N. *et al.* A Paleogenomic Reconstruction of the Deep Population History of the Andes. *Cell* **181**, 1131-1145.e21 (2020).

22. Ioannidis, A. G. *et al.* Native American gene flow into Polynesia predating Easter Island settlement. *Nature* **583**, 572–577 (2020).
23. Barbieri, C. *et al.* The Current Genomic Landscape of Western South America: Andes, Amazonia, and Pacific Coast. *Molecular Biology and Evolution* **36**, 2698–2713 (2019).

Reviewer Reports on the First Revision:

Referee #1 (Remarks to the Author):

I have no further comments, and accept the reply from the authors as satisfactory.

Referee #2 (Remarks to the Author):

The revision looks good, and I appreciate the data tables the authors have added in response to my request. I played around with some of the data, and now feel more confident about the results. I do have some minor comments:

1.) A general thing about this manuscript: I think the format differs so drastically from the one that presumably would eventually be published in Nature, that I am a bit worried that extensive changes might change some of the language to an extent that I would not anymore feel comfortable as a reviewer, given that I have seen only this version. For example, the abstract is clearly way too long, and nearly all the Figure captions are super extensive. But it matters quite a bit what the cuts will be, and how confident things are going to be expressed. Obviously this should be an editorial decision, but as a reviewer I would be interested to see a more final version and check that the cuts (which I anticipate) do not alter important points.

2.) One thing that I saw in the data, and that I already suspected, is that the analysis of where the Native American component may have come from (Figure 3c) is suggesting a bit more confidence than there really is. The standard errors are so wide that the top 40 populations are essentially all overlapping in F3. What is written in the text is not wrong (line 368) "This f3-statistic was maximised for ancient and present-day populations from the Central Andean Highlands". But I think I would suggest to modify that statement to include a statement of caution about how certain we can really be, regionally. For example, it could be useful to label the points on the map with a thicker circle if they are not statistically different from the top hit (Campayanuk). This could even be done using D-Statistics of the form $D(\text{Yoruba}, \text{Nat Am tracts in Ancient Rapanui}, \text{Campayanuk}, \text{Nat Am X})$, which would then probably show non-significant Z-scores for much of X from South America and perhaps beyond. This would then more accurately express our knowledge about the origin of the Native American component. Note that below I also discuss this in light of a comment by reviewer #4 in lines 490 following.

3.) The sentence starting "Notwithstanding" in the Abstract, Line 41: I am still critical of it, and I don't think I was clear enough in my first comment. The combination of a praise ("achievements") and a critique ("overexploitation") makes this come across a bit condescending, as if we were in a position to evaluate their (presumed historic) behavior, even if the critique is not on the side of the authors. I don't think I see either of the two parts of the sentence problematic by themselves. But combining the two in one sentence, that's what triggers me a bit, as if you want to compare the two things. The praise part ("Notwithstanding their achievements") then sounds a bit like you're throwing them a bone. I don't know, maybe I'm oversensitive here, but I would like to express a warning to the authors that this could result in criticism that distracts from the rest of the paper. I won't insist on this any further, but please consider it as a warning.

4.) I think an uninitiated reader might still be confused about "Rapanui" vs "Rapa Nui". The brief explanation in the introduction ("demonym for the island") is not helping, as a random reader wouldn't know what "demonym" means and it doesn't make it crystal clear that one is name for a place, the other is name for a people. I would try to express this in simple English, and arguably already in the abstract (depending on how much of that will remain, see my point 1 above).

5.) Supplement ToC needs updating. SI 9 is not on page 43.

6.) L280: Half-Sentence ending with "and, " is not complete.

Finally, I was asked to also review the authors' responses to reviewer #4, as this reviewer appears to be unavailable for a second review.

I think the replies are all convincing. Regarding the apparent unavailability of data from Ioannidis 2020 and 2021 (both in Nature), I think that's quite a shame. I would recommend that perhaps this journal could pull some strings that data from the two publications is provided for cross-analysis? In any case, this manuscript is now relatively far in the process, and I would not expect the authors at this point to include a full-scale analysis of the data, even if it became available. I do agree with the authors that I don't think that data is central to the conclusions, but of course a minimal comparative set of analyses with the Ioannidis data would have been nice.

Regarding the reply titled "Source of Native American admixture": See my comment above about the uncertainty around finding a source inside the Americas. In line with my comments above, I would suggest to modify the text section from L 490 (as mentioned in their reply) further to not just include a general caveat "as more [...] data from these regions [...] become available" but also a detailed caution that the error bars of the statistics are perhaps even too broad to home in on the Andes. At least that would be something that should be explored/shown, for example with a D-statistic that I suggest above.

Referee #3 (Remarks to the Author):

Upon revisiting the manuscript titled "Genomic insights into the demographic history of Rapa Nui," I have conducted a thorough examination of the authors' utilization of Bayesian modeling, with a specific focus on their innovative approach. While the authors provide a step-by-step explanation in the Supplementary Information (SI), the complexity of the OxCal model poses challenges for readers unfamiliar with the tool to comprehend the intricacies of each step.

For instance, the statement, "We built a Sequence model using two Phases into which date estimates were placed based on the historical date when the skeletal material was collected," inaccurately describes the process, as this is a 2 Phase model overlapping. Therefore, it is imperative to rectify this description and elucidate the criteria behind this decision.

Given the novelty of the approach, it is essential for the authors to furnish a more detailed and accessible explanation, enabling readers to gain a better understanding of OxCal models. Moreover, the choice of using the "date" command instead of "before()" needs a thorough explanation, clarifying the rationale behind selecting one command over the other when they serve similar functions.

I further scrutinized various models, considering variations in generation time and the inclusion/exclusion of radiocarbon dates, confirming the authors' assertion that the results are consistently identical, thereby enhancing the overall reliability of the findings. However, I recommend the authors provide a more explicit description of the OxCal's "Difference" and "introgression" command, including its purpose and utility within OxCal. Additionally, elucidating the mathematical formula employed to calculate the mean number of generations elapsed since introgression based on per-individual tracts estimates and how this is converted into a numerical estimate of the total elapsed time between the radiocarbon date and the admixture date would enhance clarity.

The sensitivity analysis, which explores variations in generation time estimates and the inclusion/exclusion of certain radiocarbon dates, is worthy and contributes to establishing the robustness of the models to input parameter changes. In reference to the start boundary of the model, they wrote: "The results show that, with the constraints applied, the start boundary of the model is equivalent to 1872—1881 CE (at 68.3%)." In here to which model do you refer? When I run the model the starting boundary of the phase 1877 is 1868—1881 CE (at 68.3%). This discrepancy requires clarification in the SI, as it is crucial for the accurate interpretation and potential reproduction of this novel approach in future studies.

The manuscript's discussion on the temporal gap between introgression date estimates and the first peopling estimate is a notable strength, offering valuable insights into result interpretation. However, quantifying the absence of sensitivity to variations in input parameters with statistical measures or tests would further bolster the manuscript.

Last but not least, I recommend consolidating all radiocarbon-related supplementary information into a single section with two subtitles, rather than dispersing it across S3 and placing S16 at the end of the SI. This restructuring would enhance reader convenience, eliminating the need to navigate back and forth between sections. Additionally, renaming S2 as 'DNA Laboratory Procedures' would help avoid confusion with Radiocarbon Laboratory Procedures.

In conclusion, while the methods employed are appropriate for addressing the research questions, the novelty of the approach necessitates a more comprehensive explanation in the SI to ensure clarity and reproducibility for a wider audience.

The authors present whole-genome sequencing data from 15 ancient individuals from Rapa Nui, one of the most remote islands in the world. The individuals originate from a collection in Paris, without primary Archeological context - but the authors present C14 dates suggesting that they date to the 18th or 19th century. The authors managed to produce relatively high coverage genomes, for the first time from ancient individuals from Rapa Nui. This data allows the authors to produce new evidence regarding two essential and much-debated questions of Rapa Nui population history: 1) Showing ca. 10% admixture with Native American genomes, evidencing pre-European trans-Pacific contact the authors here date to ca. 1400 and 2) Reconstructing population size history to generate new evidence regarding an intensely discussed "demographic collapse" ca. 1600 on Rapa Nui before European contact due to resource overexploitation ('ecocide').

The aDNA data generation and processing is highly competent, e.g., how imputation (a critical step in this work) is validated is at the forefront of the aDNA field. The authors convincingly show that the genomes lack European admixture but are closely related to present-day Rapa Nui - which critically confirms the ancient sample to be authentic and most likely from Rapa Nui indeed.

The article is exceptionally well written, including useful and well-crafted figures. The article seems to cite the literature well (as far as I can tell as an aDNA researcher), however the (dis)agreement with the two Ioannidis et al papers that analyze genomes of present-day Polynesians to address related questions (e.g. admixture with Native Americans and time of settlements in Eastern Polynesia) could be clearer (even after revision).

My core expertise is in ROH/IBD sharing and demographic inference, and I will comment mainly on that part of this work (the other reviewers already commented extensively and competently on the other parts). The authors produced suitable data to address the question of "ecocide" - and also the bottlenecked history of East Polynesian expansions produced ample signal to pioneer such analysis.

As I explain below (A: Critical Comments), I believe that there are some inconsistencies regarding haplotype sharing analysis and using an LD signal to reconstruct population size trajectory. Those are cutting-edge methods beyond the "established" standard in the aDNA field - therefore they warrant extra caution. Any signal critically double-checked - which the authors in some parts already do, in some less so, as I outline below.

The question the authors address here is very consequential and the "ecocide" hypothesis will likely reach a very broad audience. That's why I want to be double-sure - as any "correction" later on would be damaging to the credibility of the aDNA field and its ability to reconstruct population sizes. I hope the authors understand that before I can sign off on this part of the work the below inconsistencies have to be resolved or at least explicitly discussed.

The authors have already done an impressive range of cutting-edge bioinformatic analysis well beyond any typical aDNA paper. That's why I am confident that they can address my comments without much delay. In case extra questions arise, I am also happy to assist in any way the authors would find useful.

A Critical Comments:

ROH (within individuals) and IBD segments (between individuals) feature centrally in the analysis; and are shown in Fig. 1 & 2. Both ROH and IBD calling are highly sensitive to genotype errors (e.g. even small rates of false heterozygotes or false opposing homzygotes, respectively, start breaking inferred segments up). Regarding ROH, the authors observe this themselves in their downsampling experiments.

For ROH, a recent advance specifically detects ROH in low-coverage aDNA (hapROH), and is designed to have as little false positive rate and length bias as possible. After request of one other reviewer, the authors compared results of their „custom“ pipeline (using Plink and imputed data) to hapROH results. That analysis revealed some notable differences, in particular for long ROH:

The authors speak about „qualitatively similar results“. This is only true for the two conclusions of no close parental relatedness (offspring of most first cousins and even many second cousins would have mostly >50cm their genome in ROH>20cm, see Ringbauer 2021); and overall amount of ROH.

But the length distribution is markedly different! E.g. all 15 ancient Rapa Nui have ROH

>12cm in pseudo-haploid hapROH, while only 5/15 do for PLINK. Importantly, these longer ROH are exactly where one would could read off signals of a recent bottleneck!

The authors prominently show the PLINK ROH length distribution in the main text (in Fig. 2), write about difference in long ROH to present-day Rapa Nui, and explicitly state that "However, none of the 15 individuals carry excessively long ROH ($\geq 16\text{cM}$)" which strongly disagrees with hapROH. Therefore, the true ROH distribution would be very important. But the authors write in the SI that "While it is challenging to determine which of the three ROH sets is the most accurate...". Here I disagree.

I believe that in the high coverage genomes that long ROH should be very „obvious“.

One could simply zoom into the genetic data. E.g. RNI014, where hapROH infers one ROH>20cm, one could zoom into this region using hapROH’s functionality (see Fig. S7 from hapROH SI):

For RNI014, a 25x genome (!) it should become highly evident where there is long ROH, and whether Plink ROH has an erroneous gap (because it does not merge gaps caused by low SNP density or sporadic erroneous SNP calls). On the other hand, hapROH might too aggressively merge true „gaps“ in that low Ne scenario, but that should become obvious too.

The „manual“ resolution of this discrepancy between hapROH and Plink should be discussed. If this Zoom in reveals that hapROH is more accurate – then it should be put in the main figures. The length distribution of long ROH matters, and one should show the „accurate“ one.

A2) A major inconsistency is that ROH and IBD sharing should be closely corresponding but as I outline now does not. ROH is IBD within the two haplotypes of an individual. While a pair of diploid individuals has four possible haplotype combinations, ROH only one possible combination. Therefore, total IBD sharing per pair should be 4x as much as ROH per individual.

However, Fig. 1 shows that the ancient Rapa Nui have ca. 1200cm IBD>5cm per pair, but Fig. 2 shows that they only have ~150 cm ROH >4cm. That is far off from the expected factor of 4x, considering that >5cm cutoff is even higher than >4cm.

There are several explanations for this discrepancy. Most importantly that IBD calling has high rates of false positives (FP), „over-calling“ in length, or perhaps also that very short IBD overlap and „merged“ which will be especially strong since the high amount of overall IBD sharing; see Chiang et al). For modern DNA, IBD calling around 5-6cm gets very tricky with high FP. Such High FP are especially a concern here as the imputed ancient data likely has different FP rates than using the modern DNA, which can cause batch effects.

I note that the authors could use ancIBD (Ringbauer et al 2023) to double-check for longer IBD, as it is well calibrated to avoid FP >8cm (staying on purpose away for the „troublesome“ shorter IBD).

As this is a major discrepancy between two main figures (Fig. 1 and Fig. 2); it has to be explicitly discussed.

A3) In Fig. 2 only ROH of one present-day Rapa Nui is shown. There are more samples available – why was this one selected? And, one individual can be an outlier, so it would be much more informative showing ROH of multiple IIDs. The same holds for the other modern populations; but modern Rapa Nui are a especially critical omission.

A4) For the population size reconstruction, the authors use an LD signal - which is implicitly uses the IBD/ROH signal.

I would ask the authors to explain why they did not even attempt to use IBD_Ne on called IBD signals. I suspect the authors did not trust the IBD calls enough (in light also of the above inconsistencies reasonable!). However, as IBD_Ne would be the established and “obvious” way to go about effective population size reconstruction, the authors should justify why they did not do it.

IBD_ne is cited over 250 times, and widely used in present-day genomes ([10.1016/j.ajhg.2015.07.012](https://doi.org/10.1016/j.ajhg.2015.07.012)), while hapNe_LD is cited four times (because it is so new). If IBD calls would be perfect - IBD_Ne would arguably be the better method (see next point).

A5) hapNe_LD uses an LD signal - however, the authors also use an LD signal to date admixture. That seems slightly odd a first glance - one uses a very similar signal to infer two different processes while ignoring the other.

As bottlenecked populations that likely kept admixing (continued East Polynesians contacts) and admixture with a continentally different ancestry (Native American) also produce strong LD - it is a priori not clear whether this influences the LD that hapNe_LD uses. The authors should comment on this, at least heuristically.

A6) A major worry of mine is that the bottleneck detection did not reveal a recent bottleneck in present-day data – despite the major collapse to 110 individuals in the 19th century after the slave raids and epidemics, of which only 1/3 reproduced (and all present-day Rapa Nui claim as ancestors).

Yet, the authors do not detect any recent crash using hapNe_LD in the modern data:

Figure S23. Recent effective population size

Moreover, the ROH sharing of ancient and present-day Rapa Nui seems comparable – and the present-day individuals have less IBD sharing among themselves than then ancients.

The authors should discuss this more explicitly, and why they think modest amounts of recent-admixture masked this signal. Was it too short to have an effect? Which raises the question, would we detect a short ancient collapse?

It generally seems that hapNe_LD is very bad in detecting short sudden crashes, see e.g. main figure 2c:

The software seems to „oversmooth“ likely due to regularization. That’s why the authors do (impressive) additional simulations to match hapNe_LD curves. But that also is a bit of an odd approach, matching hapNe_LD inferred summary curves but not the actually underlying statistic (LD) – and I have a very bad intuition on how robust and sensitive that approach actually is.

A7) In light of the above points, I am worried that putting a strong inference on hapNe_LD curves alone might miss some crucial signal for population size reconstructions.

As I believe that hapROH ROH curves 4-20cm should be very accurate (and the authors hopefully can double-check that), and the ROH curve based on 15 individuals seems very robustly inferrable from 15 IIDs in such low Ne scenarios. Therefore, I would suggest one final „double-check“:

Namely, simulating ROH of several individuals under the best-fit model, and comparing ROH to the empirically inferred ROH in the ancient Rapa Nui. If that looks like a good match, I would be convinced about the „steadily-increasing“ scenario.

I note that the authors already have their simulations set up – and it is rather easy to extract ROH from msprime output.

Major Comments:

B1) Surprisingly, the authors did not call Y haplogroups! This is a standard analysis in the aDNA field - and should really be done as it could reveal further strong evidence regarding the admixture. Uniparental haplogroups are effectively only one locus - but can be highly diagnostic of continental ancestry, and moreover give first insights into sex-biased processes.

The authors called 15 mtDNA haplogroups, and those are all typical of East Polynesians (the authors should also mention this in the main text). That can happen simply by chance even if one assumes ca. 10% Native American ancestry - but it also raises a curiosity of what the Y haplogroups are.

B2) The authors used standard aDNA methods (based on average pairwise mismatch rates, PMR) to detect up to second-degree relatives, and did not find any. However, IBD can go further – and the authors have such data in their hand. Notably, the SI shows that RN05 and RN12 have ca. 400 cm IBD >35cm (Fig. S19), standing out clearly from the rest of the ancient pairs! That, rather clearly, points towards this pair being relatives ca. 3-4th degree (see Ringbauer et al 2023). This pair already has the lowest lowest PMR in Fig. S20.

This exception also proves the rule that none of the other pairs are related up to the ~4th degree – which further improves the value of this dataset (as the ROH/IBD signal is consistent in a set of non-related individuals, showing how „general“ the signal is!). I believe

that the manuscript would benefit from explicitly stating that.

Minor Comments

C1) The authors run standard contamination estimates, based on mtDNA and X chromosomes in males. Based on this results, I am quite sure that there is generally indeed little contamination.

Notably, the fact that the authors can call long ROH in all ancient individuals indirectly shows lack of substantial contamination (which would „destroy“ ROH). One could apply hapCon_ROH (introduced in Posth et al 2022, now publicly available, see hapROH website) to directly measure that and to obtain robust autosomal contamination estimates. This tool is usually a gimmick, as typical ancient individuals do not have the coverage or the ROH stretches necessary. But here both is given for all 15 ancient individuals!

Author Rebuttals to First Revision:

We would like to thank the referees for providing valuable feedback for our manuscript. Please find below our detailed response to the referees' comments (in black), our replies (in blue), and specific changes to manuscript text (*in italics*). Note that cited lines correspond to line numbers in the newly submitted manuscript. Figures that are only included as a response to the referees' comments but not in the main text or supplementary information are labelled as 'Figure Rx'.

Referee #1 (Remarks to the Author):

I have no further comments, and accept the reply from the authors as satisfactory.

We thank the referee for their overall review.

Referee #2 (Remarks to the Author):

The revision looks good, and I appreciate the data tables the authors have added in response to my request. I played around with some of the data, and now feel more confident about the results. I do have some minor comments:

1.) A general thing about this manuscript: I think the format differs so drastically from the one that presumably would eventually be published in Nature, that I am a bit worried that extensive changes might change some of the language to an extent that I would not anymore feel comfortable as a reviewer, given that I have seen only this version. For example, the abstract is clearly way too long, and nearly all the Figure captions are super extensive. But it matters quite a bit what the cuts will be, and how confident things are going to be expressed. Obviously this should be an editorial decision, but as a reviewer I would be interested to see a more final version and check that the cuts (which I anticipate) do not alter important points.

We thank the referee for this comment. We have now edited the main text to follow closely Nature's formatting guidelines (<https://www.nature.com/nature/for-authors/formatting-guide>). That said, except for the abstract, which used to be really too long as the reviewer points out, our main text and figures (number and dimensions) were actually already quite close to what is expected. Specifically, we only had to cut around 10% of the main text, which was achieved by following more closely the Nature format for references, and by removing some sentences from the introduction regarding the archaeological background and some explicit information about marital practices on the island (which now appears only as a reference). We now believe that if our manuscript is eventually accepted as a research article, the version that we just resubmitted should be very close to the final one in terms of abstract, main text and figures.

2.) One thing that I saw in the data, and that I already suspected, is that the analysis of `_where_` the Native American component may have come from (Figure 3c) is suggesting a bit more confidence than there really is. The standard errors are so wide that the top 40 populations are essentially all overlapping in F3. What is written in the text is not wrong (line 368) "This f_3 -statistic was maximised for ancient and present-day populations from the Central Andean Highlands". But I think I would suggest to modify that statement to include a statement of caution about how certain we can really be, regionally. For example, it could be useful to label the points on the map with a thicker circle if they are not statistically different from the top hit (Campayanuk). This could even be done using D-Statistics of the form $D(\text{Yoruba}, \text{Nat Am tracts in Ancient Rapanui}, \text{Campayanuk}, \text{Nat Am X})$, which would then probably show non-significant Z-scores for much of X from South America and perhaps beyond. This would then more accurately express our knowledge about the origin of the Native American component. Note that below I also discuss this in light of a comment by reviewer #4 in lines 490 following.

We thank the referee for this comment. We fully agree with the referee that the resolution of the available data should be considered for the interpretation of our results and that the uncertainty and standard errors should also be made immediately visible in the figures. To improve our f_3 -statistics representation, we have modified Figure 3c and included a new Extended Data Figure 2. In Figure 3c, we changed the outlines of the symbols to distinguish point estimates that are included within the 95% confidence interval of the maximum f_3 -statistic ($f_3(\text{Yoruba}; \text{Campanayuq}, \text{Native American tracts in Ancient Rapanui})$) from those that are not. Extended Data Figure 2 shows all point estimates in Figure 3c, together with their 95% confidence intervals. We note that this degree of resolution is not unexpected given that the reference dataset (S15.4) is restricted to the 1240K sites and more importantly, Native American populations are related to each other through short internal branches, e.g., (Reich et al. 2012; Moreno-Mayar et al. 2018). This is therefore an intrinsically difficult problem made even harder by the limited availability of genomic data representing Native American genetic diversity.

Hence, for our results interpretation we consider the relative ordering of the f_3 -statistics, where Andean populations yield the largest f_3 values.

Line 347 now reads:

"We note that point estimates for some non-Andean populations overlap with the confidence intervals of the highest f_3 (and D) values, likely due to low resolution of the reference dataset (S15.4) and short internal branches separating Native American groups^{55,56}. However, Andean populations (top 12 f_3 -statistics) consistently yield the highest f -statistics (Extended Data Figure 2)."

Figure 3c and its caption are displayed as follows:

" [...] **c.** f_3 -statistics showing the genetic affinities between the Native American tracts in 'Ancient Rapanui' and ancient and present-day Native American populations. For each 'Ancient Rapanui', we masked Polynesian ancestry tracts and pooled Native American ancestry tracts to estimate f_3 -statistics. Lighter colours represent greater shared drift between a Native American population (X) and the Native American tracts in ancient Rapanui. We label the five Native American populations that lead to the largest f_3 -statistics. Point shapes represent the age of Native American populations in years before present. Points with a darker outline correspond to point estimates overlapping with the 95% confidence interval of the highest f_3 -statistic (illustrated in Extended Data Figure 2). "

Extended Data Figure 2 and its caption are displayed as follows:

Extended Data Figure 2. f_3 -statistics showing the genetic affinities between the Native American tracts in 'Ancient Rapanui' and ancient and present-day Native American populations. This figure shows 95% confidence intervals for the f_3 -statistics reported in **Figure 3c**. The horizontal line indicates point estimates overlapping with the 95% confidence interval of the highest f_3 -statistic. The top 12 f_3 values correspond to ancient and present-day Andean populations. For present-day Native American populations symbols and colours correspond to their language family following⁵⁵ and for ancient Native American populations, they correspond to their sampling location and age. Abbreviations for ancient populations: 'NAme': North America, 'EoAndes': East of Andes, 'SCone': Southern Cone, >8k: ancient data pre-dating 8,000 years ago, 8-4k: ancient data ranging between 8,000 and 4,000 years ago, <4k: ancient data post-dating 4,000 years ago. Raw results are reported in **Table S13**. "

3.) The sentence starting "Notwithstanding" in the Abstract, Line 41: I am still critical of it, and I don't think I was clear enough in my first comment. The combination of a praise ("achievements") and a critique ("overexploitation") makes this come across a bit condescending, as if we were in a position to evaluate their (presumed historic) behavior, even if the critique is not on the side of the authors. I don't think I see either of the two parts of the sentence problematic by themselves. But combining the two in one sentence, that's what triggers me a bit, as if you want to compare the two things. The praise part ("Notwithstanding their achievements") then sounds a bit like you're throwing them a bone. I don't know, maybe I'm oversensitive here, but I would like to express a warning to the authors that this could result in criticism that distracts from the rest of the paper. I won't insist on this any further, but please consider it as a warning.

We followed the referee's suggestion to bring the text closer to the *Nature* format in this round. As part of the editing process, this sentence has been removed from the abstract.

4.) I think an uninitiated reader might still be confused about "Rapanui" vs "Rapa Nui". The brief explanation in the introduction ("demonym for the island") is not helping, as a random reader wouldn't know what "demonym" means and it doesn't make it crystal clear that one is name for a place, the other is name for a people. I would try to express this in simple English, and arguably already in the abstract (depending on how much of that will remain, see my point 1 above).

We have considerably shortened the abstract to comply better with the *Nature* format. Therefore, we would prefer to keep the abstract in its current form (word count: 199). Nevertheless, we include the following clarification in the Introduction, in Line 57:

"The following five centuries saw the Rapanui, the inhabitants of Rapa Nui, develop a culture characterised by iconic giant stone statues (moai) and monumental stone platforms (ahu)^{1,5}."

5.) Supplement ToC needs updating. SI 9 is not on page 43.

We thank the referee for spotting this. We have now updated the table of contents.

6.) L280: Half-Sentence ending with "and, " is not complete.

Line 245 now reads:

"We use the 15 'Ancient Rapanui' genomes to test the ecocide hypothesis using biological data since they postdate the proposed collapse and yet, they are unlikely to be affected by the demographic events that followed the 1860s Peruvian slave raids^{1,7}. "

Finally, I was asked to also review the authors' responses to reviewer #4, as this reviewer appears to be unavailable for a second review.

We thank the referee for looking at these replies as well.

I think the replies are all convincing. Regarding the apparent unavailability of data from Ioannidis 2020 and 2021 (both in Nature), I think that's quite a shame. I would recommend that perhaps this journal could pull some strings that data from the two publications is provided for cross-analysis? In any case, this manuscript is now relatively far in the process, and I would not expect the authors at this point to include a full-scale analysis of the data, even if it became available. I do agree with the authors that I don't think that data is central to the conclusions, but of course a minimal comparative set of analyses with the Ioannidis data would have been nice.

We agree with the referee that co-analysing the Ancient Rapanui genome data with the Ioannidis et al. SNP array dataset would have been interesting. We have been in contact with the editor about this.

Regarding the reply titled "Source of Native American admixture": See my comment above about the uncertainty around finding a source inside the Americas. In line with my comments above, I would suggest to modify the text section from L 490 (as mentioned in their reply) further to not just include a general caveat "as more [...] data from these regions [...] become available" but also a detailed caution that the error bars of the statistics are perhaps even too broad to home in on the Andes. At least that would be something that should be explored/shown, for example with a D-statistic that I suggest above.

Please see our response to referee's comment 2.) above. Following the referee's suggestion, we have modified Figure 3c and added new Extended Data Figure 2 to illustrate

the results' uncertainty. Additionally, we have added a discussion about it and how we interpret the results in the main text (see above).

Referee #3 (Remarks to the Author):

Upon revisiting the manuscript titled "Genomic insights into the demographic history of Rapa Nui," I have conducted a thorough examination of the authors' utilization of Bayesian modeling, with a specific focus on their innovative approach. While the authors provide a step-by-step explanation in the Supplementary Information (SI), the complexity of the OxCal model poses challenges for readers unfamiliar with the tool to comprehend the intricacies of each step.

For instance, the statement, "We built a Sequence model using two Phases into which date estimates were placed based on the historical date when the skeletal material was collected," inaccurately describes the process, as this is a 2 Phase model overlapping. Therefore, it is imperative to rectify this description and elucidate the criteria behind this decision.

We thank the referee for noticing this. It should of course be a two-phase model incorporating two overlapping phases. We have rectified the text accordingly.

Given the novelty of the approach, it is essential for the authors to furnish a more detailed and accessible explanation, enabling readers to gain a better understanding of OxCal models. Moreover, the choice of using the "date" command instead of "before()" needs a thorough explanation, clarifying the rationale behind selecting one command over the other when they serve similar functions.

Thank you for this comment. There is only a very slight difference between `Date` and `Before` commands, as the referee points out. We used the `Date` command rather than `Before` because the constraint we are applying (the fact that the skeletal material was collected by 1877 or 1935 and therefore the ages must pre-date these historic time points) is outside the realm of the phase models themselves; they are external. In looking more into this we decided that, while it was appropriate to use `Date`, it would be more correct to place the command outside the `End Boundary` for each of the two Phases, so this is the approach we have now taken. This results in a more reliable model, despite a decrease in the precision of some of the posterior ranges. On the positive side it also showed that there was less concern regarding the reliability of the BRAMS-4316 date. While it had a slightly lower agreement index it was not a major outlier. For this reason, we did not undertake sensitivity testing in leaving it out of a second model, and just ran three models for each of the three generation time estimates.

We also tested whether there was any difference between using `Before` or `Date` in these models, and we found nothing significant. We have updated the manuscript and the full S116 using these new approaches.

I further scrutinized various models, considering variations in generation time and the inclusion/exclusion of radiocarbon dates, confirming the authors' assertion that the results are consistently identical, thereby enhancing the overall reliability of the findings. However, I recommend the authors provide a more explicit description of the OxCal's "Difference" and "introgression" command, including its purpose and utility within OxCal. Additionally, elucidating the mathematical formula employed to calculate the mean number of generations elapsed since introgression based on per-individual tracts estimates and how this is converted into a numerical estimate of the total elapsed time between the radiocarbon date and the admixture date would enhance clarity.

We thank the referee for testing the models independently.

`Difference` is a calculation determining the difference between two probability distribution functions (PDF); one PDF is subtracted from another. It becomes significantly different if the resulting PDF does not overlap with zero.

"Introgression" in the model is not an OxCal command, it is simply a name given to the age estimates we generated from each date based on the mean number of generations and generation times since South American admixture, which in turn is determined based on the per-individual *tracts* estimates. Each number of generations calculated from each dated sample was multiplied by the three generation times we tested (25, 29 or 30 years/generation) and used to calculate the introgression age (e.g., 17 generations x 30 years = 510 years). We added an estimate of the uncertainty on the generation times at $\pm 10\%$, whilst also estimating that each genetically derived generation time estimate has its own uncertainty, with an upper 95% and lower 95% value. We calculated an overall uncertainty using these two values (e.g., 510 ± 58 yr).

The `Difference` command was then used to determine the time elapsed between the modelled radiocarbon ages and the introgression estimates:

```
Difference("BRAMS-4316Offset","BRAMS-4316","Introgression",N(510,58));
```

The `Date` command was used to calculate an overall admixture PDF from all the results thus:

```
Date("Introgression",U(1000,1900));
```

(The uniform prior applied allows the model to converge faster within a conservative window ranging from 1000 to 1900 CE).

We have modified the text in SI16 to explain this in more detail and hope this is clearer now.

The sensitivity analysis, which explores variations in generation time estimates and the inclusion/exclusion of certain radiocarbon dates, is worthy and contributes to establishing the robustness of the models to input parameter changes. In reference to the start boundary of the model, they wrote: "The results show that, with the constraints applied, the start boundary of the model is equivalent to 1872—1881 CE (at 68.3%)". In here to which model do you refer? When I run the model the starting boundary of the phase 1877 is 1868—1881 CE (at 68.3%). This discrepancy requires clarification in the SI, as it is crucial for the accurate interpretation and potential reproduction of this novel approach in future studies.

Thank you for this comment. We agree this could be made clearer. We have modified the section to discuss the values in more detail and commented on the variability of the start boundaries between models.

Basically, across all models, the start and end boundaries for the two phases give very similar results. Here, for example, are the start boundaries for Phase 1 across all 6 models we previously ran, the same applies for the three models we now include (Figure R1).

Figure R1. Start boundaries for Phase 1 across all 6 models we previously ran.

Of course, it is important to be aware that every run of a Bayesian model will vary very slightly, the question is whether this is significant and therefore sensitive in terms of the repeatability of the model and the relationship between the priors and the likelihoods in the model. In this case we would argue that 1-4 year variations in a boundary estimate are not significant, especially when considering the precision of the actual calibration curve is $\pm 9-10$ years.

The manuscript's discussion on the temporal gap between introgression date estimates and the first peopling estimate is a notable strength, offering valuable insights into result interpretation. However, quantifying the absence of sensitivity to variations in input parameters with statistical measures or tests would further bolster the manuscript.

We have modified the supplementary text to show the variation in the models generated. The results show that for the 29- and 30-year generation times the models show introgression dates fall in the mid 14th century while the 25-year generation time model is slightly later.

Figure S49. Comparison of the posteriors for the Introgression date estimate for each model. The results are robust to variation in the input estimates for generation times because they overlap statistically with one another. "

We tested the differences in these PDFs using Difference, particularly for Model 1.1 (25 yr generation time) and each overlapped with zero, suggesting they are not distinguishable (Figure R2). We updated the text to describe this.

Figure R2. Differences between models 2.1 and 3.1, and model 1.1.

Last but not least, I recommend consolidating all radiocarbon-related supplementary information into a single section with two subtitles, rather than dispersing it across S3 and placing S16 at the end of the SI. This restructuring would enhance reader convenience, eliminating the need to navigate back and forth between sections. Additionally, renaming S2 as 'DNA Laboratory Procedures' would help avoid confusion with Radiocarbon Laboratory Procedures.

We thank the referee for this comment. We feel strongly it is better to use separate sections, since the supplement is ordered to follow the main text closely. More specifically, the second part (S116) depends on the genetic dates which are discussed after the initial radiocarbon section, so we would like to keep them apart rather than merge them into a single section.

In conclusion, while the methods employed are appropriate for addressing the research questions, the novelty of the approach necessitates a more comprehensive explanation in the SI to ensure clarity and reproducibility for a wider audience.

Referee #5 (Remarks to the Author):

The full review includes some figures for clarification; these can be found in the attached PDF version.

Review

The authors present whole-genome sequencing data from 15 ancient individuals from Rapa Nui, one of the most remote islands in the world. The individuals originate from a collection in Paris, without primary Archeological context - but the authors present C14 dates suggesting that they date to the 18th or 19th century. The authors managed to produce relatively high coverage genomes, for the first time from ancient individuals from Rapa Nui. This data allows the authors to produce new evidence regarding two essential and much-debated questions of Rapa Nui population history: 1) Showing ca. 10% admixture with Native American genomes, evidencing pre-European trans-Pacific contact the authors here date to ca. 1400 and 2) Reconstructing population size history to generate new evidence regarding an intensely discussed “demographic collapse” ca. 1600 on Rapa Nui before European contact due to resource overexploitation ('ecocide').

The aDNA data generation and processing is highly competent, e.g., how imputation (a critical step in this work) is validated is at the forefront of the aDNA field. The authors convincingly show that the genomes lack European admixture but are closely related to present-day Rapa Nui - which critically confirms the ancient sample to be authentic and most likely from Rapa Nui indeed.

The article is exceptionally well-written, including valuable and well-crafted figures. The article seems to cite the literature well (as far as I can tell as an aDNA researcher); however, the (dis)agreement with the two Ioannidis et al. papers that analyze genomes of present-day Polynesians to address related questions (e.g., admixture with Native Americans and time of settlements in Eastern Polynesia) could be more precise (even after revision).

In this manuscript, we discuss our results based on previous work on genome-wide data from ancient (Fehren-Schmitz et al. 2017) and present-day (Moreno-Mayar et al. 2014;

Ioannidis et al. 2020; Ioannidis et al. 2021) Rapanui individuals. In summary, our results differ substantially from the only study to date which includes ancient data from Rapa Nui but is—in our view—very much in line with previous work relying on present-day genome-wide data from the island (Moreno-Mayar et al. 2014; Ioannidis et al. 2020).

More specifically, our results do not support the conclusion in (Fehren-Schmitz et al. 2017), that Native American ancestry was introduced in the island following European contact. Instead, using the 15 'Ancient Rapanui' genomes in this study—which do not bear European admixture—we find that the Rapanui carry ~10% Native American ancestry and we confidently estimate this admixture event pre-dates European contact in 1722. We present this debate in the abstract in and our results in Line 37 and Line 45, respectively:

"Second, the possibility of trans-Pacific voyages to the Americas pre-dating European contact is still debated. "

"Furthermore, the ancient and present-day Rapanui carry similar proportions of Native American admixture (~10%). Using a novel Bayesian approach to integrate genetic and radiocarbon dates, we estimate this admixture event occurred ~1250-1430CE. "

And elaborate on the current evidence in Line 100:

"However, surprisingly, the only two ancient DNA studies of ancient Rapanui to date did not find evidence for Native American admixture^{24,25}. The first study focused on mitochondrial DNA from 12 individuals, whereas the second analysed low-depth whole-genome data from five individuals dating before and after European contact. In the latter, despite their low depth of coverage (0.0004-0.0041X), downstream population genetic analyses confirmed that the five ancient individuals were Polynesian. However, even though the analysed human remains were post-dating the inferred Native American admixture time, no Native American ancestry was reported in these ancient genomes, casting doubt on the findings based on data from present-day populations. "

Furthermore, we reanalyse the data from (Fehren-Schmitz et al. 2017) and detail our results in Line 323:

"To reconcile these results with the previous study where no evidence of Native American ancestry was found in ancient Rapanui individuals, we reanalysed the publicly available data from²⁵. In agreement with²⁵, we did not obtain statistically significant D-statistics suggesting Native American admixture. However, through a downsampling experiment, we show that low depth can explain the absence of statistically significant D-statistics (FigureS14, SI8). Furthermore, using our ADMIXTURE approach, we detected a Native American-like component in three out of five previously published ancient Rapanui represented by low-depth sequencing data (FigureS36, SI5.6, 13). "

And in Line 438:

"Although it is well established that the Polynesian Rapanui ancestors arrived on the island during a period of rapid and purposeful exploration voyages across thousands of kilometres of sea³, the possibility of subsequent round trips to the Americas remains controversial. In particular, the only other ancient whole-genome Rapanui study to date did not find evidence for Rapanui-Native American contact. Through downsampling experiments of our data, we find that the average depth of coverage of that dataset (0.0004-0.0041X) does not yield enough power to carry out the specific statistical tests using D-statistics we applied here (FigureS14, S18). Moreover, by running per-individual ADMIXTURE analyses aimed at minimising the effect of strong genetic drift in Remote Oceanians⁷⁰, we estimated that three out of the five previously published individuals carry >4.7% of a Native American-like ancestry component. "

We also compare our conclusions to more recent work based on genome-wide data from present-day Rapanui and other Polynesian islanders. In this case, while the discussion and interpretation of the results differ, we find a strong concordance in terms of quantitative results. Our analyses and results support that the Rapanui-Native American contact occurred before European contact as shown in (Moreno-Mayar et al. 2014). In (Ioannidis et al. 2020), the authors detected Native American gene flow in present-day Polynesians from specific islands. In particular, they dated the admixture events in Mangareva to 1230CE, in Palliser to 1230CE, in North Marquesas to 1200CE and in South Marquesas to 1150CE. For the present-day Rapanui with little European ancestry, they estimated a later time at 1380CE. The authors attribute the more recent Rapanui-Native American admixture date to more recent gene flow events. In our study, we estimated that the Rapanui-Native American contact took place between ~1250CE and ~1430CE. These estimates are in agreement with those in (Ioannidis et al. 2020). One of the main differences from a quantitative point of view is that it is unclear what the confidence intervals are in (Ioannidis et al. 2020). In any case, our inferred lower bound is very close to the 1150CE-1230CE, the earlier admixture time inferred for other islands. Thus, a single admixture event is feasible for all the analysed populations. However, it is also possible that there were more than one Native American admixture events in Polynesia, and the later dates from (Moreno-Mayar et al. 2014; Ioannidis et al. 2020), as well as our upper bound could reflect that. If we consider the Rapa Nui peopling time as in (Ioannidis et al. 2021), *i.e.*, 1210CE, and our range of dates for Native American admixture, it seems likely that admixture happened after peopling, which supports the scenario of multiple pulses of Native American admixture in Polynesia.

We present our admixture dating results in the context of the peopling of Rapa Nui in updated Line 387:

"Assuming 29 years per generation our favoured model gave a range of ~1336-1402CE (at 68.3% prob.) and ~1246-1425CE (95.4%) (Figure 4d, SI16). These estimates overlap significantly with the most recent estimates for the peopling of Rapa Nui (1150-1280CE¹¹) and strongly suggest the admixture event did not pre-date the peopling. Moreover, they show that the Polynesian ancestors of Rapanui were in contact with Native Americans—significantly—before the first appearance of Europeans on the island (364±41 years before 1722)."

Furthermore, we discuss these results in updated Line 459:

"Although our findings strongly support pre-European trans-Pacific contacts, it remains challenging to establish the number and directionality of the trips that mediated them using genomic data. Our admixture date estimates are compatible with single-pulse estimates reported for present-day Rapanui^{22,23}. Importantly, they overlap with the most recent estimates for the peopling of the island¹¹, suggesting that a brief time may separate the admixture event and the peopling of Rapa Nui. Yet, that they postdate the estimates obtained for other present-day Polynesians by ~100-200 years²³—and are concordant across ancient and present-day Rapanui individuals^{22,23}—suggests Transpacific contact could have occurred more than once. Notably, archaeological evidence and oral history^{71,72} attest that Polynesian peoples held the technology and know-how to embark on round trips to the Americas before Europeans reached South America. Thus, we anticipate that additional pre-European ancient genomic data from other Polynesian islands will allow a more nuanced reconstruction of this process."

We feel we could of course further expand the discussion but – besides not having reached the word limit - whatever we could add would be even more speculative.

My core expertise is in ROH/IBD sharing and demographic inference, and I will comment mainly on that part of this work (the other reviewers have already commented extensively and competently on the other parts). The authors produced suitable data to address the question of “ecocide” - and also the bottlenecked history of East Polynesian expansions produced ample signal to pioneer such analysis.

As I explain below (A: Critical Comments), I believe that there are some inconsistencies regarding haplotype sharing analysis and using an LD signal to reconstruct population size trajectory. Those are cutting-edge methods beyond the “established” standard in the aDNA field - therefore, they warrant extra caution. Any signal critically double-checked - which the authors in some parts already do, in some less so, as I outline below.

The question the authors address here is very consequential, and the “ecocide” hypothesis will likely reach a broad audience. That’s why I want to be double-sure - as any “correction” later on would be damaging to the credibility of the aDNA field and its ability to reconstruct

population sizes. I hope the authors understand that before I can sign off on this part of the work, the inconsistencies below must be resolved or at least explicitly discussed.

We believe we share the referee's point of view: we want to get it right. We also believe that the peer-review process is very valuable to achieve this goal since researchers who are not co-authors of a specific study may spot aspects missed entirely by the co-authors of a given paper. That said, we were not expecting a fifth referee at this stage of the peer-review process. An additional referee would have been more expected if we had failed to address some of the previous points raised by the other four referees. Hence, while we very much appreciate the referee's time and very thorough work, we are also hoping that, especially at this point of the process, the focus is on our main conclusions and whether those conclusions are supported by the analyses we conducted.

The authors have already done an impressive range of cutting-edge bioinformatic analyses well beyond any typical aDNA paper. That's why I am confident they can promptly address my comments. In case extra questions arise, I am also happy to assist in any way the authors would find helpful.

Harald Ringbauer

A Critical Comments:

ROH (within individuals) and IBD segments (between individuals) feature centrally in the analysis, shown in Fig. 1 & 2. Both ROH and IBD calling are highly sensitive to genotype errors (e.g., even low rates of false heterozygotes or false opposing homozygotes, respectively, start breaking inferred segments up). Regarding ROH, the authors observe this themselves in their downsampling experiments.

For ROH, a recent advance specifically detects ROH in low-coverage aDNA (hapROH) and is designed to have as little false positive rate and length bias as possible. After the request of one other reviewer, the authors compared the results of their custom pipeline (using PLINK on imputed data) to hapROH results.

To clarify, one of the reasons we did not pick *hapROH* (a great tool developed by the referee himself) in the first place is that many of the 'Ancient Rapanui' genomes are not as low coverage as in other studies we have worked on (all 'Ancient Rapanui' genomes are above 0.4X, 3/15 range between 0.4 and 0.8X, 8/15 range between 1.2X and 3.7X, and 4/15 range between 5.6 and 25.6 X). Therefore, diploid genotype imputation should work well for all of them, including for ROH inference as we have shown (Sousa Da Mota et al. 2023; Allentoft et al. 2024). Furthermore, these are whole-genome shotgun sequencing data, which should allow us to get genome-wide information that is not biased by a capture experiment.

That analysis revealed some notable differences, in particular for long ROH:

The authors speak about „qualitatively similar results“. This is only true for the two conclusions of no close parental relatedness (offspring of most first cousins and even many second cousins would have mostly $>50\text{cM}$ of their genome in $\text{ROH}>20\text{cM}$, see Ringbauer 2021); and overall amount of ROH.

But the length distribution is markedly different! E.g., all 15 ancient Rapa Nui have $\text{ROH}>12\text{cM}$ in pseudo-haploid hapROH, while only 5/15 do for PLINK.

After conducting the additional analyses outlined below, we consider that our ROH results remain qualitatively similar—regardless of the inference method—in terms of our main conclusion: that the 'Ancient Rapanui' ROH length distribution is substantially different to that observed in populations with high consanguinity, e.g., the Paiteer Suruí. This is true regardless of the ROH inference method. However, we agree that we could have been more precise in the main text regarding the fact that the length distribution is different across methods so that the reader also sees that the reported absolute numbers should be considered with some caution.

We have now modified our main text to reflect that, and we have also updated our numbers regarding ROH lengths distributions as we have seen—thanks to the work requested by the referee—that the ROHs tend to be more precise when filtering out transitions. Line 218 now reads:

"To obtain a general overview of the genomic diversity on the island and potential inbreeding, we identified runs of homozygosity (ROH)⁴¹ in the 15 'Ancient Rapanui' genomes and a set of reference ancient and present-day worldwide genomes^{31,42,43} using PLINK⁴⁴ with imputed genotypes excluding transition polymorphisms and hapROH⁴⁵ with pseudohaploid data (SI11). Compared to other worldwide populations, 'Ancient Rapanui' carry a large proportion of their genomes in ROH, e.g., average length for Rapanui=204cM vs average length for Eurasia=18cM for PLINK (Figure2a, TableS16, see SI11 for hapROH results). However, most of those ROH are relatively short; in average, 92% of the total ROH are $<12\text{cM}$ and none of the 15 individuals carry a long proportion of their genome in long ROH. In contrast, in individuals with high consanguinity, e.g., the Paiteer Suruí from southern Amazon⁴⁶ and the Punjabi from Pakistan⁴⁷ a large proportion of ROH is long: the two Paiteer Suruí individuals carry more than 40% $\text{ROH}\geq 12\text{cM}$ ($>140\text{cM}$) and Punjabi-2 bears 18cM out of 23cM in $\text{ROH}\geq 12\text{cM}$. Interestingly, we observed that the 'Ancient Rapanui' ROH distribution is similar to that of a present-day Rapanui genome (P2077)⁴³. Nevertheless, the latter carries longer ROHs than the ancient individuals (39cM of the genome in $\geq 20\text{cM}$ runs). Furthermore, in contrast to the 'Ancient Rapanui', the ROH distribution among six other present-day Rapanui²² was highly variable (SI11.4). While the exact distribution of ROH depends on the inference method (hapROH infers longer ROH), the estimated inbreeding coefficients (SI10.1.2) and the relative differences between the ROH length distributions in

the 'Ancient Rapanui' and populations with a history of consanguinity support that the Rapanui have a low historical effective population size. Yet, unions between kins were seemingly infrequent pre-Peruvian slave raids and could have become more frequent in recent times⁴⁸. "

Importantly, these longer ROHs are precisely where one would read off signals of a recent bottleneck!

We address that point below, in our response to comment A7. The model that resembles the observed data most closely in our effective population size analyses includes a very strong bottleneck taking place around the time of the peopling of Rapa Nui. Regarding the hypothesised bottleneck in the 1600s, our intuition was that it is entirely possible to have 15/15 individuals with >12cm ROH without the population having undergone an extreme bottleneck in the previous few centuries (which is what we test extensively). We now test this with simulations and confirm our original intuition on that specific point (please see our reply to comment A7).

The authors prominently show the PLINK ROH length distribution in the main text (in Fig. 2), write about the difference in long ROH to present-day Rapa Nui, and explicitly state that "However, none of the 15 individuals carry excessively long ROH ($\geq 16\text{cM}$)" which strongly disagrees with hapROH.

As discussed above, we agree we should be more precise here, as we used to only discuss the *PLINK* results in the main text. We have now corrected our main text accordingly. Specifically, we now make it clear that the reported numbers are specific to *PLINK*.

Therefore, the actual ROH distribution would be crucial. However, the authors write in the SI: "While it is challenging to determine which of the three ROH sets is the most accurate...". Here I disagree.

I believe that in the high-coverage genomes, that long ROH should be very „obvious“.

One could zoom into the genetic data. For RNI014, where hapROH infers one ROH>20cm, one could zoom into this region using hapROH's functionality (see Fig. S7 from hapROH SI):

[FIGURE here]

For RNI014, a 25x genome (!), it should become highly evident where there is long ROH, and whether Plink ROH has an erroneous gap (because it does not merge gaps caused by low SNP density or sporadic erroneous SNP calls). On the other hand, hapROH might too

aggressively merge actual „gaps“ in that low Ne scenario, but that should become obvious too.

The „manual“ resolution of this discrepancy between hapROH and Plink should be discussed. If this Zoom-in reveals that hapROH is more accurate, it should be put in the prominent figures. The length distribution of long ROH matters, and one should show the „accurate“ one.

We want to reiterate that using either *hapROH* or *PLINK*-inferred ROH leads to similar conclusions in relative terms: we found the 'Ancient Rapanui' to be very different from populations for which inbreeding is known to be more common. Furthermore, ROH estimates for simulated data under a strong collapse scenario are different from what we observe in the 'Ancient Rapanui' (comment A7). But to further address the referee's concern, we now inferred ROH with *PLINK* using 1. all available sites in 1000 Genomes ('*PLINK* all sites 1KG'), 2. only transversions in 1000 Genomes ('*PLINK* transversions 1KG'), and 3. the sites resulting from intersecting 1000 Genomes and the 1240K capture sites ('*PLINK* 1240K'). Cumulative results are displayed in the new Figure S25 as follows:

Figure S25. HapROH and PLINK inferred runs of homozygosity (ROH) estimates for worldwide and 'Ancient Rapanui' genomes. Total lengths of ROH categorised by segment size for 11 present-day genomes, the two 'Ancient Polynesians' and the 15 'Ancient Rapanui', estimated with, from top to bottom, hapROH and pseudohaploid genomes PLINK and diploid genomes (imputed in the case of the ancient genomes) at the intersection of the

1000 Genomes and 1240K capture SNPs, PLINK and diploid genomes when restricting to transversions in the 1000 Genomes (1KG), and PLINK and diploid genomes for all SNPs in 1KG. Individual results are presented in Tables S17-20. "

For the 'Ancient Rapanui', the *hapROH* estimates tend to contain more $ROH \geq 12$ cM than the *PLINK* counterparts, but the total amount of ROH is reasonably similar across methods. In this updated figure, we show that 15/15 of the ancient Rapanui have $ROH \geq 12$ cM for *hapROH*, while we found 9/5, 11/15 and 5/15 for 'PLINK 1240K', 'PLINK transversions 1KG' and 'PLINK all sites 1KG', respectively. However, we point out that the *hapROH* and *PLINK* inferences for the present-day Paitei Suruí (a group with a small population size and high consanguinity) are substantially different (see below).

Following the referee's suggestion, we zoomed-in on the region where only *hapROH* found an $ROH \geq 20$ cM in RN14 (Figure R3). To further validate this analysis, we also inferred ROH using *PLINK* on called diploid genotypes, *i.e.*, not imputed for RN14 (the 'Ancient Rapanui' genome with the highest depth of coverage). It turns out that this region is at the very end of the chromosome and—after correcting an error in the conversion from base pairs to cM which impacted ROHs detected at the genetic map boundary—we detect the same ROH segment with *PLINK* when using the imputed data (all SNPs, transversions only and 1240K sites) and high-coverage data (called diploid genotypes on transversions SNPs only). An interesting observation from this additional work is that we tend to detect shorter segments on the high-coverage dataset (called genotypes) when transitions are included, even when considering the highest depth genome.

Figure R3. Zoom-in of ROH detected in chromosome 11 for RN14 (26X). We plot the ROH detected with, from top to bottom, *PLINK* and all available 1000 Genomes (1KG) sites for the high-coverage called genotypes ('Plink HC 1KG'), *PLINK* and all available 1000 Genomes sites for the high-coverage imputed genotypes ('Plink imputed HC 1KG'), *PLINK* and transversions sites in 1000 Genomes for the high-coverage called genotypes ('Plink HC Tv 1KG'), *PLINK* and transversions sites in 1000 Genomes for the high-coverage imputed genotypes ('Plink imputed Tv 1KG'), *PLINK* and the intersection between 1000 Genomes and 1240K capture sites for the high-coverage called genotypes ('Plink HC 1240K'), *PLINK* and the intersection between 1000 Genomes and 1240K capture sites for high-coverage imputed genotypes ('Plink imputed HC 1240K'), and *hapROH* for the pseudo-haploid version

of RN14 ('hapROH pseudo-haploid'). The ROH segments are coloured according to their size bin. At the bottom, we show the *hapROH* posterior probability in grey.

To further distinguish the methods, we plotted all the ROH detected with the different methods/data for all 'Ancient Rapanui' and some of the present-day individuals (a present-day Rapanui and the two Paiter Suruí individuals from the SGDP). For the Rapanui genomes, we found that, overall, the ROH estimated with *hapROH* and *PLINK* tend to agree, as shown, for example, on chromosome 5 in RN13, a 17X genome sequenced from UDG-treated libraries (Figure R4).

Figure R4. Zoom-in of ROH detected in chromosome 5 for RN5 (17x). We plot the ROH detected with, from top to bottom, *PLINK* and all available 1000 Genomes (1KG) sites for the high-coverage called genotypes ('Plink HC 1KG'), *PLINK* and all available 1000 Genomes sites for the high-coverage imputed genotypes ('Plink imputed HC 1KG'), *PLINK* and transversions sites in 1000 Genomes for the high-coverage called genotypes ('Plink HC Tv 1KG'), *PLINK* and transversions sites in 1000 Genomes for the high-coverage imputed genotypes ('Plink imputed Tv 1KG'), *PLINK* and the intersection between 1000 Genomes and 1240K capture sites for the high-coverage called genotypes ('Plink HC 1240K'), *PLINK* and the intersection between 1000 Genomes and 1240K capture sites for high-coverage imputed genotypes ('Plink imputed HC 1240K'), and *hapROH* for the pseudo-haploid version of RN13 ('hapROH pseudo-haploid'). The ROH segments are coloured according to their size bin. At the bottom, we show the *hapROH* posterior probability in grey.

However, we found some ROH disagreement between *hapROH* and *PLINK* estimates. In some cases, the level of agreement depended on the particular *PLINK* set, e.g., for some segments, '*PLINK* 1240K' and *hapROH* inferred ROH of similar length, that differed from '*PLINK* 1KG' and '*PLINK* transversions 1KG'. These differences make it harder to determine which method is better, i.e., differences may arise because of the chosen tool or because of the set of SNPs. See, for example, the ROH in RN12 chr4 that *hapROH* and '*PLINK* imputed 1240K' estimate to be longer than 16 cM, but '*PLINK* imputed Tv 1KG' estimates to be between 12 cM and 16 cM.

Figure R5. Zoom-in of ROH detected in chromosome 4 for RN12. We plot the ROH detected with, from top to bottom, *PLINK* and all available 1000 Genomes sites for the imputed genotypes ('Plink imputed 1KG'), *PLINK* and transversions sites in 1000 Genomes for the imputed genotypes ('Plink imputed Tv 1KG'), *PLINK* and the intersection between 1000 Genomes and 1240K capture sites for imputed genotypes ('Plink imputed 1240K'), and *hapROH* for the pseudo-haploid version of RN12 ('hapROH pseudo-haploid'). The ROH segments are coloured according to their size bin. At the bottom, we show the *hapROH* posterior probability in grey.

Nevertheless, some of these differences seemingly originate from an aggressive merge of ROH segments by *hapROH*. Below, we show here some examples where the posterior probability of *hapROH* seemingly indicates that a segment could be the result of an aggressive merge, while it was indeed shorter or absent from *PLINK* estimates:

Figure R6. Zoom-in of ROH detected in chromosome 2 for RN5 (17x). We plot the ROH detected with, from top to bottom, *PLINK* and all available 1000 Genomes (1KG) sites for the high-coverage called genotypes ('Plink HC 1KG'), *PLINK* and all available 1000 Genomes sites for the high-coverage imputed genotypes ('Plink imputed HC 1KG'), *PLINK* and transversions sites in 1000 Genomes for the high-coverage called genotypes ('Plink HC Tv 1KG'), *PLINK* and transversions sites in 1000 Genomes for the high-coverage imputed genotypes ('Plink imputed Tv 1KG'), *PLINK* and the intersection between 1000 Genomes and 1240K capture sites for the high-coverage called genotypes ('Plink HC 1240K'), *PLINK* and the intersection between 1000 Genomes and 1240K capture sites for high-coverage imputed genotypes ('Plink imputed HC 1240K'), and *hapROH* for the pseudo-haploid version of RN13 ('hapROH pseudo-haploid'). The ROH segments are coloured according to their size bin. At the bottom, we show the *hapROH* posterior probability in grey.

We also observe this in other lower coverage 'Ancient Rapanui' genomes.

Figure R7. Zoom-in of ROH detected in RN03 (chromosome 1), RN08 (chromosome 16), RN10 (chromosome 1), RN12 (chromosome 20) and RN15 (chromosome 3). For each panel, we plot the ROH detected with, from top to bottom, *PLINK* and all available 1000 Genomes sites for the imputed genotypes ('Plink imputed 1KG'), *PLINK* and transversions sites in 1000 Genomes for the imputed genotypes ('Plink imputed Tv 1KG'), *PLINK* and the intersection between 1000 Genomes and 1240K capture sites for imputed genotypes ('Plink imputed 1240K'), and *hapROH* for the pseudo-haploid version of the 'Ancient Rapanui' individual ('hapROH pseudo-haploid'). The ROH segments are coloured according to their size bin. At the bottom of each panel, we show the *hapROH* posterior probability in grey.

Given these differences we are not entirely sure what to conclude in terms of deciding which method is better for those cases.

As mentioned above, the largest discrepancies between *hapROH* and *PLINK* were surprisingly not found for ancient data but in the case of the two high-coverage present-day Paiteer Suruí genomes, whose error rates are expected to be low. In this case, *hapROH* detects very large ROH that is much shorter than *PLINK* estimates. We zoomed in on those genomes as well and show the results for chr13 and chr10. Here, the inferred ROH are shorter for *PLINK*, regardless of whether we restrict the analyses to 1240K or 1KG.

Figure R8. Zoom-in of ROH detected in Surui-1 (chromosome 13) and Surui-2 (chromosome 10). For each panel, we plot the ROH detected with, from top to bottom, *PLINK* and all available 1000 Genomes (1KG) sites for the high-coverage called genotypes ('Plink HC 1KG'), *PLINK* and the intersection between 1000 Genomes and 1240K capture sites for the high-coverage called genotypes ('Plink HC 1240K') and *hapROH* for the pseudo-haploid version of the two present-day Paiteer Suruí individuals ('hapROH pseudo-haploid'). The ROH segments are coloured according to their size bin. At the bottom of each panel, we show the *hapROH* posterior probability in grey.

In conclusion, our analyses show that *hapROH* and *PLINK* ROH estimates tend to agree. However, given that *hapROH* detected ROH in some cases that seemed too aggressively merged based on the posterior, and as we favour whole-genome data, we chose to display the *PLINK* results in the main text. However, we note that we continue to find it challenging to determine which method is best. However, we again emphasise that our main objective is to compare the ROH distribution in the 'Ancient Rapanui' with the ROH found in individuals from other populations. As such, given the differences discussed above, we consider that *PLINK* yields more accurate results, in particular for the high-coverage modern genomes, against whom we compare the 'Ancient Rapanui' genomes.

Importantly, thanks to zooming-in on the individual inferred ROH, we realised that aDNA *post-mortem* damage can substantially affect the inferred ROH when using all 1KG sites, as we show for ROH detected in chromosome 8 for RN14 (Figure S24). Here, the *PLINK* estimates match the *hapROH* estimate when filtering out the transitions. Therefore, for the revised manuscript, we decided to restrict our ROH analyses to transversion polymorphisms, which we consider has considerably improved our estimates. Figure S24 is displayed as follows

Figure S24. Zoom-in of ROH detected in chromosome 8 for RN14 (26X). We plot the ROH detected with, from top to bottom, PLINK and all available 1000 Genomes (1KG) sites for the high-coverage called genotypes ('Plink HC 1KG'), PLINK and all available 1000 Genomes sites for the high-coverage imputed genotypes ('Plink imputed HC 1KG'), PLINK and transversions sites in 1000 Genomes for the high-coverage called genotypes ('Plink HC Tv 1KG'), PLINK and transversions sites in 1000 Genomes for the high-coverage imputed genotypes ('Plink imputed Tv 1KG'), PLINK and the intersection between 1000 Genomes and 1240K capture sites for the high-coverage called genotypes ('Plink HC 1240K'), PLINK and the intersection between 1000 Genomes and 1240K capture sites for high-coverage imputed genotypes ('Plink imputed HC 1240K'), and hapROH for the pseudo-haploid version of RN14 ('hapROH pseudo-haploid'). The ROH segments are coloured according to their size bin. At the bottom, we show the hapROH posterior probability in grey. "

A2) A significant inconsistency is that ROH and IBD sharing should closely correspond, but as I outline do not currently. ROH is IBD within the two haplotypes of an individual. While a pair of diploid individuals have four possible haplotype combinations for haplotype sharing, ROH has only one possible combination. Therefore, total IBD sharing per pair should be 4x as much as ROH per individual. However, Fig. 1 shows that the ancient Rapa Nui have ca. 1200cm IBD>5cm per pair, but Fig. 2 shows that they only have ~150 cm ROH >4cm. That is far off from the expected factor of 4x, considering that >5cm cutoff is even higher than >4cm.

There are several explanations for this discrepancy. Most importantly, IBD calling has high rates of false positives (FP), „over-calling“ in length, or perhaps also that very short IBD overlap and „merged, “ which will be especially strong since the high amount of overall IBD sharing; see Chiang et al). For modern DNA, IBD calling around 5-6cm gets very tricky with high FP. Such High FP rates are especially a concern here as the imputed ancient data likely has different FP rates than using the modern DNA, which can cause batch effects.

I note that the authors could use ancIBD (Ringbauer et al. 2023) to double-check for longer IBD, as it is well calibrated to avoid FP >8cm (staying on purpose away for the „troublesome“ shorter IBD).

As this is a significant discrepancy between two leading figures (Fig. 1 and Fig. 2); it has to be explicitly discussed.

We thank the referee for the reminder concerning the difficulty in calling shorter IBD segments. We have followed the referee's suggestion and used *ancIBD* to confirm our analysis based on *IBDseq* calls. Since the reference panel distributed together with *ancIBD* is based on the 1240K SNP sites, we restricted this analysis to the ancient Polynesian individuals for which whole-genome data is available ('Ancient Rapanui' and 'Ancient

Polynesians') and not to Fijians or other Polynesians (from Tuvalu, Futuna, Tokelau, Samoa, Tonga, Niue, the Cook Islands and Rapa Nui) who were genotyped using a different SNP array. In brief, the IBD segment calls from both methods follow a similar trend. In particular, we observe a close correspondence for IBD segments longer than 15cM and, as anticipated by the referee, an excess of shorter segments called with *IBDseq* (Figures S20 and S21). In other words, it does seem that shorter IBD segments calls are not as robust (they depend on the methodology) as longer ones. Using the *ancIBD* IBD, we now also find that the ROH and IBD segments in the ancient Rapanui are consistent with the expectation that the total IBD sharing per pair of individuals is approximately four times the amount of ROH per individual, as shown below:

Figure R9. Boxplots of total IBD per pair of ancient Rapanui and ROH per individual. For the total amount of ROH, we show the estimates we obtained with *hapROH* and *PLINK*. To ease comparison between IBD and ROH amounts, we also represent four times the total ROH estimated with each tool. The individual datapoints are overlaid on the boxplots.

We emphasise that in this work, we rely on the geographic patterns of IBD sharing to support our MDS and *f*-statistics results showing that the 'Ancient Rapanui' are most closely related to present-day Rapanui and not to other Polynesians (most of them genotyped using a SNP array). Therefore, to keep as much publicly available reference data in our comparisons as possible, we kept the *IBDseq* results in the main text and figure. Yet, we relied on the relative amounts of *IBDseq* calls to reach our conclusion. To account for the robustness observation above, in the main results we now focus only on the IBD calls that are not method dependent. We have modified Figure 1b to include only segments >15cM, but we keep the full per-individual results in Figures S19 and S20 and Tables S14 and S15. Note that our conclusion showing that the 'Ancient Rapanui' share the largest IBD genomic fraction with present-day Rapanui than they do with other Polynesians is the same regardless of the

lengths of the segments including in the analyses. We now clarify this in Line 188, which now reads:

"Note that IBDseq operates on unphased genotypes and that Remote Oceanians are characterised by historically low effective population sizes, which is expected to give rise to extensive IBD sharing. Furthermore, calling shorter IBD segments accurately is challenging and can vary depending on the calling method (SI9). Thus, we focus on longer IBD segments (>15cM) that are not method dependent (SI9, FigureS21). Given these challenges, our reported IBD segment length values should not be interpreted in absolute but in relative terms, i.e., in the context of the comparisons shown in Figure1b. "

Figure 1 and its caption are displayed as follows:

"[...] **b.** Average IBD sharing between pairs of individuals from different Polynesian groups as estimated using *IBDseq*. For all possible pairs of individuals between two groups, e.g., the 45 possible pairs between the 15 'Ancient Rapanui' and the three present-day Rapanui, we show the average cumulative length of segments shared IBD, stratified by segment length (colour scheme). We show results for the 15 'Ancient Rapanui', three representative present-day Rapanui with low European admixture and two 'Ancient Polynesians' with unknown sampling location (see orange panel labels). For this analysis, we imputed the ancient individual sequence data to obtain diploid genotypes. Results for each individual (not pooled means) are presented in Figure S19 (including called IBD segments <15cM) and *IBDseq* estimates stratified by length are presented in Table S14. "

We included a description of the comparison in S19:

*"We used ancIBD⁷⁹ as a complementary method to evaluate the *IBDseq* calls. Since the reference panel distributed together with ancIBD is based on the 1240K SNP sites, we restricted this analysis to the ancient Polynesian individuals for which whole-genome data is available ('Ancient Rapanui' and 'Ancient Polynesians') and not to other Polynesians who were genotyped using a SNP array (Section S5.1). In Figure S20 (Table S15), we show per-individual IBD sharing similar to Figure S19. The IBD segment calls from both methods follow a similar trend. In particular, we observe a close correspondence for IBD segments longer than 15cM. We also observe a linear relationship between the short IBD segments called by *IBDseq* and ancIBD with the latter method calling systematically fewer shorter segments (Figure S21). Therefore, in Figure 1b we only present results for the IBD segments >15cM that are not method dependent. "*

Figures S20 and S21 and their captions are displayed as follows:

"**Figure S20.** IBD sharing between each ancient Polynesian individual for which whole-genome data is available ('Ancient Rapanui' and 'Ancient Polynesians'), as estimated using *ancIBD*. For each pair of individuals, we plot the cumulative length of the genome shared IBD across increasing IBD segment length bins. Individual *ancIBD* estimates stratified by length are presented in Table S15."

"**Figure S21**. Comparison between IBD segments called by IBDseq and ancIBD. For each pair of ancient Polynesian individuals, we plot the cumulative length of the genome shared IBD across increasing IBD segment length bins (see results for each method in Figures S19 and S20). For reference, we plot the $x=y$ line in black."

A3) In Fig. 2, only the ROH of one present-day Rapa Nui is shown. There are more samples available – why was this one selected? And one individual can be an outlier, so it would be much more informative to show ROH of multiple IIDs. The same holds for the other modern populations where only one or two individuals are depicted, but for modern Rapa Nui is an especially critical omission.

We had only included the ROH of a present-day genome, because this is the only present-day Polynesian whole genome included in our analyses, and it is therefore the only truly comparable data we have in our panel. However, now, we also inferred ROH for the present-day Polynesians in the SNP-array dataset (750K SNPs), including five present-day Polynesians and added this result in the SI. To compare with the Rapanui genomes (15 ancient genomes and one present-day genome), we restricted the data to the SNPs on the

array. With this new analysis, we observe that—unlike for 'Ancient Rapanui'—the ROH distribution in the present-day Rapanui is heterogeneous. In particular, we found one individual with a very high amount of long ROH, while others have very little. We describe these results in a new section S11.4:

S11.4. Comparing ROH in ancient and present-day Rapanui individuals

To compare ROH in ancient and present-day Rapanui, we inferred ROH for the Polynesians in the SNP-array dataset, which includes high-quality genetic data for five present-day Rapanui (that were not whole-genome amplified ^{46,88}). Additionally, we estimated ROH for the two imputed 'Ancient Polynesians', the 15 imputed 'Ancient Rapanui' and P2077, a high-coverage present-day Rapanui genome ²⁹, using the intersection of the 1000 Genomes and the SNP-array sites and PLINK as described in Section S11.1.1. In contrast to the ancient Rapanui individuals, the ROH sizes and total amounts were highly variable in the present-day Rapanui (Figure S26). For instance, P2083 had a total ROH of 29.4 cM (four segments in the $4 \leq \text{ROH} < 8$ cM size bin and one segment in the $8 \leq \text{ROH} < 12$ cM size bin), while P2094 had 582.7 cM, 41% of which were in $\text{ROH} \geq 20$ cM (241.0 cM), likely as a result of consanguinity. These individuals also had a variable amount of European-related ancestry, ranging between $< 1\%$ in P2080 and P2095 and 46% in P2083 (Section S13). P2083 and P2092, with more 40% of European ancestry, had the lowest amounts of ROH. For ROH in the $4 \leq \text{ROH} < 8$ cM size bin, the ancient individuals and the present-day Rapanui with limited European-like ancestry (P2080, P2095 and P2094) show similar amounts of ROH.

Figure S26. ROH inferred for Polynesians in SNP-array dataset. ROH inferred with PLINK for the Polynesian individuals in the SNP-array dataset, including five present-day Rapanui, and for genomes restricting to the overlap of 1000 Genomes sites and the SNP-array, namely, a present-day Rapanui (P2077), the two 'Ancient Polynesians' (Bot15 and Bot17) and the 15 'Ancient Rapanui'. Bars are coloured according to the length categories in the legend. "

A4) For the population size reconstruction, the authors use an LD signal - which implicitly uses the IBD/ROH signal.

I would ask the authors to explain why they did not even attempt to use IBD_Ne on called IBD signals. I suspect the authors did not trust the IBD calls enough (in light also of the above inconsistencies reasonable!). However, the authors should justify why they did not do it because IBD_Ne would be the established and “obvious” way to go about effective population size reconstruction.

IBD_ne is cited over 250 times, and widely used in present-day genomes (10.1016/j.ajhg.2015.07.012), while hapNe_LD is cited four times (because it is so new). If IBD calls would be perfect - IBD_Ne would arguably be the better method (see next point).

Following the referee's suggestion, we have added the following clarification in Line 248:

"We used HapNe-LD⁴⁹—which relies on linkage disequilibrium information and does not require high-confidence IBD calls—to reconstruct the Rapanui effective population size over the last 100 generations. "

In addition, we note that we opted for *HapNe-LD* since its original publication shows it outperforms *IBDNe* (which sometimes infers spurious wiggles (Browning and Browning 2015)) and *HapNe-IBD* when high-confidence IBD segment calls are not available (likely our case for shorter segments) (Fournier et al. 2023). More importantly, we note that we do not use the inferred *HapNe-LD* trajectories to provide an absolute estimate of the Ancient Rapanui population size. Instead, we use them as 'summary statistics' that we compare across simulations with a known population history to explore the collapse hypothesis. We now detail this in Line 274:

"We highlight that the inferred HapNe-LD curves are not intended for estimating absolute effective population sizes. Rather, we use them to discriminate population histories using simulations. "

A5) hapNe_LD uses an LD signal - however, the authors also use an LD signal to date admixture. That seems slightly odd at first glance - one uses a very similar signal to infer two different processes while ignoring the other.

As bottlenecked populations that likely kept admixing (continued East Polynesian contacts) and admixture with a continentally different ancestry (Native American) also produce strong LD - it is a priori not clear whether this influences the LD that hapNe_LD uses. The authors should comment on this, at least heuristically.

We agree with the referee that admixture would be expected to introduce variability in the population, thus potentially increasing effective population size estimates. It is with this possibility in mind that we opted for simulating data under known demographic scenarios to

test the collapse hypothesis, instead of directly interpreting the *HapNe-LD* curves. In these simulations, we include the inferred Native American admixture (SI12, Figure S28) and only varied the two bottleneck and growth rate parameters. In Line 260, we clarify this rationale:

"However, as a given HapNe-LD curve could ultimately correspond to more than one demographic scenario and the 'Ancient Rapanui' carry Native American admixture (see below), we also used extensive population genetics simulations⁵² to test whether this result could be consistent with a 1600s ecocide and collapse scenario. We simulated genome data assuming a population history compatible with the most recent findings for the Rapanui population (FigureS28) and characterised by five main parameters. "

In addition, we now simulated Rapanui-like genomes under a 'no Bottleneck 2' scenario without Native American admixture and found the *HapNe-LD* curves were qualitatively similar.

Figure R10. Simulations and *HapNe-LD* effective population size inference for 15 ancient genomes under a 'no Bottleneck 2' scenario with and without Native American admixture. Black lines correspond to 10 independent simulation replicates for each scenario. Purple lines correspond to the *HapNe-LD* results for the 15 'Ancient Rapanui' individuals.

Given that Polynesian and Native American populations are considerably differentiated, we expect ~10% Native American admixture to have a stronger effect in the Rapanui effective population size than, e.g., gene flow from closely related Polynesians from other islands. Although such contacts likely occurred, particularly close to the time of initial peopling of the island, we do not expect them to be common close to the time of the hypothesised 'ecocide' in the 1600s. Based on the documented observations by crew members of the Dutch expedition in 1722, the Rapanui only had 'leaky' canoes (Boersema 2015), thus complicating long distance travel. This absence of appropriate ships was perhaps related to the lack of wood caused by the deforestation that led to complete disappearance of trees in the 1600s (Hunt and Lipo 2011). Therefore, we consider that the Rapanui were not likely to be in contact with other peoples during the time in question, because 1) they presumably lacked

seafaring ships in the 18th century (~100 years after the hypothesized collapse), 2) Pitcairn and Mangareva, the closest inhabited Polynesian islands, are located ~2,000 km and ~2,600 km from Rapa Nui, respectively, and 3) we dated the Native American admixture event to centuries before the proposed collapse, *i.e.*, we find no evidence of later contacts.

Thus, we consider that admixture with other Polynesians is unlikely to mask a signal linked to a population collapse in the 1600s. Furthermore, we note that simulations with a growth rate ≥ 0.004 per year after Bottleneck 1 (peopling of the island) were markedly different from the observed data (FigureS30), thus suggesting that the Rapanui population did not increase at an unexpected rate (Boersema 2015) between the peopling of the island and the 1600s.

Finally, for the case of admixture dating, in (Moreno-Mayar et al. 2014) we tested the effect of a strong bottleneck on admixture dates and found that *tracts* estimates are essentially robust to a range of population bottlenecks.

We edited SI12.4 to discuss the possibility of the hypothesised population collapse being masked by gene flow from other regions, as shown below:

"Gene flow from other Polynesian islands or regions could potentially obscure evidence of the strong collapse that presumably took place in the 17th century. While we cannot say that such contacts did not occur, we believe that they were at most infrequent, as i) the Rapanui did not have seafaring boats (the Dutch expedition described their canoes as 'leaky' ~100 years after the hypothesised collapse ¹⁰⁴), ii) the nearest currently inhabited Polynesian islands are Pitcairn and Mangareva at ~2000km and ~2600km respectively, and iii) Native American gene flow occurred before the 17th century and is therefore unlikely to have masked the hypothesised population collapse. However, the question of how much travel took place between Rapa Nui and the rest of Polynesia may be answered in the future as more ancient genomes from the region are sequenced."

A6) A major worry of mine is that the bottleneck detection did not reveal a recent bottleneck in present-day data – despite the collapse to 110 individuals in the 19th century after the slave raids and epidemics, of which only 1/3 reproduced (and all present-day Rapa Nui claim as ancestors).

Yet, the authors do not detect any recent crashes using hapNe_LD in the modern data:

[FIGURE here]

Moreover, the ROH sharing of ancient and present-day Rapa Nui seems comparable – and the present-day individuals have less IBD sharing among themselves than the ancients.

The authors should discuss this more explicitly and explain why they think modest amounts of recent-admixture masked this signal. Was it too short of a population bottleneck to have an effect on IBD/LD? This, in turn, raises the question, would we detect a short ancient collapse?

It generally seems that hapNe_LD is very bad in detecting short sudden crashes, see e.g., main figure 2c:

[FIGURE here]

hapNe_LD inference seems to „oversmooth“ likely due to regularization. That’s why the authors do (impressive) additional simulations to match hapNe_LD curves. But that also is a bit of an odd approach, matching hapNe_LD inferred summary curves but not the actually underlying statistic (LD) – and I have a very bad intuition on how robust and sensitive that approach actually is.

In our study, we performed simulations to examine the estimates produced by *HapNe-LD* under different bottleneck scenarios because we wanted to determine whether a recent population bottleneck (the 'ecocide' hypothesis) could explain the *HapNe-LD* results. While *HapNe-LD* does not exactly reproduce the modelled population size changes, it has a very consistent behaviour in the presence of an old and a recent bottleneck: no recovery in population size (monotonic population decline). We agree with the referee that the absence of a population crash in the *HapNe-LD* curves for the present-day Rapanui is puzzling, given the very strong bottleneck in the 1860s.

To shed some light into the factors that could explain this result, we simulated present-day Rapanui data using the parameters that we found to fit our data the best. We investigated the effect of European admixture (17% (Moreno-Mayar et al. 2014)) and we varied the strength of the bottleneck. We found that admixture did not mask the collapse signature and, strikingly, the N_e estimates for simulated data were closest to the real data when there was no bottleneck. This analysis shows that 1) we could always detect a bottleneck when the simulations included one, and 2) we could not properly model the sampled present-day Rapanui individuals. In fact, this present-day sample is highly heterogeneous in terms of European-related admixture and ROH distribution (see our response to comment A3, SI11.4 and Figure S26), suggesting important changes in the recent history. The population of Rapa Nui reduced to 110 individuals as a result of the smallpox outbreak in the 1860s, but as of 2017 the island had a total of 7,750 inhabitants, which represents an average growth rate of 2.8% per year (10 times as much as the growth rate for exponential growth that we have considered before). These profound changes following the annexation of Rapa Nui by Chile in 1888, that include migration and admixture events complicate the modelling considerably and probably mask the population collapse.

We edited section S12.2.2.2 as follows to explicitly discuss this point:

"Given these parameters ($Sb1=0.1$, $Tb1=52$ generations, $Sb2=1.0$, $Tb2=21$ generations, $\alpha=0.002 \text{ year}^{-1}$), we simulated eight present-day Rapanui-like genomes to understand why we could not detect the population crash associated with the smallpox outbreak in the 1860s that reduced the population size from 4000 to 110 individuals⁷⁴, i.e., 2.8% of the population survived. We simulated genomes (10 replicates) with and without a population collapse in the 1860s (Bottleneck 3, $Tb3=4$ generations) and with and without European admixture (17% as in⁴⁶) following the collapse when there was one.

Regardless of European admixture, when we modelled a strong collapse of $Sb3=0.1$ (10% survival), the recent effective population size trajectories were first constant and then decreased until the present (Figure S31a,b). We then varied the strength of the bottleneck while keeping the European admixture event. We found that the demographic model that yields the population size trajectories that are most similar to that inferred for the real data is the one without a bottleneck in the 1860s (Figure S31c and Figure S32). Given that a bottleneck of considerable strength ($Sb3=\{0.1, 0.3, 0.5\}$) always had an effect on the effective population size and that the smallpox collapse is a well-documented event, we believe that we could not properly model the population from which the present-day Rapanui individuals originate.

After the 19th century, Rapa Nui was no longer isolated, and there was an influx of migrants, leading to significant gene flow from other populations. In fact, the genetic clustering results for these eight individuals show a high variability in ancestry, with a European-like component above 40% in two of the individuals (Figure S36). This lack of homogeneity is also reflected in the ROH distribution (Figure S26). In other words, we believe that, when it comes to modelling N_e , our model does not capture the main features of the sample of present-day individual genome-wide data we had access to, which in turn probably explains why we do not detect the population crash that almost decimated the Rapanui. However, ultimately, future work and data should help explain why we cannot detect the more recent bottleneck in present-day data. "

Figure S31. Recent effective population size for simulated present-day Rapanui-like genomes in models with European admixture and recent population collapse as result of the smallpox outbreak. Using the simulation parameters we found to yield the most similar population size (N_e) curves to the one obtained for the ancient Rapanui genomes, we inferred population size with HapNe-LD when: **a.** a strong bottleneck ($Sb3=0.1$) takes place followed by 17% of European gene flow into the population; **b.** a strong bottleneck and no European admixture; **c.** no bottleneck and 17% European admixture. The N_e estimates from real present-day and ancient Rapanui genetic data are represented in blue and purple, respectively, and the estimates for the simulated data are represented in black. The first row contains the logarithm base 10 of the population size as a function of generations and the second row has the slope of these curves (non-logarithmic scale). "

Figure S32. Effect of a 'smallpox bottleneck' strength on the effective population size of simulated present-day individuals in the presence of European admixture. We

varied the strength (Sb_3) of the 'smallpox bottleneck' between 0.1 (10% of the individuals survive) and 1.0 (no bottleneck) (a-e). The demographic model includes 17% of European gene flow one generation after the bottleneck. We simulated each set of parameters with 10 replicates (black curves). Present-day and ancient Rapanui N_e estimates are depicted in blue and purple, respectively. The first row contains the N_e trajectories (logarithmic scales) and the second row their rate of change. "

We refer to these new analyses in the main text in Line 293:

"See S12.2.2.2 for a discussion on the challenges for modelling present-day Rapanui population size trajectories due to their heterogeneous ancestry profiles. "

A7) In light of the above points, I am worried that putting a critical result on hapNe_LD curves alone might miss some crucial signal for population size reconstructions.

As I believe that hapROH ROH curves 4-20cm should be very accurate (and the authors hopefully can double-check that), and the ROH curve based on 15 individuals seems very robustly inferrable from 15 IIDs in such low N_e scenarios. Therefore, I would suggest one final „double-check“:

Namely, simulating ROH of several individuals under the best-fit model and comparing ROH to the empirically inferred ROH in the ancient Rapa Nui. If that looks like a good match, I would be convinced about the „steadily increasing “ scenario.

I note that the authors already have their simulations set up – and it is relatively easy to extract ROH from msprime output.

We inferred ROH from the simulated genome data. In this case, we varied the *Bottleneck 2* strength (Sb_2) and fixed the remaining four parameters to their 'best' values. Based on a graphical inspection of the distributions of total amount of ROH (SROH) across Sb_2 values, we observed that the SROH distributions in the ancient Rapanui genomes (*hapROH* and *PLINK*-inferred ROH) did not match a very strong *Bottleneck 2* scenario ($Sb_2=0.1$ and $Sb_2=0.2$). Even though there is a trend towards greater SROH with increasing *Bottleneck 2* strength, the SROH distributions could not be easily distinguished for $Sb_2 \geq 0.3$ and the 'Ancient Rapanui' distribution seemed to fall in that range. We include these results in the new section S12.3 *Summary statistics to test the 'ecocide' hypothesis* for which we include the new figures and description below:

S12.3.2 Results

*We analysed the distribution of total amount of ROH (SROH) when simulating a population with a similar history as the Rapanui, but with a *Bottleneck 2* with varying strength.*

Regardless of the minimum ROH length that we considered for computing SROH, we found that SROH decreases with decreasing bottleneck strength (Figure S33 and Figure S34). A strong bottleneck with $Sb2=0.1$ (only 10% of the population survives) stood out as the most distinct, yielding median SROH values of 364 cM and 75 cM for ROH with at least 4 cM and 16 cM, respectively, while a bottleneck with $Sb2=0.5$ (50% of the population survives) had median SROH of 185 cM and 18 cM for the same ROH thresholds, respectively.

We compared the SROH distributions from the simulated data with the SROH distribution for the imputed 'Ancient Rapanui' genomes as obtained using PLINK and hapROH. Since hapROH yielded comparatively longer ROH than PLINK, we use its estimates as an upper bound of SROH in the ancient individuals. We found the hapROH and PLINK-inferred SROH to be similar, with differences arising at a minimum ROH of 8 cM, in which case hapROH SROH was greater. For $SROH \geq 4$ cM, hapROH and PLINK had identical distributions that were most similar to SROH in simulated data with $Sb2=0.3$, $Sb2=0.5$ and $Sb2=0.7$. For higher ROH thresholds, the PLINK SROH distribution matches better the $Sb2=0.7$ (weak Bottleneck 2) and $Sb2=1.0$ (no Bottleneck 2) SROH distributions. In conclusion, based on the whole hapROH and PLINK SROH distributions, the SROH in the 'Ancient Rapanui' could only be distinguished from SROH of simulated genomes whose populations underwent very strong bottlenecks of $Sb2=0.1$ and $Sb2=0.2$. This can be partly explained by SROH distributions in simulated genomes with $Sb2$ equal to 0.3, 0.5, 0.7 and 1.0 not differing much. However, we note that their respective HapNe-LD curves can verify that the simulated genomes with $Sb2=1.0$ are the closest to the observed data. "

In addition, we include the following in the discussion section SI12.4:

"We also found the total amount of ROH distributions for data simulated under a very strong Bottleneck 2 ($Sb2=0.1$ and $Sb2=0.2$) to be clearly distinct from the observed data, thus allowing further rejection of the 'ecocide' scenario. Jared Diamond estimated that, prior to the collapse in the 1600s, there were 15,000 Rapanui inhabiting the island⁹⁰ and, if we consider the 1,500-3,000 population-size estimates for the 18th century (based on the European visitors' records¹⁰⁴) as the number of individuals left after the collapse, this would represent a collapse with a strength between 0.1 and 0.2, that we can confidently reject with our effective population size and ROH analyses. "

Finally, we refer to these analyses in the main text in Line 286:

"We used the SROH (total ROH in an individual) distribution as an alternative summary statistic to the HapNe-LD curves (SI12.3). We found that the 'Ancient Rapanui' SROH distribution did not match scenarios involving a very strong Bottleneck 2 ($Sb2 \leq 0.2$) (Figures S33-34). "

Figure S33. Comparing total amount of ROH (SROH) in 'Ancient Rapanui' and in simulated genomes while varying Bottleneck 2 strength (Sb_2). Boxplots of SROH for simulated genomes with Sb_2 between 0.1 and 1.0 and the imputed ancient genomes ('PLINK' and 'hapROH'), where horizontal lines represent, from bottom to top, the first quartile, the median and the third quartile, and the whiskers lengths are 1.5 times the interquartile range. For each value of Sb_2 , 15 genomes were simulated from 10 populations

simulated with the same parameters, that is, 10 replicates, and points from the same simulation (for the same Sb_2 value) are coloured equally with their respective medians in the same colour (the short horizontal lines). "

"Figure S34. Using runs of homozygosity (ROH) and recent effective size (Ne) as statistics to test the 'ecocide' hypothesis. The first three columns contain the density of total amount of ROH (SROH) when considering a minimum size of 4 cM, 12 cM and 16 cM, respectively. The third column contains Ne estimates obtained with HapNe-LD. Density (hapROH and PLINK-inferred ROH in yellow and blue, respectively) and Ne (purple) curves estimated with the imputed 'Ancient Rapanui' data are shown on the first row and throughout, while the remaining rows include the same estimates for simulated genomes with varying Bottleneck 2 strength (Sb2). We produced 10 replicates for each set of parameters."

Major Comments:

B1) Surprisingly, the authors did not call Y haplogroups! This is a standard analysis in the aDNA field - and should really be done as it could reveal further strong evidence regarding the admixture. Uniparental haplogroups are effectively only one locus - but can be highly diagnostic of continental ancestry and, moreover give first insights into sex-biased processes.

The authors called 15 mtDNA haplogroups, which are typical of East Polynesians (the authors should also mention this in the main text). That can happen simply by chance even if one assumes ca. 10% Native American ancestry - but it also raises a curiosity about the Y haplogroups.

We agree with the referee that uniparental markers are relevant in this case, particularly as a starting point for studying sex-biased demographic processes. Following the referee's suggestion, we called Y-chromosome haplogroups for the eight 'Ancient Rapanui' individuals using *pathPhynder*. We describe the analysis in a new Supplementary Section S4.4.4. and report our results in Table S1. In brief, we found all XY individuals cluster with individuals carrying Y-chromosome haplogroup C1b2a, which supports the Polynesian origin of the sequenced individuals.

We note that in this revision, we have tried to cut the main text to get closer to *Nature's* format. Therefore, we have not included results from uniparental markers in the main text to be able to prioritise results based on whole-genome analyses. However, haplogroup assignments are reported in Table S1, and we expect they will be a starting point for a follow-up study characterising potential sex-bias in the Rapanui-Native American admixture event.

New Supplementary Section 4.4.4. reads as follows

"S4.4.4. 'Ancient Rapanui' Y-chromosome haplogroups

We used *pathPhynder*⁴¹ and its associated ISOGG Y-chromosome reference data (last curated 13 May, 2021) to place the XY 'Ancient Rapanui' individuals into a phylogeny including a worldwide set of Y-chromosomes. Since a number of individuals are represented by low-depth data, this strategy allowed us to prevent potential miscalls (caused by missing diagnostic SNPs) by callers designed for low-missingness present-day data. We ran *pathPhynder* restricting to transversion polymorphisms and set the remaining parameters to their default values. *pathPhynder* haplogroup assignments are reported in Table S1. The eight XY individuals were placed together with individuals carrying haplogroup C1b2a1c. Haplogroup C1b is most widely present in East Asia and Oceania^{37,42,43}. Thus, these

assignments, together with the mitochondrial DNA haplogroups and the genomic ancestry analyses (see below) support the Polynesian origin of the 15 ancient individuals. "

B2) The authors used standard aDNA methods (based on average pairwise mismatch rates, PMR) to detect up to second-degree relatives and did not find any. However, IBD can go further – and the authors have such data in their hands. Notably, the SI shows that RN05 and RN12 have ca. 400 cm IBD >35cm (Fig. S19), standing out clearly from the rest of the ancient pairs! That clearly points to this pair being relatives ca. 3-4th degree (see Ringbauer et al. 2023). This pair already has the lowest lowest PMR in Fig. S20.

This exception also proves the rule that none of the other pairs are related up to the ~4th degree – which further improves the value of this dataset (as the ROH/IBD signal is consistent in a set of non-related individuals, showing how „general“ the signal is!). I believe that the manuscript would benefit from explicitly stating that.

We thank the referee for noting this interesting result. We have included this observation in the main text in Line 208:

"Using READ³⁹ and NGSrelate⁴⁰, we did not find any first- or second-degree relatives among the 'Ancient Rapanui' (Figures S22,23, S110). These results are supported by the highest estimated IBD>15cM sharing between any two individuals being <1,000cM (S19, TableS14). "

And in Supplementary Section 9:

"Furthermore, the estimated IBD sharing distribution supports that all individuals are unlikely to be related to each other as third degree (or closer) relatives ⁷⁹ (Section 10). The highest sharing (>15cM) between any two individuals (RN05 and RN12) is ~900cM, which could indicate they are third- or fourth-degree relatives."

Minor Comments

C1) The authors run standard contamination estimates based on mtDNA and X chromosomes in males. Based on these results, I am quite sure there is generally little contamination.

Notably, the fact that the authors can call long ROH in all ancient individuals indirectly shows a lack of substantial contamination (which would „destroy“ ROH). One could apply hapCon_ROH (introduced in Posth et al 2022, now publicly available, see hapROH website) to directly measure that and to obtain robust autosomal contamination estimates. This tool is usually a gimmick, as typical ancient individuals do not have the necessary coverage or ROH stretches. But here, both are given to all 15 ancient individuals!

We appreciate all the useful methods developed by the referee himself, which are great contributions to the field. However, since the referee states he is not worried about this aspect—we are not concerned either given our reported contamination estimates—unlike for the IBD and ROH analyses where we accounted for the referee's suggestion to also include analyses using the referee's software, we rather not repeat our contamination analyses with a third software (also developed by the referee).

C2) Several recent works that use aDNA to reconstruct ancient population sizes could be cited for context (e.g., Waldmann et al 2022, or Sirak et al 2021).

We looked into both of the suggested publications on the population history of medieval German Jews and Christian Period Nubians, co-authored by the referee. Although we agree with the referee that adding citations to similar works would improve the contextual information in our manuscript, due to formatting constraints and following Referee 2's and the editor's suggestion, we are trying to bring the main text closer to *Nature's* format. Thus, we would prefer to prioritise keeping as much text as possible that is central to our work, results and interpretations.

References

- Allentoft ME, Sikora M, Refoyo-Martínez A, Irving-Pease EK, Fischer A, Barrie W, Ingason A, Stenderup J, Sjögren K-G, Pearson A, et al. 2024. Population genomics of post-glacial western Eurasia. *Nature*. 625(7994):301–311. <https://doi.org/10.1038/s41586-023-06865-0>
- Boersema JJ. 2015. *The Survival of Easter Island: Dwindling Resources and Cultural Resilience*. 1st ed. Webb D, translator. [place unknown]: Cambridge University Press. <https://doi.org/10.1017/CBO9781139226639>
- Browning SR, Browning BL. 2015. Accurate Non-parametric Estimation of Recent Effective Population Size from Segments of Identity by Descent. *The American Journal of Human Genetics*. 97(3):404–418. <https://doi.org/10.1016/j.ajhg.2015.07.012>
- Fehren-Schmitz L, Jarman CL, Harkins KM, Kayser M, Popp BN, Skoglund P. 2017. Genetic Ancestry of Rapanui before and after European Contact. *Current Biology*. 27(20):3209–3215.e6. <https://doi.org/10.1016/j.cub.2017.09.029>
- Fournier R, Tsangalidou Z, Reich D, Palamara PF. 2023. Haplotype-based inference of recent effective population size in modern and ancient DNA samples. *Nat Commun*. 14(1):7945. <https://doi.org/10.1038/s41467-023-43522-6>
- Hunt TL, Lipo CP. 2011. *The statues that walked: unraveling the mystery of Easter Island*. 1st Free Press hardcover ed. New York: Free Press.
- Ioannidis AG, Blanco-Portillo J, Sandoval K, Hagelberg E, Barberena-Jonas C, Hill AVS, Rodríguez-Rodríguez JE, Fox K, Robson K, Haoa-Cardinali S, et al. 2021. Paths and timings of the peopling of Polynesia inferred from genomic networks. *Nature*. 597(7877):522–526. <https://doi.org/10.1038/s41586-021-03902-8>
- Ioannidis AG, Blanco-Portillo J, Sandoval K, Hagelberg E, Miquel-Poblete JF, Moreno-Mayar JV, Rodríguez-Rodríguez JE, Quinto-Cortés CD, Auckland K, Parks T, et al. 2020. Native American gene flow into Polynesia predating Easter Island settlement. *Nature*. 583(7817):572–577. <https://doi.org/10.1038/s41586-020-2487-2>
- Moreno-Mayar JV, Rasmussen S, Seguin-Orlando A, Rasmussen M, Liang M, Flåm ST, Lie BA, Gilfillan GD, Nielsen R, Thorsby E, et al. 2014. Genome-wide Ancestry Patterns in Rapanui Suggest Pre-European Admixture with Native Americans. *Current Biology*. 24(21):2518–2525. <https://doi.org/10.1016/j.cub.2014.09.057>
- Moreno-Mayar JV, Vinner L, de Barros Damgaard P, de la Fuente C, Chan J, Spence JP, Allentoft ME, Vimala T, Racimo F, Pinotti T, et al. 2018. Early human dispersals within the Americas. *Science*. 362(6419):eaav2621. <https://doi.org/10.1126/science.aav2621>
- Reich D, Patterson N, Campbell D, Tandon A, Mazieres S, Ray N, Parra MV, Rojas W, Duque C, Mesa N, et al. 2012. Reconstructing Native American population history. *Nature*. 488(7411):370–374. <https://doi.org/10.1038/nature11258>

Sousa Da Mota B, Rubinacci S, Cruz Dávalos DI, G. Amorim CE, Sikora M, Johannsen NN, Szmyt MH, Włodarczak P, Szczepanek A, Przybyła MM, et al. 2023. Imputation of ancient human genomes. *Nat Commun.* 14(1):3660. <https://doi.org/10.1038/s41467-023-39202-0>

Reviewer Reports on the Second Revision:

Referee #2 (Remarks to the Author):

The authors have addressed all my comments and from my perspective the paper is ready.

Referee #3 (Remarks to the Author):

I have carefully reviewed the revised version of the manuscript titled "Ancient Rapanui genomes reveal resilience and pre-European contact with the Americas." and compared it to the original manuscript, along taking into account the additional material provided.

The authors have meticulously addressed the concerns raised during the previous round of review.

I have been focusing on the radiocarbon dating and Bayesian modelling, and now they have significantly improved the clarity and rigor of the manuscript. In particular, the statistical analyses have been appropriately presented. I have no further comments to add on these aspects.

Additionally, I understand and appreciate the decision of the authors to keep certain paragraphs separate in the Supplementary Information.

In my opinion, the paper now meets the standards for publication.

Overall, I find that the points raised in the previous round of review have been satisfactorily addressed.

Referee #5

The authors have substantially revised their analysis and writing. By comparing to well-tested haplotype-sharing tools, the authors identified spurious signals (e.g., false positives of *IBDseq* for shorter IBD; aDNA damage breaking up long ROH in *PLINK*). Those edits have resolved the inconsistencies and effectively addressed my previous concerns. Therefore, I now strongly recommend this extraordinary article for publication.

- 1) Importantly, by focussing on longer shared haplotypes in the main figures (both for IBD and ROH segments), the authors have improved the power of those results and, importantly, vastly reduced the noise.
Sharing IBD>15cm (now depicted in Fig. 1) is a strong and precise signal of recent genealogical connections (with almost no technical uncertainty). The substantial sharing of IBD on that level (and longer) between the ancient and present-day Rapa Nui individuals is impressive; there is no room for doubt now.
- 2) The authors identified ways to improve their ROH calling pipeline, so the ancient Rapa Nui ROH distribution now more closely resembles hapROH's.
- 3) The additional ROH length distribution checks strongly confirm the absence of a very strong recent bottleneck, adding an independent line of evidence for a central claim ("no major ecocide").

I only have a few minor comments left on accuracy and text improvements. I stress that none of those suggestions are make-or-break, as the article's main claims are backed up in the current form.

I know this late round of comments has been frustrating for the authors, but I sincerely hope they agree that their edits substantially improved their cutting-edge analysis.

Major Comment:

A) The authors implicitly assume that hapROH does not work as well for modern genomes (or at least worse than their PLINK). This assumption is surprising because if something works for low-coverage aDNA, it will likely also work well for less problematic modern high-coverage data. To showcase this point, I provide direct evidence using public versions of the two Surui genomes they used below.

Rebuttal:

"As mentioned above, the largest discrepancies between hapROH and PLINK were surprisingly not found for ancient data but in the case of the two high-coverage present-day Paiter Suruí genomes, whose error rates are expected to be low. In this case, hapROH detects very large ROH that is much shorter than PLINK estimates."

[...]

"However, we note that we continue to find it challenging to determine which method is best. However, we again emphasise that our main objective is to compare the ROH distribution in the 'Ancient Rapanui' with the ROH found in individuals from other populations. As such, given the differences discussed above, we consider that PLINK yields more accurate results, in particular for the high-coverage modern genomes, against whom we compare the 'Ancient Rapanui' genomes"

(Figure R8, Rebuttal)

Those two Surui genomes are in the AADR, and one can run (using hapROH diploid mode) and visualize those two chromosomes (using heterozygote genotypes) - here are the results of a brief analysis:

There is definitive evidence that the long ROHs are not multiple ROHs (observe the blue dots, which are the raw signal).

The evidence for PLINK splitting up true long ROH is overwhelming. This does not affect any central claim in the paper, so I leave it to the authors to choose if they want to be as accurate as possible in their prominent figure 1.

Minor Comments:

B) L192-196:

“Thus, we focus on longer IBD segments (>15cM) that are not method dependent (SI9, FigureS21). Given these challenges, our reported IBD segment length values should not be interpreted in absolute but in relative terms, i.e., in the context of the comparisons shown in Figure1b.”

In your revision, you show that inferred IBD sharing >15cm is in agreement between IBD and IBDseq —and ancIBD guarantees you excellent calls in that length regime - there is no room for technical error. Therefore, those inferred IBD values are very close to the truth. The

“relative terms” phrase is, at best, overcautious and, at worst, discourages readers from looking at the “true” signal.

Perhaps you refer to shorter IBD from deeper relatedness (although even that is mostly a non-issue for so long IBD >15cm), but it is unclear from the current text version.

C) L209-212

Using READ and NGSrelate, we did not find any first- or second-degree relatives among the 'Ancient Rapanui' (Figures S22,23, SI10). These results are supported by the highest estimated IBD>15cM sharing between any two individuals being <1,000cM (SI9, TableS14).”

You have strong evidence (there is little doubt) - for one single 3-4th 3-degree relative and nothing else on that level or closer. You could mention this in the main text (rather than the technical <1000cm, which you could also lower this value if you remove the relative pair by stating it before).

Also, READ and NGSrelate are effectively the same signals (pairwise genotype mismatches), so you can only cite one method in the main (as you do for IBD - where you go some length to avoid mentioning two methods).

D) At first glance, it is surprising that the eight males and the 15 ancient individuals all carry specific Polynesian Y and mtDNA haplogroups, giving the 10% admixture estimate. But that is well within statistical noise. You could explicitly state that in the SI to prevent commenters from misinterpreting that.

Author Rebuttals to Second Revision:

We thank the referees for providing their feedback. Please find below our detailed response to the referees' comments (in black), our replies (in blue), and specific changes to manuscript text (*in italics*). Note that cited lines correspond to line numbers in the newly submitted manuscript.

Referee #2 (Remarks to the Author):

The authors have addressed all my comments and from my perspective the paper is ready.

We thank the referee for their overall review.

Referee #3 (Remarks to the Author):

I have carefully reviewed the revised version of the manuscript titled "Ancient Rapanui genomes reveal resilience and pre-European contact with the Americas." and compared it to the original manuscript, along taking into account the additional material provided. The authors have meticulously addressed the concerns raised during the previous round of review.

I have been focusing on the radiocarbon dating and Bayesian modelling, and now they have significantly improved the clarity and rigor of the manuscript. In particular, the statistical analyses have been appropriately presented. I have no further comments to add on these aspects.

Additionally, I understand and appreciate the decision of the authors to keep certain paragraphs separate in the Supplementary Information.

In my opinion, the paper now meets the standards for publication.

Overall, I find that the points raised in the previous round of review have been satisfactorily addressed.

We thank the referee for their overall review.

Referee #5 (Remarks to the Author):

The authors have substantially revised their analysis and writing. By comparing to well-tested haplotype-sharing tools, the authors identified spurious signals (e.g., false positives of IBDseq for shorter IBD; aDNA damage breaking up long ROH in PLINK). Those edits have resolved the inconsistencies and effectively addressed my previous concerns. Therefore, I now strongly recommend this extraordinary article for publication.

Importantly, by focussing on longer shared haplotypes in the main figures (both for IBD and ROH segments), the authors have improved the power of those results and, importantly, vastly reduced the noise.

Sharing IBD>15cm (now depicted in Fig. 1) is a strong and precise signal of recent genealogical connections (with almost no technical uncertainty). The substantial sharing of IBD on that level (and longer) between the ancient and present-day Rapa Nui individuals is impressive; there is no room for doubt now.

The authors identified ways to improve their ROH calling pipeline, so the ancient Rapa Nui ROH distribution now more closely resembles hapROH's.

The additional ROH length distribution checks strongly confirm the absence of a very strong recent bottleneck, adding an independent line of evidence for a central claim ("no major ecocide").

We agree with this summary, and we think in particular that, even though the overall conclusions have not changed, our IBD and ROH analyses have substantially improved; we especially thank the referee for the feedback on those points that helped reduce the noise/technical issues in the IBD analyses, make the IBD and ROH analyses comparable to each other as well as make our ROH analysis more consistent between modern and ancient datasets (see also below).

I only have a few minor comments left on accuracy and text improvements. I stress that none of those suggestions are make-or-break, as the article's main claims are backed up in the current form.

I know this late round of comments has been frustrating for the authors, but I sincerely hope they agree that their edits substantially improved their cutting-edge analysis.

Our main problem was that we were a bit exhausted by the fact that the comments came rather late. However, we thank the referee for taking the time to read through our work carefully and for his expert feedback on ROH and IBD. It also 'feels right' from a scientific perspective that the referee is now satisfied with the results we present in the paper.

Major Comment:

A) The authors implicitly assume that hapROH does not work as well for modern genomes (or at least worse than their PLINK). This assumption is surprising because if something works for low-coverage aDNA, it will likely also work well for less problematic modern high-coverage data. To showcase this point, I provide direct evidence using public versions of the two Surui genomes they used below.

We were favouring the approach which used 'more of the data' - since we used genotype calls for the modern data in PLINK - and we also generally feel more comfortable with whole-genome data than 1240K data.

Rebuttal:

“As mentioned above, the largest discrepancies between hapROH and PLINK were surprisingly not found for ancient data but in the case of the two high-coverage present-day Paiteer Suruí genomes, whose error rates are expected to be low. In this case, hapROH detects very large ROH that is much shorter than PLINK estimates.”

[...]

“However, we note that we continue to find it challenging to determine which method is best. However, we again emphasise that our main objective is to compare the ROH distribution in the 'Ancient Rapanui' with the ROH found in individuals from other populations. As such, given the differences discussed above, we consider that PLINK yields more accurate results, in particular for the high-coverage modern genomes, against whom we compare the 'Ancient Rapanui' genomes”

[FIGURE HERE]

(Figure R8, Rebuttal)

Those two Surui genomes are in the AADR, and one can run (using hapROH diploid mode) and visualize those two chromosomes (using heterozygote genotypes) - here are the results of a brief analysis:

[FIGURE HERE]

There is definitive evidence that the long ROHs are not multiple ROHs (observe the blue dots, which are the raw signal).

The evidence for PLINK splitting up true long ROH is overwhelming. This does not affect any central claim in the paper, so I leave it to the authors to choose if they want to be as accurate as possible in their prominent Figure 2.

We thank the reviewer for sharing his ROH analysis for the Paiteer Suruí genomes. To understand better where the discrepancy came from, we ran some additional analyses.

First, instead of using the variants provided in <https://reichdata.hms.harvard.edu/pub/datasets/sgdp/> in *PLINK* format and then running our custom *PLINK* approach, as we proceeded before ('*PLINKv0*' in Figure R1 below), we called genotypes directly from the corresponding SGDP bam files at the 1000 Genomes bi-allelic SNP sites using *bcftools* ('*PLINK new calls*' in Figure R1). We inferred ROH with *PLINK* from these newly called dataset.

Second, we generated an imputed dataset after generating genotype likelihoods for the modern genomes' bam files. We then inferred ROH with *PLINK* for the imputed version of the modern genomes ('PLINK imputed' in Figure R1).

We found that the resulting ROH are considerably more similar between *hapROH* and our *PLINK* approach in the regions we had zoomed-in before. In other words, the discrepancy between the *hapROH* and *PLINK* ROH estimates seems to come largely from the dataset we used before and much less from the methods or approaches themselves.

Specifically, zooming into the problematic regions, we found that the ROH inferred on the newly called data yielded a longer ROH (≥ 20 cM) in chromosomes 13 and 10 for Suruí-1 and Suruí-2, respectively (see Figure R1 below). However, the *PLINK* estimates were still shorter than the *hapROH* estimates: 25cM vs. 35cM (Suruí-1) and 28cM vs. 36cM (Suruí-2). In contrast, *PLINK*-inferred ROH for the imputed data and *hapROH* estimates were practically identical for the abovementioned regions.

Figure R1. Comparison of *hapROH* and *PLINK*-inferred ROH across two chromosomes in the genomes of two Paiter Suruí individuals (SGDP dataset). ROH segments are coloured according to the length category they fall into (ROH size bins are the same as in Figure 2a). We plotted ROH estimates using *PLINK* and *hapROH*, from top to bottom: i) *PLINK*-inferred ROH for the original dataset ('PLINKv0'), *PLINK* ROH estimates for the newly called diploid genotypes for ii) all available 1000 Genomes bi-allelic sites ('PLINK new calls'), and iii) the intersection of these with the 1240K SNP dataset ('PLINK new calls 1240K'), iv) *PLINK*-inferred ROH for the imputed version of the newly called genotypes ('PLINK imputed'), v) *hapROH* in diploid mode estimates for the imputed genotypes ('hapROH diploid'), and vi) *hapROH* in haploid mode results ('hapROH haploid').

Even though the results are very similar in those regions in chr10 and chr13, the total ROH length remains substantially greater for *hapROH* than for *PLINK*-inferred ROH even when using imputed genotypes as input data for *PLINK* (Figure S26 below). We are still not entirely

sure which of the results is closer to the truth, and it seems that 1240K versus whole genome data, but also to some extent the methodology, has an impact.

That said, we find it preferable to start from the bam files for both modern and ancient genomes to have more consistent results between modern and ancient individual samples. So, we thank the referee for pointing us in this direction.

For consistency, we now moved the *hapROH* (haploid mode) plots to the main text (Figure 2a) for this revision. Furthermore, we feel more comfortable with our *hapROH* results having better understood the source of the discrepancy between the two approaches. We edited the main text to include *hapROH* estimates and updated the Supplementary Information file (section SI10).

Line 221 now reads:

'To obtain a general overview of the genomic diversity on the island and potential inbreeding, we identified runs of homozygosity (ROH) in the 15 'Ancient Rapanui' genomes and a set of reference ancient and present-day worldwide genomes^{32,42,43} using hapROH⁴⁴ with pseudohaploid data and PLINK⁴⁵ with imputed genotypes excluding transition polymorphisms (SI10). Compared to other worldwide populations, 'Ancient Rapanui' carry a large proportion of their genomes in ROH, e.g., average length for Rapanui=198cM vs average length for Eurasia=25cM for hapROH (Figure2a, TableS18, see SI10 and TableS19 for PLINK results). However, most of these ROH are relatively short; on average, 80% of the total ROH are <12cM and none of the 15 individuals carry a long proportion of their genome in long ROH. In contrast, in individuals with high consanguinity, e.g., the Paiteer Suruí from southern Amazon⁴⁶ and the Punjabi from Pakistan⁴⁷ a large proportion of ROH is long: the two Paiteer Suruí individuals carry more than 76% ROH \geq 12cM (>335cM) and Punjabi-2 bears 18cM out of 30cM in ROH \geq 12cM. Interestingly, we observed that the 'Ancient Rapanui' ROH distribution is similar to that of a present-day Rapanui genome (P2077)⁴³. Nevertheless, the latter carries longer ROHs than the ancient individuals (38cM of the genome in \geq 20cM runs). Furthermore, in contrast to the 'Ancient Rapanui', the ROH distribution among six other present-day Rapanui5 was highly variable (SI10.4). While the exact distribution of ROH depends on the inference method (PLINK infers shorter ROH), the estimated inbreeding coefficients (SI9.1.2) and the relative differences between the ROH length distributions in the 'Ancient Rapanui' and populations with a history of consanguinity support that the Rapanui have a low historical effective population size. Yet, unions between kins were seemingly infrequent pre-Peruvian slave raids and could have become more frequent in recent times⁴⁸. '

Updated FigureS26 is now displayed as follows:

Figure S26. HapROH and PLINK inferred runs of homozygosity (ROH) estimates for worldwide and 'Ancient Rapanui' genomes. Total lengths of ROH categorised by segment size for 11 present-day genomes, the two 'Ancient Polynesians' and the 15 'Ancient Rapanui', estimated with, from top to bottom, hapROH and pseudohaploid genomes, hapROH and (imputed) diploid genomes, PLINK and imputed diploid genomes at the

intersection of the 1000 Genomes and 1240K capture SNPs, and PLINK and diploid genomes when restricting to transversions in the 1000 Genomes. Individual results are presented in Tables S18-21.

Updated Figure 2 is now displayed as follows:

Figure 2 | Runs of homozygosity (ROH) and Rapanui population size estimates through time. **a.** Total ROH stratified by length in worldwide present-day and ancient Polynesian genomes (SI 10). **b.** HapNe-LD effective population size estimates for 15 imputed 'Ancient Rapanui' genomes. Assuming the ancient individuals were born in ~1800CE (Figure 4) and 29 years/generation, we indicate the latest estimate for the peopling of Rapanui (1250CE), and the 1600s collapse proposed by the 'ecocide' theory. **c.** msprime coalescent-based simulations and HapNe-LD effective population size inference for 15 ancient genomes under a model with two bottlenecks followed by growth. The oldest bottleneck (Bottleneck 1) represents the peopling of Rapa Nui. The more recent (Bottleneck 2) represents the 'ecocide' theory collapse. Bottlenecks are defined by a time ($Tb1, Tb2$) and a strength ($Sb1, Sb2$) parameter. Strengths indicate the proportion of the population left after each bottleneck, e.g., a bottleneck with strength 0.1 is very strong (10% of the population is left). We assume the population grows exponentially with rate α after each bottleneck. We compared estimates for the observed and simulated data across a grid of bottleneck and growth parameters ($Tb1, Sb1, Tb2, Sb2$ and α). The heatmap shows a measure of the difference between the effective population size estimates for observed (panel b) and simulated data for a set of representative simulation parameters, across 10 replicates (full range; SI 11). We consider different times for each bottleneck ($Tb1, Tb2$) and

three strengths: 0.1 (strong), 0.5 (intermediate) and 1.0 (non-existent). $\alpha=0.002$ is the growth rate that minimised the difference between the inferences for observed and simulated data. For reference, we plot the estimates for two simulations ('One strong bottleneck' and 'Two strong bottlenecks'). Black lines correspond to 10 independent simulation replicates for each parameter set. Shaded areas show 95% bootstrap confidence intervals.

Minor Comments:

B) L192-196:

“Thus, we focus on longer IBD segments (>15cM) that are not method dependent (SI9, FigureS21). Given these challenges, our reported IBD segment length values should not be interpreted in absolute but in relative terms, i.e., in the context of the comparisons shown in Figure1b.”

In your revision, you show that inferred IBD sharing >15cm is in agreement between IBD and IBDseq—and ancIBD guarantees you excellent calls in that length regime - there is no room for technical error. Therefore, those inferred IBD values are very close to the truth. The “relative terms” phrase is, at best, overcautious and, at worst, discourages readers from looking at the “true” signal.

Perhaps you refer to shorter IBD from deeper relatedness (although even that is mostly a non-issue for long IBD >15cm), but the current text version does not clarify this.

We agree with the referee. We have removed this sentence from our revised manuscript.

C) L209-212

Using READ and NGSrelate, we did not find any first- or second-degree relatives among the 'Ancient Rapanui' (Figures S22,23, SI10). These results are supported by the highest estimated IBD>15cM sharing between any two individuals being <1,000cM (SI9, TableS14).”

You have strong evidence (there is little doubt) - for one single 3-4th 3-degree relative and nothing else on that level or closer. You could mention this in the main text (rather than the technical <1000cm, which you could also lower this value if you remove the relative pair by stating it before).

Also, READ and NGSrelate are effectively the same signals (pairwise genotype mismatches), so you can only cite one method in the main (as you do for IBD - where you go some length to avoid mentioning two methods).

Following the referee's suggestion, we have now moved this result in the main text. Line 212 now reads:

Using READ³⁹ and NGSrelate²⁴⁰, we did not find any first- or second-degree relatives among the 'Ancient Rapanui' (Figures S22,23, SI10) and detected a single pair of third-fourth-degree relatives based on IBD sharing (highest estimated IBD>15cM sharing between any two individuals is <1,000cM (SI9, TableS14)).

We actually did not intend to avoid mentioning any method, this was the result of revising together the main and the supplement and trying to keep the main text short. As part of this revision, in addition to the supplement, we now include a summary of the methodology in a new Methods section where we cite all the methods we used for data analysis in the main text as well.

D) At first glance, it is surprising that the eight males and the 15 ancient individuals all carry specific Polynesian Y and mtDNA haplogroups, giving the 10% admixture estimate. But that is well within statistical noise. You could explicitly state that in the SI to prevent commenters from misinterpreting that.

We have now added the following note to Supplementary Section S3.4.4.

Although we only detect Polynesian (and no Native American) mtDNA and Y-chromosome haplogroups in the 'Ancient Rapanui', we consider these results are within statistical noise based on the low Native American admixture proportion ~10% (Section S13,14). We anticipate that as more ancient genomes from the region are sequenced, a fuller picture of the Polynesian uniparental genetic pool will become available.

Note: After receiving the last round of comments from the referees, we were granted access to the SNP array data from Ioannidis et al. 2020 and Ioannidis et al. 2021 on 28 May, 2024. In our latest revised manuscript, we included an additional set of f3 statistics using these data (SI7.2, Figure S12, Table S5). In brief, we find that the 'Ancient Rapanui' share the most drift with present-day Rapanui individuals, among a wider set of Oceanian populations. This result further supports that the 15 ancient individuals sequenced in this study are Rapanui in origin. Furthermore, they are in line with a scenario of population continuity in the island despite the strong population decline and population movements that resulted from European contact and the Peruvian slave raids during the 1860s.